# Transducing Language Models

**Vésteinn Snæbjarnarson**[*,Q,R]    **Samuel Kiegeland**[*,Q,X]    **Tianyu Liu**[Q]
**Reda Boumasmoud**[Q]    **Ryan Cotterell**[Q]    **Tim Vieira**[Q]
[Q]ETH Zürich    [R]University of Copenhagen    [X]CHI-FRO
{vest.snae, tim.f.vieira}@gmail.com    reda.boumasmoud@math.ethz.ch
{samuel.kiegeland, tianyu.liu, ryan.cotterell}@inf.ethz.ch

## Abstract

Modern language models define distributions over strings, but downstream tasks often require different output formats. For instance, a model that generates byte-pair strings does not directly produce word-level predictions, and a DNA model does not directly produce amino-acid sequences. In such cases, a deterministic string-to-string transformation can convert the model's output to the desired form. This is a familiar pattern in probability theory: applying a function $f$ to a random variable $X \sim p$ yields a transformed random variable $f(X)$ with an induced distribution. While such transformations are occasionally used in language modeling, prior work does not treat them as yielding new, fully functional language models. We formalize this perspective and introduce a general framework for language models derived from deterministic string-to-string transformations. We focus on transformations representable as finite-state transducers—a commonly used state-machine abstraction for efficient string-to-string mappings. We develop algorithms that compose a language model with an FST to *marginalize* over source strings mapping to a given target, propagating probabilities through the transducer without altering model parameters and enabling *conditioning* on transformed outputs. We present an exact algorithm, an efficient approximation, and a theoretical analysis. We conduct experiments in three domains: converting language models from tokens to bytes, from tokens to words, and from DNA to amino acids. These experiments demonstrate inference-time adaptation of pretrained language models to match application-specific output requirements.

⌂ https://github.com/rycolab/transducing-language-models

## 1 Introduction

Language models (LMs) define distributions over strings. Yet, the strings they produce often do not match the requirements of downstream applications, so practitioners resort to ad hoc post-processing. We call this the **string mismatch problem**. For example, in natural language processing, modern language models typically generate byte-pair encoded strings (Sennrich et al., 2016), while downstream tasks may require words or characters instead (see Ex. (2), below). Similarly, DNA language models generate nucleobase sequences, whereas many applications require amino acid sequences (Ex. (5)).

Adding a string-to-string transformation to a generation pipeline is a common engineering solution, such as normalizing output or mapping subword tokens to bytes. Formally, this defines a new language model over *transformed* strings. However, while sampling remains straightforward, other operations—such as computing the probability of a transformed string or conditioning on transformed outputs—become intractable. Consider, for instance, the mapping from a string in any casing to its lowercase version, as in the use-case depicted in Fig. 1. While lowercasing a given input is trivial, converting the original distribution to a distribution over lowercased words is not.

This work treats string-to-string transformations as a first-class component of the language modeling pipeline. We show how to equip these transformed models with the familiar autoregressive interface—incremental next-symbol distributions and prefix probabilities—making them interoperable with

---

[*]Equal contribution.

any system built for standard autoregressive language models. This approach to the string mismatch problem is principled, modular, and can be substantially cheaper than retraining. Moreover, the transformations often guarantee adherence to the requirements of the downstream applications.

Consider language models over English text—the same utterance can be encoded in many ways. For example,

(1)    *Dr. Lemaître was flabbergasted.* 🤯

The byte-pair encoding used by GPT-4o (OpenAI, 2024) encodes Ex. (1) as the following string of subword tokens:

(2)
```
Dr    .   _L   ema   \C3\AEtre   _was   _fl   ab
5822  13  451  4603  29135       673    1548  378
berg  asted  .   _\F0\9F\A4  \AF
9667  23030  13  93643       107
```

The byte-pair segmentation of Ex. (1) into subwords is based on character substring frequency. However, many applications seek different units. For instance, computational psycholinguistics (e.g., Giulianelli et al., 2024) and controlled generation (e.g., Lew et al., 2023; Xefteri et al., 2025) both require custom units. This also holds if one wishes to derive distributions over words, such as those defined by the Penn Treebank (PTB) annotation guidelines (Marcus et al., 1993), a variation of which is shown below:

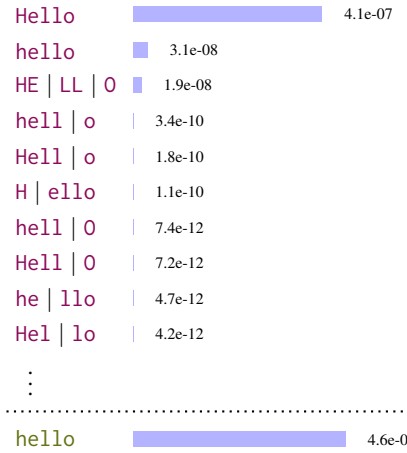

Figure 1: GPT-2 probabilities for the BPE `token strings` that, when lowercased, match `hello`. The total probability of all such sequences is at the bottom.

(3)    DR.    LEMAÎTRE    WAS    FLABBERGASTED    .    🤯

For other applications, e.g., spelling correction, we might also wish to represent Ex. (2) using a string of characters or UTF-8 byte representation as follows:[1]

(4)
```
D  r  .  _  L  e  m  a\C3\AE  t  r  e  _  w  a  s  _  f  l  a  b  b  e  r  g  a  s  t  e  d  .  _\F0\9F\A4\AF
68 114 46 32 76 101 109 97 195 174 116 114 101 32 119 97 115 32 102 108 97 98 98 101 114 103 97 115 116 101 100 46 32 240 159 164 175
```

In genetics, we have another example of varying representations. Consider the DNA sequence given in Ex. (5). The sequence is one of many that translate into the hormone *oxytocin*, typically written as the amino acid sequence in Ex. (6), as represented below:

(5)    T G T T A C A T A C A A A A T T G T C C T C T A G G T        (6)    C Y I Q N C P L G

Transforming a language model is generally non-trivial. Even transformations with simple colloquial descriptions can be hard to apply. The conversion to bytes (ex. (4)) can be performed using a lookup table that specifies the sequence of bytes a subword token maps to. Similarly, the conversion from DNA to protein sequences (ex. (5)) relies on mappings from three-letter sequences of DNA bases to a single amino acid. The grammatical word segmentation (ex. (3)) cannot be described as easily; it requires *lookahead* to determine whether, e.g., punctuation should stand alone, if it is part of the 'DR.' title, or a decimal number '3.50'. Simple rules alone do not ensure exact conversions. Despite their simplicity, the byte and amino-acid transformations are not straightforward: the number of token sequences that map to each output sequence grows exponentially over the length of the output, and a proper language model transformation must account for all source-sequence probabilities.[2] The decimal and title examples highlight another level of complexity, where the transformation rules require careful analysis of the surrounding context. Recent papers have thus focused on practical methods for specific subsets of these transformations, in particular for estimating byte-level probabilities from subword models (Phan et al., 2024; Vieira et al., 2025a; Hayase et al., 2025).[3] We generalize these approaches to handle all of the examples given above.

This work introduces a framework for string-to-string conversion using *finite-state transducers* (FSTs). FSTs encode a powerful yet tractable family of relations, including those mentioned above. We com-

---

[1]Note that UTF-8 allows multiple encodings of some strings. For example, the character î can be encoded *composed* (\C3\AE) or *decomposed* (i\CC\82). See https://unicode.org/reports/tr15/.

[2]This can also be seen in Fig. 1, where an exponentially increasing number of input prefixes map to a given output prefix: with each additional symbol on the output, the number of source sequences doubles.

[3]Our approach is most similar to that of Vieira et al. (2025a), which considers the case of *strict-prefix monotone transformations* (§6), which includes the subword-to-byte transformation.

pose pretrained language models with transducers that encode such transformations, and refer to the compositions as *transduced language models*. FSTs provide explicit structure for tracing how probabilities from the original model should map to output sequences. This allows us to develop exact and approximate algorithms for efficient sampling, scoring, and conditioning on transformed strings, all without modifying the underlying language model. We give sufficient conditions for when the transformations can be made exactly (§6) and approximations when exact transformations are infeasible (§5).

To validate our approach, we construct FSTs for the three use cases above: (i) converting tokens to bytes, (ii) inserting orthographic boundaries following the Penn Treebank tokenizer, and (iii) converting DNA sequences to sequences over amino acids. We then employ commonly used pretrained language models over the input units of the FSTs, and compose them with the FSTs to obtain language models over the output tokens. Finally, we use these settings to benchmark the theoretical and algorithmic contributions. In particular, we find that using a practical approximation is sufficient to obtain a good estimate at a fraction of the computational cost.

## 2 BACKGROUND[4]

**Strings.** Let $\mathcal{X}$ be an **alphabet** (i.e., a finite, non-empty set). Let $\mathcal{X}^*$ denote the set of all finite strings over $\mathcal{X}$, including the empty string $\varepsilon_{\mathcal{X}}$. When there is no risk of ambiguity with other alphabets, we simply write $\varepsilon$. We use $\boldsymbol{x}, \boldsymbol{x}' \in \mathcal{X}^*$ to denote strings and $Z, Z' \subseteq \mathcal{X}^*$ to denote sets of strings. Let $\boldsymbol{xx}'$ denote **concatenation**. Similarly, we define the concatenation of sets of strings as $ZZ' \stackrel{\text{def}}{=} \{\boldsymbol{xx}' \mid \boldsymbol{x} \in Z, \boldsymbol{x}' \in Z'\}$, and in the singleton case as $\boldsymbol{x}Z' \stackrel{\text{def}}{=} \{\boldsymbol{x}\}Z'$, and $Z\boldsymbol{x}' \stackrel{\text{def}}{=} Z\{\boldsymbol{x}'\}$. We write $\boldsymbol{x} \preceq \boldsymbol{x}'$ when $\boldsymbol{x}$ is a **prefix** of $\boldsymbol{x}'$, and $\boldsymbol{x} \prec \boldsymbol{x}'$ when it is a **strict-prefix**. Conversely, if $\boldsymbol{x} \preceq \boldsymbol{x}'$, we say $\boldsymbol{x}'$ is an **extension** of $\boldsymbol{x}$ (a **strict extension** when $\boldsymbol{x} \prec \boldsymbol{x}'$).

**Language models.** A **language model** $p_{\mathcal{X}}$ is a probability distribution over a set of strings $\mathcal{X}^*$. Let $\text{EOS} \notin \mathcal{X}$ be a special **end-of-string symbol**. We define the **prefix probability** of $p_{\mathcal{X}}$ as the probability that a string $X \sim p_{\mathcal{X}}$ starts with a given prefix $\boldsymbol{x}$, and the **conditional prefix probability**:

$$\overrightarrow{p_{\mathcal{X}}}(\boldsymbol{x}) \stackrel{\text{def}}{=} \sum_{\boldsymbol{x}' \in \mathcal{X}^*} p_{\mathcal{X}}(\boldsymbol{xx}') \qquad \overrightarrow{p_{\mathcal{X}}}(\boldsymbol{x}' \mid \boldsymbol{x}) \stackrel{\text{def}}{=} \frac{\overrightarrow{p_{\mathcal{X}}}(\boldsymbol{xx}')}{\overrightarrow{p_{\mathcal{X}}}(\boldsymbol{x})} \qquad \overrightarrow{p_{\mathcal{X}}}(\text{EOS} \mid \boldsymbol{x}) \stackrel{\text{def}}{=} \frac{p_{\mathcal{X}}(\boldsymbol{x})}{\overrightarrow{p_{\mathcal{X}}}(\boldsymbol{x})} \quad (1)$$

when $\overrightarrow{p_{\mathcal{X}}}(\boldsymbol{x}) > 0$; otherwise we set $\overrightarrow{p_{\mathcal{X}}}(\boldsymbol{x}' \mid \boldsymbol{x}) \stackrel{\text{def}}{=} 0$ and $\overrightarrow{p_{\mathcal{X}}}(\text{EOS} \mid \boldsymbol{x}) \stackrel{\text{def}}{=} 1$. Therefore, $\overrightarrow{p_{\mathcal{X}}}(\cdot \mid \boldsymbol{x})$ is a probability distribution over $\mathcal{X} \sqcup \{\text{EOS}\}$ for all $\boldsymbol{x} \in \mathcal{X}^*$. Using this structure, any language model $p_{\mathcal{X}}$ may be factorized as $p_{\mathcal{X}}(\boldsymbol{x}) = \overrightarrow{p_{\mathcal{X}}}(\text{EOS} \mid \boldsymbol{x}) \prod_{t=1}^{|\boldsymbol{x}|} \overrightarrow{p_{\mathcal{X}}}(x_t \mid \boldsymbol{x}_{<t})$. This factorization defines a left-to-right generative process: starting from $\boldsymbol{x} = \varepsilon$, we repeatedly sample $x' \sim \overrightarrow{p_{\mathcal{X}}}(\cdot \mid \boldsymbol{x})$; if $x' = \text{EOS}$, we stop, otherwise we update $\boldsymbol{x}$ to $\boldsymbol{x}x'$. Conditional generation simply starts from the conditioning prefix instead of the empty string. We refer to the quantities in Eq. (1) as the **autoregressive interface** to the language model $p_{\mathcal{X}}$.

**Cylindrical sets.** A cylindrical set is the set of all strings with a given prefix. Let $Z, Z' \subseteq \mathcal{X}^*$; we define the **cylinder** over $Z$ as $\langle Z \rangle \stackrel{\text{def}}{=} Z\mathcal{X}^*$. We say that $Z$ is **cylindrical** if $Z = \langle Z \rangle$. The union of cylinder sets is again a cylinder set, since cylinders are upward-closed. We define the **basic cylinder** for $\boldsymbol{x}$ as $\langle \boldsymbol{x} \rangle \stackrel{\text{def}}{=} \langle \{\boldsymbol{x}\} \rangle$. The **prefix-base** operation $\text{pf}(Z)$ is defined by $\text{pf}(Z) \stackrel{\text{def}}{=} \{\boldsymbol{x} \in Z : \nexists \boldsymbol{x}' \in Z : \boldsymbol{x}' \prec \boldsymbol{x}\}$; this operation uniquely partitions $\langle Z \rangle$ into basic cylinders over $\text{pf}(Z)$. We say that $Z$ is **prefix-free** if $\text{pf}(Z) = Z$. Since $\text{pf}(Z)$ is prefix-free, the basic cylinders $\{\langle \boldsymbol{x} \rangle \mid \boldsymbol{x} \in \text{pf}(Z)\}$ are pairwise disjoint, and $\langle Z \rangle = \bigsqcup_{\boldsymbol{x} \in \text{pf}(Z)} \langle \boldsymbol{x} \rangle$.

**Transducers.** A **transducer** is a state-machine that encodes string-to-string relations $f \subseteq \mathcal{X}^* \times \mathcal{Y}^*$. When we express a relationship defined by $f$ as a transducer, we expose the computational structure needed to develop efficient algorithms. Formally, a **finite-state transducer**[5] (**FST**) $f$ is a tuple $(S, \mathcal{X}, \mathcal{Y}, I, F, T)$ where $S$ is a finite set of **states** and $\mathcal{X}$ and $\mathcal{Y}$ are alphabets of **input** and **output** symbols, respectively. The sets $I, F \subseteq S$ are the **initial** and **accepting** states. $T \subseteq S \times (\mathcal{X} \cup \{\varepsilon\}) \times (\mathcal{Y} \cup \{\varepsilon\}) \times S$ is a set of **transitions**. We render transitions $(s, x, y, s') \in T$ as $s \xrightarrow{x:y} s'$; we say the transition **scans** $x$ and **emits** $y$. We write $T(s)$ for the set of outgoing transitions from state $s$, and $T(s, x)$ for those that scan $x$.[6] The transducer $f$ defines a set of **paths** $\Pi$. Each

---

[4] §A provides a notation glossary.

[5] We refer to Pin (2021, Ch. 2 & 3) for a detailed treatment of transducers.

[6] I.e., $T(s'') \stackrel{\text{def}}{=} \{(s \xrightarrow{x:y} s') \in T : s = s''\}$, and $T(s'', x'') \stackrel{\text{def}}{=} \{(s \xrightarrow{x:y} s') \in T : s = s'', x = x''\}$.

**path** $\pi \in \Pi$ is a sequence of transitions of the form $s_0 \xrightarrow{x_1:y_1} s_1 \xrightarrow{x_2:y_2} s_2 \cdots s_{N-1} \xrightarrow{x_N:y_N} s_N$. We call $\pi$ an **accepting path** if $s_0 \in I$ and $s_N \in F$. The **relation defined** by $f$ is given by $[\![f]\!] \stackrel{\text{def}}{=} \{(\boldsymbol{x}, \boldsymbol{y}) \mid s_0 \xrightarrow{x_1:y_1} s_1 \cdots s_{N-1} \xrightarrow{x_N:y_N} s_N \in \Pi \colon s_0 \in I, s_N \in F\}$, i.e., each accepting path contributes (not necessarily uniquely) a pair of scanned and emitted strings. When every transition of a transducer scans and emits the same symbol ($x = y$), the machine acts as a **finite-state automaton** (**acceptor**) that recognizes a **language** $L \subseteq \mathcal{X}^*$—the set of strings admitted by at least one accepting path. Such a machine is a **nondeterministic finite automaton** (**NFA**) in general; it is a **deterministic finite automaton** (**DFA**) if it has a single initial state, no $\varepsilon$-transitions, and at most one transition per state and input symbol. Every NFA can be converted to an equivalent DFA by **determinization** (Rabin & Scott, 1959). For completeness, §B provides additional background.

## 3 TRANSDUCED LANGUAGE MODELS

A **transduced language model** $p_{\mathcal{Y}}$ arises from applying a string-to-string **transformation** $f \colon \mathcal{X}^* \to \mathcal{Y}^*$, encoded by a transducer $f$, to a string drawn from a **source language model** $p_{\mathcal{X}}$. Formally, if $X \sim p_{\mathcal{X}}$, then $f(X)$ has the following probability mass function:

$$p_{\mathcal{Y}}(\boldsymbol{y}) \stackrel{\text{def}}{=} \Pr_{X \sim p_{\mathcal{X}}}[\boldsymbol{y} = f(X)] = \sum_{\boldsymbol{x} \in f^{-1}(\boldsymbol{y})} p_{\mathcal{X}}(\boldsymbol{x}) \tag{2}$$

where $f^{-1}(\boldsymbol{y})$ is the **preimage** of $\boldsymbol{y}$, $f^{-1}(\boldsymbol{y}) \stackrel{\text{def}}{=} \{\boldsymbol{x} \in \mathcal{X}^* \colon \boldsymbol{y} = f(\boldsymbol{x})\}$. Put differently, in Eq. (2), we sum over the strings $\boldsymbol{x}$ such that $f(\boldsymbol{x}) = \boldsymbol{y}$. Unfortunately, evaluating $p_{\mathcal{Y}}(\boldsymbol{y})$ exactly using Eq. (2) is generally infeasible (since the preimage $f^{-1}(\boldsymbol{y})$ can be very large), even though exact sampling from $p_{\mathcal{Y}}$ is efficient (by sampling $\boldsymbol{x} \sim p_{\mathcal{X}}$ and applying $f$). Like all language models, a transduced language model $p_{\mathcal{Y}}$ has prefix and conditional prefix probability functions; its prefix probability is

$$\overrightarrow{p_{\mathcal{Y}}}(\boldsymbol{y}) = \Pr_{X \sim p_{\mathcal{X}}}[\boldsymbol{y} \preceq f(X)] = \sum_{\boldsymbol{x} \in \mathcal{P}(\boldsymbol{y})} p_{\mathcal{X}}(\boldsymbol{x}) \tag{3}$$

where $\mathcal{P}(\boldsymbol{y})$ is the **precover** of $\boldsymbol{y}$, with respect to $f$, defined as $\mathcal{P}(\boldsymbol{y}) \stackrel{\text{def}}{=} \{\boldsymbol{x} \in \mathcal{X}^* \colon \boldsymbol{y} \preceq f(\boldsymbol{x})\}$.[7]

Prefix probabilities yield a conditional factorization of string probability (see §2), enabling efficient left-to-right autoregressive generation. We develop a method in §4 that allows us to compute the sum in Eq. (3) in *finite* time for a general class of mappings, such as those mentioned in the introduction (i.e., normalizing text, inserting orthographic word boundaries, or converting DNA to amino-acid sequences). In §5, we present algorithms to compute these quantities.

## 4 DECOMPOSING THE PRECOVER

In §3, we saw that if we can sum over the precover of $\boldsymbol{y}$, we can calculate $\overrightarrow{p_{\mathcal{Y}}}(\boldsymbol{y})$ (Eq. (3)), unlocking an autoregressive interface to the transduced language model. The following two examples illustrate how we can often compute this infinite sum by exploiting structural properties of the transducer.

**Example 1.** *The transducer below lowercases a string. For the target string* ab, *the precover is the infinite set* $\mathcal{P}(\text{ab}) = \langle\{\text{AB}, \text{Ab}, \text{aB}, \text{ab}\}\rangle$. *Given a model* $p_{\mathcal{X}}$ *over the input language, the derivation on the right (4a–4c) applies Eq.* (3) *to express* $\overrightarrow{p_{\mathcal{Y}}}(\text{ab})$ *as a sum of four source-language prefix probabilities:*

$$\overrightarrow{p_{\mathcal{Y}}}(\text{ab}) = \sum_{\boldsymbol{x} \in \mathcal{P}(\text{ab})} p_{\mathcal{X}}(\boldsymbol{x}) \tag{4a}$$

$$= \sum_{\boldsymbol{x}' \in \{\text{AB,Ab,aB,ab}\}} \sum_{\boldsymbol{x} \in \langle \boldsymbol{x}' \rangle} p_{\mathcal{X}}(\boldsymbol{x}) \tag{4b}$$

$$= \overrightarrow{p_{\mathcal{X}}}(\text{AB}) + \overrightarrow{p_{\mathcal{X}}}(\text{Ab}) + \overrightarrow{p_{\mathcal{X}}}(\text{aB}) + \overrightarrow{p_{\mathcal{X}}}(\text{ab}) \tag{4c}$$

In Example 1, each input symbol maps to exactly one output symbol, so the precover decomposes neatly into cylinders. The next example shows what happens when some source strings cover the target string, while their extensions do not.

---

[7]Note that the precover depends on $f$; we suppress this dependency when it is clear from context.

**Example 2.** *The transducer below implements a* newspeak[8] *rewrite rule: the word* bad *is replaced by* ungood*. For the target string* ba*, the precover does not decompose neatly into cylinders: since* bad *does not map to a string prefixed by* ba*, the cylinder* $\langle bad \rangle$ *does not contribute to* $\overrightarrow{p_{\mathcal{Y}}}(ba)$*. The derivation on the right (5–5c) shows how this cylinder is excluded.*[9]

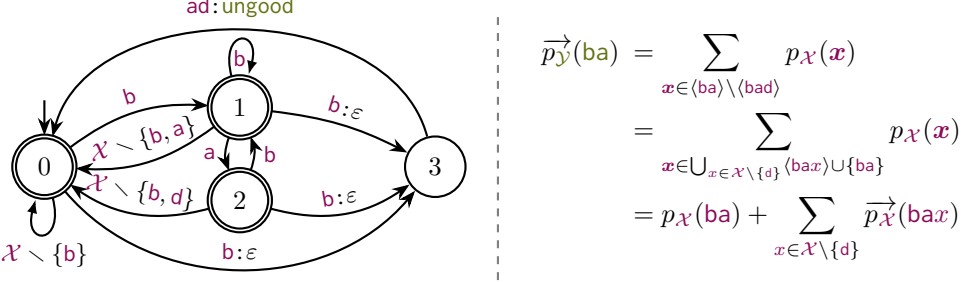

$$\overrightarrow{p_{\mathcal{Y}}}(ba) = \sum_{x \in \langle ba \rangle \setminus \langle bad \rangle} p_{\mathcal{X}}(x) \tag{5a}$$

$$= \sum_{x \in \bigcup_{x \in \mathcal{X} \setminus \{d\}} \langle bax \rangle \cup \{ba\}} p_{\mathcal{X}}(x) \tag{5b}$$

$$= p_{\mathcal{X}}(ba) + \sum_{x \in \mathcal{X} \setminus \{d\}} \overrightarrow{p_{\mathcal{X}}}(bax) \tag{5c}$$

Because bad $\in \langle ba \rangle$ but bad $\notin \mathcal{P}(ba)$, the precover cannot be decomposed entirely into cylinders. Instead, we decompose it into two disjoint parts: a maximal cylindrical subset and its complement in the precover. The *quotient* collects the shortest element of each cylinder; the complement is the *remainder*. The final step (5c) illustrates a **computational shortcut**; for any $y$ we can decompose $\overrightarrow{p_{\mathcal{Y}}}(y)$:

$$\overrightarrow{p_{\mathcal{Y}}}(y) = \underbrace{\sum_{x \in \mathcal{Q}(y)} \overrightarrow{p_{\mathcal{X}}}(x)}_{\text{Quotient}} + \underbrace{\sum_{x \in \mathcal{R}(y)} p_{\mathcal{X}}(x)}_{\text{Remainder}} \tag{6}$$

The precover can also be represented as an FSA, as shown below for $\mathcal{P}(ba)$. The single string accepted at R (teal node) marks the remainder element ba while the state Q (yellow node) accepts $\langle ba \cdot (\mathcal{X} \setminus \{d\}) \rangle$ and marks the cylinder over the quotient.

We now formalize the remainder, quotient, and decomposition for any string-to-string function $f$.

**The prefix decomposition of the precover.** Let $f \colon \mathcal{X}^* \to \mathcal{Y}^*$ be a map. For each $y \in \mathcal{Y}^*$, define $\mathcal{C}(y) \stackrel{\text{def}}{=} \{x \in \mathcal{X}^* \colon \langle x \rangle \subseteq \mathcal{P}(y)\}$; this set is a cylinder—if $x \in \mathcal{C}(y)$ then every extension of $x$ also belongs to $\mathcal{C}(y)$—making it the largest cylinder contained in $\mathcal{P}(y)$. We define the **quotient** and **remainder** of $y$ with respect to $f$ as

$$\mathcal{Q}(y) \stackrel{\text{def}}{=} \text{pf}(\mathcal{C}(y)) \qquad \text{and} \qquad \mathcal{R}(y) \stackrel{\text{def}}{=} \mathcal{P}(y) \setminus \mathcal{C}(y) \tag{7}$$

We call the pair $(\mathcal{Q}(y), \mathcal{R}(y))$ the **optimal prefix decomposition** of $\mathcal{P}(y)$, characterized by three conditions: (i) $\mathcal{Q}(y)$ is *prefix-free*, (ii) $\mathcal{P}(y) = \langle \mathcal{Q}(y) \rangle \sqcup \mathcal{R}(y)$ (*validity*), and (iii) $\langle \mathcal{Q}(y) \rangle = \mathcal{C}(y)$ (*maximality*)—$\mathcal{Q}(y)$ identifies the largest cylinder in $\mathcal{P}(y)$. This decomposition (indeed, any valid one) lets us compute prefix probabilities using the shortcut:

$$\overrightarrow{p_{\mathcal{Y}}}(y) = \sum_{x \in \mathcal{P}(y)} p_{\mathcal{X}}(x) = \sum_{x \in \mathcal{C}(y) \sqcup \mathcal{R}(y)} p_{\mathcal{X}}(x) = \sum_{\substack{x \in \mathcal{Q}(y) \\ x' \in \mathcal{X}^*}} p_{\mathcal{X}}(xx') + \sum_{x \in \mathcal{R}(y)} p_{\mathcal{X}}(x) = \sum_{x \in \mathcal{Q}(y)} \overrightarrow{p_{\mathcal{X}}}(x) + \sum_{x \in \mathcal{R}(y)} p_{\mathcal{X}}(x) \tag{8a}$$

§5 provides an algorithm for computing the prefix decomposition; §6 identifies when it is finite.

## 5 ALGORITHMS

We now present an algorithm for computing prefix decompositions by exploiting the explicit structure of a transducer that encodes the function. Combined with the computational shortcut (Eq. (6)), this gives an autoregressive interface to transduced language models. We first describe the algorithm abstractly, then instantiate its checks using a transducer, and finally discuss optimizations.

---

[8]*Newspeak* is the controlled language from Orwell's *1984* (Orwell, 1949); the example bad $\to$ ungood is canonical. A complete Newspeak transducer is left as an exercise for the Party.

[9]Arcs labeled by a single symbol are *copy transitions*, e.g., b is shorthand for b : b; a set label such as $\mathcal{X} \setminus \{b\}$ denotes one copy-transition arc per symbol in the set.

Fig. 2 gives the decomposition algorithm (decompose), which maintains a queue of candidate source strings and explores them by breadth-first search (BFS), optionally pruning low-probability candidates at each step (described below).[10] Each dequeued string $x$ undergoes three checks, defined in terms of the precover $\mathcal{P}(y)$:

1. *Cylinder*: $\texttt{is\_cylinder}(x, y) \iff \langle x \rangle \subseteq \mathcal{P}(y)$, i.e., every extension of $x$ covers $y$. $x$ is added to the quotient set Q and not explored further (line 12).
2. *Member*: $\texttt{is\_member}(x, y) \iff x \in \mathcal{P}(y)$, i.e., $x$ itself covers $y$. When is_cylinder is false but is_member is true, $x$ is added to the remainder set R; its extensions are still explored (line 15).
3. *Live*: $\texttt{is\_live}(x, y) \iff \exists x'' \in \mathcal{X}^* : xx'' \in \mathcal{P}(y)$, i.e., some extension of $x$ belongs to the precover. Only live extensions are enqueued (line 18).

Because strings are processed shortest-first, if any prefix of the current string had already entered the quotient, the current string would never have been enqueued. Once the queue is exhausted, the algorithm returns the prefix decomposition. We instantiate these checks concretely in §5.1.

```
 1 def decompose(y): # memoize
 2     N ← |y|
 3     if N = 0:
 4         q ← Queue({ε})
 5     else:
 6         (Q′,R′) ← decompose(y_{<N})
 7         q ← Queue(Q′ ∪ R′)
 8     (Q,R) ← (∅,∅)
 9     while |q| > 0:
10         q′ ← ∅
11         for x ∈ q:
12             if is_cylinder(x,y):
13                 Q.add(x)
14                 continue
15             if is_member(x,y):
16                 R.add(x)
17             for x′ ∈ X:
18                 if is_live(xx′,y):
19                     q′.add(xx′)
20         q ← prune(q′)
21     return (Q,R)
```

Figure 2: Decomposition algorithm.

**Theorem 5.1** (Correctness of decompose). *If $\mathcal{P}(y)$ admits a finite decomposition and the three checks exactly implement the conditions above with no pruning, then decompose($y$) terminates and its output (Q, R) is the optimal prefix decomposition (Eq. (7)).*

*Proof.* We verify the three conditions of the optimal prefix decomposition (Eq. (7)).

1. *Prefix-freeness* (Q is prefix-free): When is_cylinder succeeds for $x$ (line 12), the continue (line 14) skips the extension loop, so no $xx'$ is enqueued. Since strings enter the queue only as single-symbol extensions of dequeued strings, no extension of $x$ is ever enqueued or dequeued, and thus no two elements of Q share a prefix.
2. *Validity* ($\mathcal{P}(y) = \langle Q \rangle \sqcup R$): Let $x \in \mathcal{P}(y)$; we first show that $x \in \langle Q \rangle \sqcup R$. We show by induction on prefix length that either a prefix of $x$ enters Q or $x$ itself is dequeued and added to Q or R. The base case holds: $\varepsilon$ is enqueued. If $x_{<k}$ is dequeued and not placed in Q, then $x_{<k+1}$ is enqueued by the extension loop: since $x_{<k+1} \preceq x$ and $x \in \mathcal{P}(y)$, we have $\texttt{is\_live}(x_{<k+1}, y)$ (line 18). By induction, either some prefix $x_{<j}$ enters Q—giving $x \in \langle x_{<j} \rangle \subseteq \langle Q \rangle$—or $x$ itself is dequeued, the cylinder check fails, and $x \in \mathcal{P}(y)$ gives $x \in R$ via is_member (line 15). The reverse direction is clear since elements of Q pass the exact is_cylinder check, so $\langle Q \rangle \subseteq \mathcal{P}(y)$. Similarly, elements of R pass is_member, so $R \subseteq \mathcal{P}(y)$. What remains is to show that $\langle Q \rangle$ and R are disjoint. If $x \in R$, then $x$ was dequeued and is_cylinder failed, so $x \notin Q$. No proper prefix of $x$ is in Q either—otherwise $x$ would never have been enqueued (prefix-freeness). Hence $x \notin \langle Q \rangle$.
3. *Maximality* ($\langle Q \rangle = \mathcal{C}(y)$): shortest-first processing ensures no proper prefix of a quotient element satisfies is_cylinder (line 12), so Q is the *minimal* prefix-free set of cylinders. Together with the exact cylinder check, the earliest qualifying prefix is always found, as required by Eq. (7). It remains to show $\mathcal{C}(y) \subseteq \langle Q \rangle$. By validity, $\mathcal{P}(y) = \langle Q \rangle \sqcup R$. Every element of R fails is_cylinder, so $R \cap \mathcal{C}(y) = \emptyset$. Hence $\mathcal{C}(y) \subseteq \langle Q \rangle$.

*Termination*: is_live (line 18) ensures only viable extensions are enqueued; finiteness of the decomposition guarantees the queue empties. ∎

**Conservative checks and suboptimality.** If the checks used by decompose are conservative approximations, the algorithm may produce *suboptimal* decompositions—valid but with a smaller quotient than the optimal one. A conservative liveness check (false positives) enqueues unnecessary extensions but does not affect classification: every dequeued string is still correctly classified by is_cylinder and is_member, so the result remains optimal. A conservative cylinder check (false negatives) may fail to recognize some quotient elements, placing them in R instead. The decomposition

---

[10]For efficiency, decompose should be memoized. For $|y| > 0$, the recursive call decompose($y_{<N}$) seeds the queue with the previous decomposition $Q' \cup R'$ rather than enumerating from $\varepsilon$.

remains valid, but because a successful cylinder check terminates exploration of that subtree (line 14), missing a cylinder means the BFS continues exploring extensions that would otherwise have been cut off, potentially inflating the remainder and increasing the computation. By contrast, the membership check must be exact to preserve validity.

**Approximation via pruning.** When the prefix decomposition becomes large, exhaustively enumerating and scoring can become infeasible. In these cases, we use a pruning strategy (prune) that sorts candidates by prefix probability and removes those whose cumulative probability mass falls below a specified threshold $\tau$. This discards low-probability candidates to keep decomposition tractable. Since pruning only removes candidates from the queue, every element found is correct—$\langle Q \rangle \sqcup R \subseteq \mathcal{P}(y)$—but the decomposition is no longer valid in general (coverage may be incomplete), so the computed prefix probability is a lower bound on the true value. Our strategy is detailed in §C.3.

## 5.1 GETTING STARTED: INSTANTIATING THE CHECKS WITH THE PRECOVER MACHINE

We now show how to instantiate the three checks (Fig. 2) using a finite-state transducer. This serves as a pedagogical introduction; §C describes a more detailed, but faster algorithm.

We represent the transformation $f$ with a transducer $\mathsf{f}$ (see §2 and §B). Given a target prefix $y$, $\text{proj}_{\mathcal{X}}(\mathsf{f} \circ y\mathcal{Y}^*)$ is an NFA that accepts exactly $\mathcal{P}(y)$. To enable the efficient state-based checks below, we determinize and trim this NFA to obtain a DFA: $\mathsf{P}_y \stackrel{\text{def}}{=} \text{trim}(\text{determinize}(\text{proj}_{\mathcal{X}}(\mathsf{f} \circ y\mathcal{Y}^*)))$. Let $\mathsf{S}_y$, $\mathsf{I}_y$, $\mathsf{F}_y$, and $\mathsf{T}_y$ denote the components of $\mathsf{P}_y$. Since $\mathsf{P}_y$ is deterministic, scanning a source string $x = x_1 \cdots x_N$ yields a unique state, which we denote by $\text{run}_y(x)$. We now describe how to implement the three checks using $\mathsf{P}_y$ (pseudocode in Fig. 3).

- *Cylinder*: We need to check whether $\langle x \rangle \subseteq \mathcal{P}(y)$ to decide if $x \in \mathcal{Q}(y)$. Let $S = \text{run}_y(x)$ be the unique state reached after scanning $x$. Since $\mathsf{P}_y$ is deterministic, $\langle x \rangle \subseteq \mathcal{P}(y)$ if and only if $S$ is **universal**: $[\![\mathsf{P}_{y[S]}]\!] = \mathcal{X}^*$, meaning every continuation of $x$ is accepted. The is_cylinder check (Fig. 3) tests universality via breadth-first search (BFS) from $S$.
- *Member*: When the is_cylinder check fails for $x$ with $S = \text{run}_y(x)$, we need to determine whether the scanned string $x$ is in $\mathcal{R}(y)$. Here it suffices to check if $S \in \mathsf{F}_y$, i.e., if $\mathsf{P}_y$ accepts $x$.
- *Live*: We construct $\mathsf{P}_y$ as a *trimmed* automaton accepting $x$ where $y \preceq f(x)$.[11] Trimming ensures that every reachable state lies on some accepting path, so liveness reduces to $\text{run}_y(x) \neq \emptyset$.

Although decompose is written in terms of source strings, each check internally computes the state $S = \text{run}_y(x)$ reached by scanning $x$ in the deterministic $\mathsf{P}_y$, and reduces to a state property.

## 5.2 OPTIMIZATIONS

The algorithm above is correct but impractical for large transducers. In §C, we describe several optimizations; we summarize the key ideas here.

**Lazy determinization.** Eagerly determinizing $\mathsf{P}_y$ is often computationally expensive. Instead, we track *frontiers*: sets of transducer states reachable after scanning a source prefix, paired with the output emitted so far. Frontiers lazily perform the subset construction (the standard NFA-to-DFA conversion; Rabin & Scott, 1959) and the composition with $y\mathcal{Y}^*$ simultaneously, avoiding both eager determinization and eager composition. The three checks—cylinder, member, and liveness—are now defined in terms of the frontier rather than a single DFA state (§C.2).

**Incremental next-symbol decomposition.** To efficiently compute $\overrightarrow{p_y}(y' \mid y)$ for all $y' \in \mathcal{Y}$, we introduce decompose_next (§C.4), which derives the decomposition of each extension $yy'$ from the decomposition of $y$. It operates in a single BFS pass that handles each $y' \in \mathcal{Y}$ simultaneously.

**Decomposition shortcuts.** Many structural properties of the decomposition allow the BFS to skip work: non-cylinder monotonicity (Prop. C.3), cylinder uniqueness (Prop. C.4), input-projection universality (§C.6), combined universality (Prop. C.5), and an all-universal fast path (Fig. 14).

**Lazier enumeration.** Rather than explicitly composing the transducer with the copy transducer $y\mathcal{Y}^*$, the frontier-based algorithms track output buffers directly, performing the composition lazily and avoiding the materialization cost. Additionally, we precompute states whose input projection

---

[11]This can also be expressed as $\mathcal{P}(y) = f^{-1}(y\mathcal{Y}^*) = [\![\text{proj}_{\mathcal{X}}(\mathsf{f} \circ y\mathcal{Y}^*)]\!]$.

```
22 def step_y(s, x):
23    return T_y(s, x) # deterministic

24 def run_y(x_1 ⋯ x_N):
25    if N = 0: return I_y # unique
26    return step_y(run_y(x_<N), x_N)

27 def is_member(x, y):
28    return run_y(x) ∈ F_y

29 def is_live(x, y):
30    # not in failure state
31    return run_y(x) ≠ ∅
```

```
32 def is_cylinder(x, y):
33    # Run BFS to find a counterexample:
34    # a nonaccepting or incomplete state
35    S ← run_y(x); V ← {S}; q ← QUEUE({S})
36    while q:
37       S ← q.pop()
38       if S ∉ F_y: return False
39       for x' ∈ X:
40          S' ← step_y(S, x')
41          if S' = ∅: return False
42          if S' ∉ V: q.add(S'); V.add(S')
43    return True   # No counterexamples
```

Figure 3: State-based instantiation of the checks from Fig. 2 on the precover DFA $\mathsf{P}_y$. Helper functions `step` and `run` advance through the DFA; `is_member` and `is_live` are single-state lookups; `is_cylinder` performs a BFS to verify universality.

accepts all of $\mathcal{X}^*$ (§C.6). When a frontier reaches such a state with a buffer that already covers $y$, the cylinder check succeeds immediately, bypassing the expensive check.

**Label pushing.** As a standard preprocessing step (see, e.g., Oncina et al., 1993; Mohri, 2003; 2009), we push output labels toward the initial state of $f$: when every path through a state $s$ emits output beginning with $y$, that $y$ is shifted to the incoming arc so it is produced one step earlier. This ensures that output is committed as early as possible, speeding up the frontier-based cylinder check.

## 6 SUFFICIENT CONDITIONS FOR FINITE DECOMPOSITIONS

The decomposition algorithms in §5 enumerate strings by breadth-first search. For the enumeration to terminate, the quotient $\mathcal{Q}(y)$ and remainder $\mathcal{R}(y)$ need to be finite for a given target string $y \in \mathcal{Y}^*$. Previous work (Vieira et al., 2025a, Props 1 and 3; also in §D.2) showed that strict-prefix monotonicity guarantees an empty remainder and a bounded quotient. This ensures that $\overrightarrow{p_y}(y)$ is computable.

For functions that are not strict-prefix monotone, however, the decomposition may be infinite. In this section, we give additional conditions for finite decomposition. We first give function-level properties that guarantee an empty remainder and bounded quotient, and then transducer-level conditions for when the quotient and remainder are finite.

We say that a map $f: \mathcal{X}^* \to \mathcal{Y}^*$ is **strict-prefix monotone** if and only if $x \prec x' \implies f(x) \prec f(x')$. Similarly $f$ is **prefix monotone** if and only if $x \preceq x' \implies f(x) \preceq f(x')$. We say that a map $f$ is **prefix-continuous** if, for every $y \in \mathcal{Y}^*$, the set $\mathcal{P}(y)$ is cylindrical, i.e., $\mathcal{R}(y) = \emptyset$. The following proposition shows that prefix monotonicity implies prefix-continuity:

**Proposition 6.1.** *The following are equivalent: (i) $f$ is prefix monotone (ii) $f(\langle x \rangle) \subseteq \langle f(x) \rangle$ for all $x \in \mathcal{X}^*$ (iii) $\mathcal{P}(f(x)) = \mathcal{C}(f(x))$ for all $x \in \mathcal{X}^*$ (iv) $f$ is prefix-continuous.* [Proof: §D.1]

Proposition 6.1 shows how we generalize Vieira et al. (2025a). First, by relaxing strict-prefix monotonicity to prefix monotonicity, we support multi-symbol lookahead in the quotient. Ex. (6) is an example of this: it is prefix monotone and requires two-symbol lookahead before committing to an output. Second, by introducing the remainder, we can handle functions that are not prefix monotone, meaning that there can be source strings that cover the target, but not all of their extensions do (e.g., Example 2).

Lemma 6.1 gives sufficient conditions on a transducer that guarantee a finite decomposition for every target string, even when the underlying function is not prefix monotone. The key notion is *safety*: a state is safe if it is IP-universal, has **finite closure** (i.e., $|[\![f_{[s]}]\!]| < \infty$), or all its successors are safe.

**Lemma 6.1.** *Let $f: \mathcal{X}^* \to \mathcal{Y}^*$ be a function realized by a transducer $f$. The decomposition $(\mathcal{Q}(y), \mathcal{R}(y))$ is finite for every $y \in \mathcal{Y}^*$ if:*

*(i)* No $\varepsilon$-output cycles: $f$ contains no cycle in which every arc outputs $\varepsilon$.
*(ii)* Safety: *Every state of $f$ is **safe**, defined inductively as the smallest set such that $s$ is safe if: (a) $s$ is IP-universal; (b) $|[\![f_{[s]}]\!]| < \infty$ (finite closure); or (c) for all transitions $s \xrightarrow{x:y} s'$, $s'$ is safe.*

Proof: See §D.3.

The conditions in Lemma 6.1 guarantee exact computation. In particular, these are satisfied by the transducers introduced in the experiments section (§7): the token-to-byte transducer $f_\alpha$ and the DNA-to-amino-acid transducer $f_{dna2aa}$, whose quotients are finite and remainders empty, but not by the PTB transducer $f_{ptb}$, whose quotients are infinite.

The finiteness of decomposition is a property of the function $f$, not of any particular transducer encoding it. The lemma's conditions are sufficient but not necessary: safety tests each state individually, whereas the decomposition algorithm tests universality of frontiers (sets of state/string pairs), which can succeed even when individual states are not safe. There are many cases where the prefix decomposition is infinite, or otherwise prohibitively large; in these cases, we let the source language model determine which prefix decomposition members carry the most significant mass and prune others (§5). This lets us greedily approximate the probabilities of the transduced language model by enumerating a high-mass subset of the decomposition.

# 7    EXPERIMENTS

We now consider three examples of transduced language models. For each use case, we follow the approach in Vieira et al. (2025a) and measure the Jensen–Shannon divergence (JSD) between the distributions obtained using the approximation via probability mass pruning mentioned in §C.3 and a reference distribution we get by choosing a pruning threshold $\tau$. We also report cross-entropy loss (§G.4), reflecting the cost of scoring specific sequences rather than full distributions. Experiments use GPT-2 Large ($p_{gpt2}$) (Radford et al., 2019), LLaMA 3.2-1B ($p_{llama1B}$) and LLaMA 3.1-8B ($p_{llama8B}$) (Team, 2024), Phi-4 ($p_{phi4}$) (14B; Abdin et al., 2024), and a DNA model trained on the human genome ($p_{dna}$).[12] For the experiments in §7, we use genlm-bytes[13] to convert token-level models to byte-level models. See §F for details on the training and evaluation setup, and §F.5 for transducer details.

Recall that a transduced language model $p_{\mathcal{Y}}$ is characterized by a source model $p_{\mathcal{X}}$ and a transducer $f$ encoding some function $f$ (§3). To make this dependency explicit, we write $p_{\mathcal{X}} \circ f \stackrel{\text{def}}{=} p_{\mathcal{Y}}$ and refer to the operation as composing. All experiments compute next-symbol distributions using the more efficient algorithm (§C.8), which reuses cached decompositions across target positions and exploits structural shortcuts (§C.5). Probability mass pruning (§C.3) retains the most likely decomposition elements such that the discarded probability mass does not exceed the pruning threshold.

Our experiments span the prefix-monotonicity spectrum (Proposition 6.1): token-to-byte (T2B) is strict-prefix monotone, DNA-to-amino-acid is prefix monotone with multi-symbol lookahead, and PTB tokenization is not prefix monotone.

**From tokens to bytes.**    We revisit the algorithm for converting models from tokens to bytes, as in Vieira et al. (2025a). This transformation can be realized by a simple transducer $f_\alpha$, with $\mathcal{O}(|\mathcal{X}|)$ states. Specifically, the transducer contains a chain for each token present in the input vocabulary $\mathcal{X}$. This structure allows us to use a shortcut (see Fig. 14) that resolves each quotient element with a single LM call. An example of such a machine is given in Fig. 4. We benchmark the algorithms in §5 using the transducers $p_{gpt2} \circ f_\alpha$, $p_{llama1B} \circ f_\alpha$, $p_{llama8B} \circ f_\alpha$, and $p_{phi4} \circ f_\alpha$ on the first ten paragraphs of the `wikitext-2-raw-v1` dataset (Merity et al., 2017) (corresponding to the first 7684 bytes). As shown in Fig. 6 (left) and Tab. 8 in §G, lower pruning thresholds ($\tau$) give lower JSD values against a reference distribution ($\tau = $ 1e-5) at the cost of throughput, measured in bytes per second. We confirm that the JSD values are similar to those achieved by Vieira et al. (2025a) in §G.1, Tab. 6. Their method is limited to strict-prefix monotone transformations, which enables a specialized trie-based algorithm that achieves higher throughput while maintaining comparable accuracy.

**From tokens to orthographic word boundaries.**    The next transformation we consider is converting language models over tokens to language models over *orthographic* words. The precise definition of what constitutes such a word varies depending on the application. In some settings, contractions such as "`wouldn't`" should be treated as a single unit. In other settings, however, a more natural segmentation would be "`would`" and "`n't`". We can accommodate any FST-based definition. Linguistic tokenizers, such as the PTB tokenizer (Marcus et al., 1993), use contextual information to

---

[12]Links to models: `https://huggingface.co/openai-community/gpt2-large`, `https://huggingface.co/meta-llama/Llama-3.2-1B`, `https://huggingface.co/meta-llama/Llama-3.1-8B`, `https://huggingface.co/microsoft/phi-4` and `https://huggingface.co/vesteinn/gpt2-dna`.

[13]genlm-bytes is an implementation of Vieira et al. (2025a); see `https://github.com/genlm/genlm-bytes`.

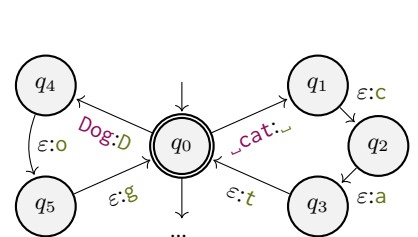
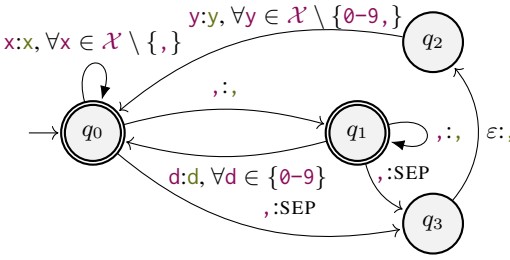

Figure 4: An FST for converting a token model into a character model. Paths for ␣cat and Dog.

Figure 5: An FST that inserts a separator (SEP) before commas followed by non-digit characters.

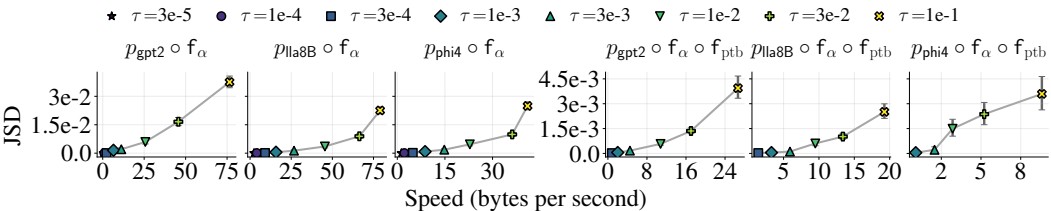

Figure 6: Average Jensen–Shannon divergence (JSD) and throughput (bytes/sec) across thresholds $\tau$. Left: $p_{\mathcal{X}} \circ f_\alpha$ (reference: $\tau = $ 1e-5). Right: $p_{\mathcal{X}} \circ f_\alpha \circ f_{\text{ptb}}$ (reference: $\tau = $ 1e-4; 3e-4 for $p_{\text{phi4}}$). Bars show 95% bootstrapped CIs. At the tightest thresholds, large decomposition sizes prevent some models from completing all paragraphs; see §G.2.

segment raw English text into linguistically meaningful units suitable for downstream NLP tasks. We construct an FST that encodes the PTB tokenizer, $f_{\text{ptb}}$; details are in §F.5. An example of a rule in the transducer is given in Fig. 5, which inserts SEP before a comma if a digit does not follow. Composing $p_{\mathcal{X}} \circ f_\alpha \circ f_{\text{ptb}}$ yields a transduced language model, mapping subword tokens to PTB tokens. To reduce the state count of the composed FST and improve efficiency, we represent $p_{\mathcal{X}} \circ f_\alpha$ using the byte-transformed models from genlm-bytes.[14] In Fig. 6 (right), we plot the average JSD against the throughput for different thresholds $\tau$, using the same dataset as in §7. The reference distribution uses $\tau = $ 1e-4 (3e-4 for $p_{\text{phi4}}$). As with the experiments in §7, we observe lower JSD at lower thresholds, albeit at the cost of throughput. For experiments using higher thresholds, see §G, Tab. 9.

**Converting DNA models to models over amino acids.** We use a transducer that converts sequences of the four DNA nucleotides to sequences over the twenty-two amino acids.[15] Let $p_{\text{dna}} \circ f_{\text{dna2aa}}$ denote the transduced model that converts DNA nucleotides into amino acids. To evaluate our approach, we sample 65 human proteins.[16] In Tab. 10, we show the average JSD and throughput for different thresholds $\tau$. Note that this transducer is particularly challenging since the set of candidates in the decomposed precover grows exponentially with the sequence length. To mitigate the combinatorial blow-up, we cap the candidate-set and report throughput and JSD while varying the cap.

## 8 CONCLUSION

We have introduced a general framework for transforming language models using transducers. Empirically, we have shown that our beam-summing approximation efficiently transduces token-based LLMs into models over bytes, words, and even amino acids, without requiring retraining. Our theoretical analysis characterizes the conditions under which such mappings can be performed exactly. The proposed approach is an effective way to repurpose existing language models and to accurately compute probabilities for any unit and transformation defined by a transducer. By reducing the problem to transducer decomposition, the framework opens the door to leveraging future advances in finite-state methods for language model adaptation. (Future work and limitations are discussed in §J.)

---

[14] https://github.com/genlm/genlm-bytes that implements Vieira et al.'s (2025a) method.

[15] I.e., $\mathcal{X} = \{A, C, G, T\}$ and $\mathcal{Y} = \{A, R, N, D, C, Q, E, G, H, I, L, K, M, F, P, S, T, V, W, Y, B, Z, *\}$.

[16] Sampled from https://www.uniprot.org/uniprotkb?query=Human, see Tab. 1 for the accession numbers.

## ACKNOWLEDGMENTS

The authors thank Ben LeBrun, Brian DuSell, Clemente Pasti, Juan Luis Gastaldi, Mario Giulianelli, and Yahya Emara for useful feedback and discussions that greatly improved this work. Vésteinn Snæbjarnarson is supported by the Pioneer Centre for AI, DNRF grant number P1.

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

APPENDIX CONTENTS

## A   NOTATION GLOSSARY

| Notation | Gloss |
|---|---|
| $\varepsilon$ | Empty string |
| EOS | End-of-string symbol |
| $x, x' \in \mathcal{X}$ | Symbols in the source alphabet $\mathcal{X}$ |
| $\boldsymbol{x}, \boldsymbol{x}' \in \mathcal{X}^*$ | Source strings |
| $\mathcal{X}^*$ | Set of all source strings |
| $Z, Z' \subseteq \mathcal{X}^*$ | Sets of source strings |
| $\boldsymbol{x}\boldsymbol{x}'$ | Concatenation (strings) |
| $ZZ'$ | Concatenation (sets of strings) |
| $y, y' \in \mathcal{Y}$ | Symbols in the target alphabet |
| $\boldsymbol{y}, \boldsymbol{y}' \in \mathcal{Y}^*$ | Target strings |
| $\mathcal{Y}^*$ | Set of all target strings |
| $f \colon \mathcal{X}^* \to \mathcal{Y}^*$ | Transformation from source strings $\mathcal{X}^*$ to target strings $\mathcal{Y}^*$ |
| $\boldsymbol{x} \preceq \boldsymbol{x}'$ | $\boldsymbol{x}$ is a prefix of $\boldsymbol{x}'$ |
| $\boldsymbol{x} \prec \boldsymbol{x}'$ | $\boldsymbol{x}$ is a strict-prefix of $\boldsymbol{x}'$ |
| $\langle Z \rangle$ | Cylinder set spanned by $Z$ |
| $\mathrm{pf}(Z)$ | Prefix-base of $Z$ |
| $p_{\mathcal{X}}$ | Language model over source strings $\mathcal{X}^*$ |
| $X$ | $\mathcal{X}^*$-valued random variable $X \sim p_{\mathcal{X}}$ |
| $\overrightarrow{p_{\mathcal{X}}}$ | Prefix probability of $p_{\mathcal{X}}$ |
| $p_{\mathcal{Y}}$ | Language model over target strings $\mathcal{Y}^*$ |
| $f(X)$ | $\mathcal{Y}^*$-valued random variable $f(X) \sim p_{\mathcal{Y}}$ |
| $\overrightarrow{p_{\mathcal{Y}}}$ | Prefix probability of $p_{\mathcal{Y}}$ |
| $\mathtt{f}$ | Transducer implementation of $f$ |
| $\hookrightarrow \mathcal{X}$ | Input alphabet |
| $\hookrightarrow \mathcal{Y}$ | Output alphabet |
| $\hookrightarrow \mathsf{S}$ | Set of states |
| $\hookrightarrow \mathsf{I} \subseteq \mathsf{S}$ | Set of initial states |
| $\hookrightarrow \mathsf{F} \subseteq \mathsf{S}$ | Set of accepting states |
| $\hookrightarrow \mathsf{T}$ | Set of transitions |
| $\hookrightarrow \mathsf{U} \subseteq \mathsf{S}$ | Set of IP-universal states (§C.6) |
| $s, s' \in \mathsf{S}$ | States |
| $(s \xrightarrow{x:y} s') \in \mathsf{T}$ | Transition from $s$ to $s'$ that scans $x$ and emits $y$ |
| $\mathsf{T}(s) \subseteq \mathsf{T}$ | Outgoing transitions from state $s$ |
| $\mathtt{f} \circ \mathtt{f}'$ | Transducer composition |
| $f \circ f'$ | Relation composition |
| $\mathrm{proj}_{\mathcal{X}}(\mathtt{f})$ | Input projection |
| $\mathtt{f}_{[s]}$ | Force-start $\mathtt{f}$ in state $s$ |
| $\boldsymbol{y}\mathcal{Y}^*$ | Either a copy-transducer that accepts $\boldsymbol{y}\mathcal{Y}^*$ or the corresponding relation. |
| $\mathtt{f} \circ \boldsymbol{y}\mathcal{Y}^*$ | A transducer whose paths are restricted to those that accept $\boldsymbol{y}\mathcal{Y}^*$. |
| $f^{-1}(\boldsymbol{y})$ | Preimage of the target string $\boldsymbol{y}$ (§3) |
| $f^{-1}(\mathcal{Y})$ | The preimage of a set $\mathcal{Y}$, $\{f^{-1}(\boldsymbol{y}) \mid \boldsymbol{y} \in \mathcal{Y}\}$ (§3) |
| $\mathcal{P}(\boldsymbol{y})$ | Precover of the target string $\boldsymbol{y}$ (§3) |
| $\mathcal{C}(\boldsymbol{y})$ | The largest cylinder set contained in $\mathcal{P}(\boldsymbol{y})$ (§4) |
| $\mathcal{Q}(\boldsymbol{y})$ | Quotient (§4) |
| $\mathcal{R}(\boldsymbol{y})$ | Remainder (§4) |

**Notation conventions.**   **Color** encodes domain: magenta denotes source-domain objects (e.g., $\boldsymbol{x}$, $\mathcal{X}$, $\mathcal{P}$, $\mathcal{Q}$, $\mathcal{R}$) and olive green denotes target-domain objects (e.g., $\boldsymbol{y}$, $\mathcal{Y}$, $\mathcal{Y}$, $f$). **Font** encodes type: lowercase italic for symbols ($x$, $y$), bold italic for strings ($\boldsymbol{x}$, $\boldsymbol{y}$), calligraphic for alphabets and sets ($\mathcal{X}$, $\mathcal{Y}$, $\mathcal{P}$, $\mathcal{Q}$, $\mathcal{R}$), sans-serif for transducer components ($\mathsf{S}$, $\mathsf{T}$, $\mathsf{I}$, $\mathsf{F}$, $\mathsf{U}$), and typewriter for the transducer itself ($\mathtt{f}$) and for concrete strings in examples ($\mathtt{abc}$, $\mathtt{xyz}$). The overrightarrow $\overrightarrow{\cdot}$ marks prefix probabilities: $\overrightarrow{p_{\mathcal{X}}}$ is the prefix probability of $p_{\mathcal{X}}$.

## B    BACKGROUND ON TRANSDUCERS

This section introduces additional information on transducers, complementing §2.

**Transducer variants.**    We say a transducer is **functional** if it defines a function, and **partially functional** if it defines a partial function. A transducer is **input-deterministic** if for every state $s \in \mathsf{S}$, $|\mathsf{T}(s, \varepsilon)| = 0$ and $|\mathsf{T}(s, x)| \leq 1$ for all $x \in \mathcal{X}$[17]. For input-deterministic transducers, each source string $\boldsymbol{x} \in \mathcal{X}^*$ has at most one accepting path that scans $\boldsymbol{x}$ and, therefore, can emit at most one target string $\boldsymbol{y} \in \mathcal{Y}^*$. Thus, every input-deterministic transducer defines a (partial) function. A **copy-transducer** is one where every transition emits the same symbol as it scans, i.e., every transition is of the form $s \xrightarrow{x:x} s'$. We abbreviate copy transitions as $s \xrightarrow{x} s'$.[18] Copy-transducers define partial identity functions: they map each string in a designated subset to itself, and drop all others. In the same way that we abbreviate copy transitions, we may implicitly map them to a set of strings via $(\boldsymbol{x}, \boldsymbol{x}) \mapsto \boldsymbol{x}$.

**Operations.**    Transducers support **composition**: given transducers $\mathsf{f}$ and $\mathsf{f}'$, their composition $\mathsf{f} \circ \mathsf{f}'$ is a transducer denoting $[\![\mathsf{f} \circ \mathsf{f}']\!] \stackrel{\text{def}}{=} [\![\mathsf{f}]\!] \circ [\![\mathsf{f}']\!]$.[19][20] We denote a transducer that encodes the relationship $[\![\mathsf{f}]\!] \circ \{(\boldsymbol{y}', \boldsymbol{y}') \mid \boldsymbol{y}' \in \langle \boldsymbol{y} \rangle\}$ with $\mathsf{f} \circ \boldsymbol{y} \mathcal{Y}^*$. We freely coerce between these representations when the intent is clear from context. We define the **input projection** operation encoding the relationship $[\![\mathsf{proj}_{\mathcal{X}}(\mathsf{f})]\!] = \{(\boldsymbol{x}, \boldsymbol{x}) \mid \exists \boldsymbol{y} \colon (\boldsymbol{x}, \boldsymbol{y}) \in [\![\mathsf{f}]\!]\}$ as $\mathsf{proj}_{\mathcal{X}}(\mathsf{f}) \stackrel{\text{def}}{=} (\mathsf{S}, \mathcal{X}, \mathcal{X}, \mathsf{I}, \mathsf{F}, \{(s \xrightarrow{x:x} s') \mid s \xrightarrow{x:y} s' \in \mathsf{T}\})$, which is a copy-transducer. Let $\mathsf{f}_{[s]}$ denote the operation of **force-starting** $\mathsf{f}$ in state $s$, $\mathsf{f}_{[s]} \stackrel{\text{def}}{=} (\mathsf{S}, \mathcal{X}, \mathcal{Y}, \{s\}, \mathsf{F}, \mathsf{T})$; this operation yields a machine defining the set of source–target suffix pairs that are generated by paths starting at a given $s$ and ending in an accepting state. We say that a state $s$ is **IP-universal** (**input-projection universal**) if $[\![\mathsf{proj}_{\mathcal{X}}(\mathsf{f}_{[s]})]\!] = \mathcal{X}^*$, i.e., no matter what input follows, the transducer can still produce output. Let $\mathsf{U} \subseteq \mathsf{S}$ denote the set of IP-universal states; this set can be precomputed for each transducer (§C.6).

Our algorithms use an **input-determinization** transformation $\mathsf{determinize}(\mathsf{f})$ that maps a (partially) functional transducer $\mathsf{f}$ to an equivalent one that is input-deterministic. In general, such a mapping is not always realizable with a finite number of states.[21] However, in the special case of copy-transducers, input-determinization is always possible, but may result in exponential blowup in the worst case.[22] We use $\mathsf{trim}(\mathsf{f})$ to denote a **trimming** operation that removes all states and edges that do not appear on any accepting path. These are standard operations that are implemented in any FST library; more details can be found in (Pin, 2021; 2025).

**Visual notation.**    We use diagrams like those shown in §E to represent transducers. Transitions without source states denote initial states; double-lined states indicate accepting states. Transitions $s \xrightarrow{x:y} s'$ are shown as arrows between states.

**Limitations.**    Finite-state transducers define the class of *rational relations* (Berstel, 1979, Ch. III). Because FSTs only have finitely many states, they are inherently limited in the relations they can represent. For example, FSTs cannot perform transformations that require unbounded matching or counting. In contrast, transducers with unbounded memory extend beyond the rational class, offering greater expressive power, but come with increased complexity and often undecidability of key properties, such as *universality*.

---

[17]The more general notion of a *subsequential transducer* additionally includes a final output function; this distinction does not arise for copy-transducers.

[18]For readers familiar with finite-state automata, every copy-transducer is isomorphic to an *acceptor*.

[19]Here, **relation composition** is given by $f \circ g \stackrel{\text{def}}{=} \{(x, z) \mid (x, y) \in f, (y, z) \in g\}$ where $f$ and $g$ are relations.

[20]Pin (2025, Ch. XIX, sec. 2) gives an efficient method for constructing $\mathsf{f} \circ \mathsf{f}'$.

[21]Choffrut (1977) gives such an algorithm, by using a power set construction on the input side as in the determinization of nondeterministic finite automata. He shows it yields a finite machine if and only if the automaton has *bounded variation*, the constraint that for any two strings whose prefix distance (the combined length of the strings with the longest shared prefix removed) is bounded, then the output prefix distance is also bounded. He also provides a testable condition for determinization, known as the twinning condition: once two runs have read the same input prefix (i.e., we cut the input at the same point), then for every common continuation, they append the same further output.

[22]For readers familiar with finite-state automata, input-determinization of a copy-transducer is isomorphic to the determinization of an equivalent finite-state automaton, see §B.

## C    EFFICIENT ALGORITHMS

The decomposition algorithm of §5 operates on a determinized, trimmed precover machine $\mathsf{P} = \mathrm{trim}(\mathrm{determinize}(\mathrm{proj}_{\mathcal{X}}(\mathsf{f} \circ \boldsymbol{y}\mathcal{Y}^*)))$. In practice, both the composition with $\boldsymbol{y}\mathcal{Y}^*$ and the explicit determinization are expensive and must be repeated for each target $\boldsymbol{y}$. The algorithms in this section perform both steps lazily, operating directly on the original transducer $\mathsf{f}$ and avoiding explicit materialization. This requires careful bookkeeping of partial outputs via *frontiers* (§C.1.1), but enables precomputation of IP-universal states and incremental reuse across targets.

We begin by stating the algorithmic goal (§C.1), then introduce the frontier data structure (§C.1.1) and show how each check reduces to a frontier query (§C.2). We write $(\mathsf{S}, \mathcal{X}, \mathcal{Y}, \mathsf{I}, \mathsf{F}, \mathsf{T}) = \mathsf{f}$ throughout and omit the dependency on $\overrightarrow{p_{\mathcal{X}}}$ since it is clear from context.

### C.1    THE GOAL: AN EFFICIENT IMPLEMENTATION OF THE AUTOREGRESSIVE INTERFACE

We seek efficient algorithms for the prefix probabilities $\overrightarrow{p_{\mathcal{Y}}}(\boldsymbol{y})$, string probabilities $p_{\mathcal{Y}}(\boldsymbol{y})$, and next-token distributions $\overrightarrow{p_{\mathcal{Y}}}(\cdot \mid \boldsymbol{y})$ of transduced language models. Specifically, we want efficient implementations of the following methods, which constitute the interface to the transduced language model (§2).

```
44 def prefix_prob(y):
45     (Q, R) ← decompose(y)
46     return ∑_{x∈Q} p⃗_X(x) + ∑_{x∈R} p_X(x)
47 def prob(y):
48     return prefix_prob(y) · next_dist(y)[EOS]
```

```
49 def next_dist(y):
50     p̄ ← {}
51     Z ← prefix_prob(y)
52     for y′ ∈ Y:
53         p̄[y′] ← prefix_prob(yy′)/Z
54     p̄[EOS] ← 1 − ∑_{y′∈Y} p̄[y′]
55     return p̄
```

The primitive operation above is `prefix_prob`, which requires a single call to `decompose`. Both `next_dist` and `prob` are derived from it: `next_dist` computes $|\mathcal{Y}|$ additional prefix probabilities and obtains $\overrightarrow{p_{\mathcal{Y}}}(\text{EOS} \mid \boldsymbol{y})$ by complement; `prob` is a one-line product. In an implementation, $\overrightarrow{p_{\mathcal{X}}}$ and $p_{\mathcal{X}}$ should be memoized so that extending $\overrightarrow{p_{\mathcal{X}}}(\boldsymbol{x})$ to $\overrightarrow{p_{\mathcal{X}}}(\boldsymbol{xx})$ requires only a single conditional evaluation rather than replaying the entire history (see §C.7). In §C.4, we show how to compute `next_dist` with a more efficient, joint decomposition, rather than $|\mathcal{Y}|+1$ separate calls to `prefix_prob`.

The three checks in Fig. 2—`is_cylinder`, `is_member`, and `is_live`—all require determining which transducer states are reachable after reading a source string $\boldsymbol{x}$, and what output each has produced relative to $\boldsymbol{y}$. We capture this information in a single data structure: the *frontier* (§C.1.1). In §C.2, we show how each check reduces to a simple query on the frontier. In practice, the quotient and remainder may be large or infinite, so we introduce pruning (§C.3) to obtain practical approximations.

### C.1.1    FRONTIER COMPUTATION

Since $\mathsf{f}$ can be nondeterministic even if it is functional, the transducer $\mathsf{f}$ may reach many states simultaneously for a given source string $\boldsymbol{x}$ and target $\boldsymbol{y}$. Each of these paths may have produced a different output buffer. The **frontier** $\mathcal{F} = \mathrm{run}_{\boldsymbol{y}}(\boldsymbol{x})$ collects this information: it is the set of $(s, \boldsymbol{b})$ pairs—where $s \in \mathsf{S}$ is a transducer state and $\boldsymbol{b} \in \mathcal{Y}^*$ is the output produced so far—reachable by reading $\boldsymbol{x}$ from the initial states, filtered to buffers $\boldsymbol{b}$ compatible with $\boldsymbol{y}$.[23] The frontier encodes all information needed for the three checks: `is_cylinder`, `is_member`, and `is_live` can each be evaluated from $\mathcal{F}$ alone, without retaining the full path history (§C.2; Fig. 8). The pseudocode in Fig. 7 describes how to compute the frontier using the three functions $\mathrm{run}_{\boldsymbol{y}}$, $\mathrm{step}_{\boldsymbol{y}}$, and $\mathrm{closure}_{\boldsymbol{y}}$.

The frontier replaces the explicit precover machine $\mathsf{P} = \mathrm{trim}(\mathrm{determinize}(\mathrm{proj}_{\mathcal{X}}(\mathsf{f} \circ \boldsymbol{y}\mathcal{Y}^*)))$ used in §5. Rather than materializing $\mathsf{P}$, the frontier tracks the same information directly on $\mathsf{f}$, avoiding both the composition with $\boldsymbol{y}\mathcal{Y}^*$ and the explicit determinization. The design reflects a two-phase structure: before a path's buffer covers the target, the frontier tracks per-state output buffers $(s, \boldsymbol{b})$ filtered by target compatibility; once the buffer reaches or passes $\boldsymbol{y}$, the frontier can be truncated for universality checking (§C.2). The frontier unifies the two phases into a single lazy data structure.

---

[23] A buffer $\boldsymbol{b}$ is compatible with $\boldsymbol{y}$ if and only if $\boldsymbol{b} \preceq \boldsymbol{y} \vee \boldsymbol{y} \preceq \boldsymbol{b}$, i.e., the buffer has either covered the target or has yet to diverge from it.

```
56  def run_y(x_1 ⋯ x_N):    # memoize          67  def closure_y(F):
57    if N = 0: return closure_y(I × {ε})      68    F' ← F
58    return step_y(run_y(x_<N), x_N)          69    q ← QUEUE(F)
                                               70    while |q| > 0:
59  def step_y(F, x):                          71      (s, b) ← q.pop()
60    F' ← ∅                                              ε:y'
61    for (s, b) in F:                         72      for (_ ——→ s') ∈ T(s, ε):
            x:y                                 73        b' ← by'
62      for (_ ——→ s') ∈ T(s, x):              74        if b' ⪯ y ∨ y ⪯ b':
63        b' ← by                              75          if (s', b') ∉ F':
64        if b' ⪯ y ∨ y ⪯ b':                  76            F'.add((s', b'))
65          F'.add((s', b'))                   77            q.add((s', b'))
66    return closure_y(F')                     78    return F'
```

Figure 7: Frontier-based state machine

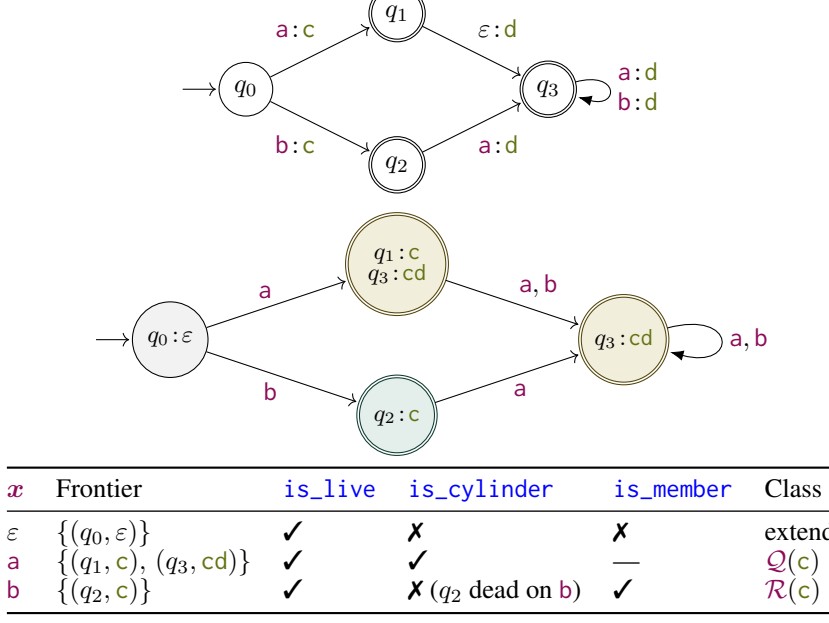

| $x$ | Frontier | is_live | is_cylinder | is_member | Class |
|---|---|---|---|---|---|
| $\varepsilon$ | $\{(q_0, \varepsilon)\}$ | ✓ | ✗ | ✗ | extend |
| a | $\{(q_1, \text{c}), (q_3, \text{cd})\}$ | ✓ | ✓ | — | $\mathcal{Q}(\text{c})$ |
| b | $\{(q_2, \text{c})\}$ | ✓ | ✗ ($q_2$ dead on b) | ✓ | $\mathcal{R}(\text{c})$ |

Figure 8: Truncated frontier example for the target $y = \text{c}$. **Top:** the transducer f. **Middle:** the frontier graph run_c, whose nodes are sets of $(s, b)$ pairs and whose edges are input symbols. Buffers are truncated to length $|y|+1 = 2$: advancing $(q_3, \text{cd})$ by any input produces cdd, truncated back to cd, yielding the self-loop. Truncation keeps the state space finite while preserving whether the buffer has reached the target and which next output symbol it commits to. Reading $x = \text{a}$ reaches two states via $\varepsilon$-closure: $q_1$ (buffer $= y$) and $q_3$ (buffer $\succ y$). Yellow nodes are universal—every input leads to a non-empty accepting successor—placing a in $\mathcal{Q}(\text{c})$. The teal node $q_2$ has no arc on b, so the BFS fails and b enters $\mathcal{R}(\text{c})$.

The frontier enjoys a useful monotonicity property: extending the target by one symbol can only remove elements from the frontier, never add new ones. This means we can compute the frontier incrementally—filtering rather than rebuilding—as the target grows.

**Proposition C.1** (Frontier containment). *For any source string $x$, target string $y_1 \cdots y_N$ with $N > 0$:*

$$\text{run}_{y_1 \cdots y_N}(x) = \{(s, b) \in \text{run}_{y_1 \cdots y_{N-1}}(x): |b| < N \vee b_N = y_N\} \tag{9}$$

*Proof.* Buffer filtering in step and closure keeps only $(s, b)$ pairs where $b$ and the target agree on their shared prefix. Since FST arcs only append to the buffer, once $b$ is long enough to have an $N$th symbol and $b_N \neq y_N$, then all descendant buffers would be filtered out by the $y_1 \cdots y_N$-compatibility filter. Therefore, filtering $\text{run}_y(x)$ at the end is equivalent to filtering at each step.

To see this, note that the ($\supseteq$) direction holds because $y_1 \cdots y_N$-compatible elements are also $y_1 \cdots y_{N-1}$-compatible (the former is strictly more restrictive). The ($\subseteq$) direction holds because elements that are $y_1 \cdots y_{N-1}$-compatible but *not* $y_1 \cdots y_N$-compatible (i.e., $|b| \geq N$ and $b_N \neq y_N$) cannot produce descendants that become $y_1 \cdots y_N$-compatible, since buffers only grow. ∎

**Proposition C.2.** *Prop. C.1 generalizes beyond single-symbol extensions: for any $y' \preceq y$:*

$$\mathsf{run}_y(x) = \{(s, b) \in \mathsf{run}_{y'}(x) : b \preceq y \vee b \succeq y\} \tag{10}$$

*Proof.* By induction on $|y| - |y'|$. The base case $y = y'$ is immediate. For the inductive step, write $y = y' y_N$ with $y' \preceq y'$. By Prop. C.1, $\mathsf{run}_y(x) = \{(s, b) \in \mathsf{run}_{y'}(x) : |b| < N \vee b_N = y_N\}$. By the induction hypothesis, $\mathsf{run}_{y'}(x) = \{(s, b) \in \mathsf{run}_{y'}(x) : b \preceq y' \vee b \succeq y'\}$. Substituting, $(s, b) \in \mathsf{run}_{y'}(x)$ survives both filters iff $b$ agrees with $y$ on all positions up to $\min(|b|, N)$, i.e., $b \preceq y \vee b \succeq y$. ∎

Prop. C.2 is the key property that enables warm-starting `decompose_next` (§C.4) from any target prefix's cached frontiers.

## C.2 Frontier-Based Checks (`is_cylinder`, `is_member`, `is_live`)

Recall that the decomposition algorithm (Fig. 2) relies on three checks (`is_cylinder`, `is_member`, `is_live`). This section describes how to compute them in terms of the frontier.

```
79  def is_cylinder(x, y):   # memoize
80      N ← |y|
81      F ← trunc_buf_y(run_y(x))
82      V ← {F}; q ← QUEUE({F})
83      while q:
84          F ← q.pop()
85          # Accepting: some pair has matched
86          # target and is accepting
87          if ¬∃ (s, b) ∈ F: b ⪰ y ∧ s ∈ F:
88              return False
89          # Complete: every input leads to a
90          # non-empty successor
91          for x' ∈ X:
92              F' ← trunc_buf_y(step_y(F, x'))
93              if F' = ∅: return False
94              if F' ∉ V: V.add(F'); q.push(F')
95      return True

96  def is_member(x, y):
97      return ∃(s, b) ∈ run_y(x):
98          b ⪰ y ∧ s ∈ F

99  def is_live(x, y):
100     return run_y(x) ≠ ∅

101 def trunc_buf_y(F):
102     N ← |y|
103     return {(s, b_≤N) | (s, b) ∈ F,
104                 b ⪯ y ∨ y ⪯ b}
```

Figure 9: Frontier-based checks. `is_cylinder`$(x, y)$ verifies universality via powerset BFS over projected frontiers. `is_member` checks for an accepting covering state; `is_live` checks whether the frontier is non-empty.

To check whether $\langle x \rangle \subseteq \mathcal{P}(y)$, the frontier-based `is_cylinder` applies `trunc_buf` to the frontier $\mathsf{run}_y(x)$, which truncates buffers to length $|y|$, yielding a **truncated frontier**—a set of $(s, b)$ pairs whose buffers are compatible with $y$. It then verifies universality via BFS over truncated frontiers: each successor frontier is obtained by applying `step_y` followed by `trunc_buf`. Unlike the ordinary frontier, where buffers can grow without bound, truncated buffers must be prefixes of $y$, so at most $|y|+1$ distinct buffers exist, making the truncated frontier state space finite. Each truncated frontier corresponds to a state of the determinized precover machine P, so the BFS terminates. The BFS checks that every truncated frontier is both accepting (some pair has $b \succeq y$ and $s \in F$) and complete (every input symbol leads to a non-empty successor). Without this exploration, two nondeterministic paths may yield the same string yet end in different states $s_1$ and $s_2$, where neither is universal on its own, although $[\![ f_{[s_1]} ]\!] \cup [\![ f_{[s_2]} ]\!] = \mathcal{X}^*$.[24] The precomputation of IP-universal states and a cylinder check shortcut are discussed in §C.6.

---

[24] Universality is decidable for finite-state automata; efficient algorithms include antichain-based simulation (De Wulf et al., 2006), bisimulation up to congruence (Bonchi & Pous, 2015), and classical equivalence checking (Meyer & Stockmeyer, 1972).

When the cylinder check fails, membership reduces to checking whether the frontier contains a covering accepting pair ($b \succeq y$ and $s \in \mathsf{F}$): if so, $x \in \mathcal{P}(y)$ and $x$ enters $\mathcal{R}(y)$.

Liveness is non-emptiness of the frontier. Since frontier pairs are already filtered to target-compatible buffers, a non-empty frontier witnesses that some extension of $x$ can reach the precover.

### C.3 Pruning (`prune`)

Even when the decomposition is finite, the quotient and remainder can be very large. Enumerating all of them is often impractical, so pruning is essential for scalability.[25] At minimum, `prune` drops source strings with zero prefix probability, since these cannot contribute any mass:

```
105  def prune(q′):
106      return {x ∈ q′ : p⃗𝒳(x) > 0}
```

This simplistic filter drops only strings with *zero* probability. In practice, we use a probability-mass pruning strategy: given candidates q′ with total mass $Z = \sum_{x \in q'} \overrightarrow{p_{\mathcal{X}}}(x)$, we sort by descending $\overrightarrow{p_{\mathcal{X}}}$ and greedily retain candidates until their cumulative mass reaches $(1 - \tau) \cdot Z$ or the capacity $n_{\max}$ is reached, whichever comes first. The resulting decomposition is approximate: discarded candidates contribute at most $\tau \cdot Z$ mass per step, but the error vanishes as $\tau \to 0$.

```
107  def prune_prob_mass(τ, n_max):
108      def prune(q′):
109          M ← |q′|
110          q′ ← sort_descending(q′, key=p⃗𝒳)
111          w ← [p⃗𝒳(x) for x ∈ q′]   # w₁ ≥ ⋯ ≥ w_M
112          W ← cumulative_sum(w)
113          Z ← W_M
114          q ← ∅
115          for m ∈ {1, … , min(n_max, M)}:
116              q.add(q′_m)
117              if W_m ≥ (1 − τ)Z: break
118          return q
119      return prune
```

**Approximation bound.** At each step, the discarded mass is at most $\tau \cdot Z$. As $\tau \to 0$ and $n_{\max} \to \infty$, the algorithm converges to the exact decomposition.

**Finite termination.** Even when the decomposition is finite, its size can be enormous. Pruning makes the algorithm tractable by bounding the queue at $n_{\max}$ candidates per step, and also guarantees termination for transducers whose decomposition is infinite (§D.3).

**Backtracking.** Pruning may occasionally lead to dead ends where no valid extension remains. When this occurs while scoring an observed sequence, the next symbol $y$ receives zero conditional probability under the approximate $\overrightarrow{p_{\mathcal{Y}}}(\cdot \mid y)$. We recover using the procedure in Fig. 10, which takes the current target string $y$, the next symbol $y$, the current pruning threshold $\tau$, and hyperparameters: maximum retries $R$, threshold floor $\tau_{\min}$, and maximum eviction depth $D_{\max}$. On each retry, $\tau$ is halved (floored at $\tau_{\min}$ to prevent unlimited q growth), and cached decompositions for prefixes of $y$ are evicted at exponentially increasing depth $D = \min(2^{i-1}, D_{\max})$, forcing `decompose` to recompute with the relaxed threshold. The original threshold is restored afterward. The procedure is heuristic: there is no guarantee that a fixed number of retries with halved thresholds will recover. Frequent backtracking signals that the pruning threshold is overly aggressive; in such cases, the overhead of repeated eviction and recomputation can dominate runtime. In our experiments, we use $R = 20$, $\tau_{\min} = 10^{-10}$, $D_{\max} = 32$. Backtracking is infrequent in practice: for $\tau \leq$ 1e-3, zero fallbacks occur across all experiments. At the coarsest threshold ($\tau =$ 1e-1), the mean number of fallbacks per paragraph reaches at most 96 for $p_{\mathsf{gpt2}} \circ \mathsf{f}_\alpha$ (max 159), 5.1 for $p_{\mathsf{gpt2}} \circ \mathsf{f}_\alpha \circ \mathsf{f}_{\mathrm{ptb}}$ (max 18), and 0.4 for $p_{\mathsf{dna}} \circ \mathsf{f}_{\mathrm{dna2aa}}$ (max 5).

---

[25]An alternative to deterministic pruning is importance sampling with a proposal distribution over source strings; we leave this direction to future work.

```
120  def backtrack(𝒚, y, τ, R, τ_min, D_max):
121      τ_0 ← τ
122      for i ∈ 1, … , R:
123          τ ← max(τ/2, τ_min)
124          D ← min(2^{i−1}, D_max)
125          for d ∈ 0, … , D−1:
126              if d > |𝒚|: break
127              evict cached decompose(𝒚_{<|𝒚|−d})
128          p̄ ← next_dist(𝒚)
129          if p̄[y] > 0:
130              τ ← τ_0
131              return p̄
132      τ ← τ_0
133      raise DeadEnd
```

Figure 10: Backtracking. On each retry, halve $\tau$ (floored at $\tau_{\min}$) and evict cached decompositions for progressively longer prefixes of $\boldsymbol{y}$ (depth $D = \min(2^{i-1}, D_{\max})$), then recompute `next_dist`. Restore $\tau$ afterward.

## C.4 FAST NEXT-SYMBOL DECOMPOSITION (`decompose_next`)

Since $\langle \mathcal{Q}(\boldsymbol{y}y) \rangle \subseteq \langle \mathcal{Q}(\boldsymbol{y}) \rangle$, we have $\mathcal{P}(\boldsymbol{y}y) = (\langle \mathcal{Q}(\boldsymbol{y}) \rangle \cap \langle \mathcal{Q}(\boldsymbol{y}y) \rangle) \sqcup \mathcal{R}(\boldsymbol{y}y)$. The decomposition for $\boldsymbol{y}y$ can therefore be obtained with incremental work starting from $\mathcal{Q}(\boldsymbol{y}) \cup \mathcal{R}(\boldsymbol{y})$. The structural reason is monotonicity: the precover for $\boldsymbol{y}$ and $\boldsymbol{y}y$ share the same alphabet and initial states; only the accepting states differ, as acceptability depends on which buffers cover the target. This is formalized by the frontier containment property (Prop. C.1).

The `decompose_next` method computes $\mathcal{Q}(\boldsymbol{y}y)$ and $\mathcal{R}(\boldsymbol{y}y)$ for all $y \in \mathcal{Y}$ simultaneously, as well as the exact preimage $f^{-1}(\boldsymbol{y})$. This is the key efficiency improvement over calling `decompose`($\boldsymbol{y}y$) independently for each $y$.

The BFS is seeded with $\mathcal{Q}(\boldsymbol{y}) \cup \mathcal{R}(\boldsymbol{y})$ (line 152). For each source string $\boldsymbol{x}$ in the queue, we first check whether `is_exact_member`($\boldsymbol{x}, \boldsymbol{y}$) (line 156) and, if so, record $\boldsymbol{x}$ in the preimage set. Next, for each $y \in$ `reachable_outputs`($\boldsymbol{x}, \boldsymbol{y}$) (defined below; line 159), we classify $\boldsymbol{x}$ as a cylinder or member for $\boldsymbol{y}y$; by functionality of f, at most one $y$ can be a cylinder (Prop. C.4). If a cylinder was found for some $y$, the entire subtree rooted at $\boldsymbol{x}$ is absorbed—we skip extension for *all* symbols (line 164). Otherwise, we extend by all reachable source symbols (line 165).

The `decompose_next` algorithm relies on two mechanisms described next: target recursion for cheap frontier computation, and three frontier-based helpers. We now introduce these.

**Target recursion.** The call `is_cylinder`($\boldsymbol{x}, \boldsymbol{y}y$) internally triggers $\text{run}_{\boldsymbol{y}y}(\boldsymbol{x})$. Target recursion is used opportunistically when the parent frontier $\text{run}_{\boldsymbol{y}}(\boldsymbol{x})$ is already cached: the frontier for the extended target is derived by filtering rather than recomputing the entire source-string chain (Prop. C.1). This is the typical case in `decompose_next`, where `decompose`($\boldsymbol{y}$) has already populated the cache. This optimization is transparent: `is_cylinder` is called unchanged; the logic lives entirely inside `run`.

**Per-symbol frontier helpers.** The BFS loop uses three frontier-based helpers.

- `is_exact_member`($\boldsymbol{x}, \boldsymbol{y}$) checks whether $\boldsymbol{x}$ maps to $\boldsymbol{y}$ exactly, i.e., `is_exact_member`($\boldsymbol{x}, \boldsymbol{y}$) $\iff$ ($f(\boldsymbol{x}) = \boldsymbol{y}$).
- `reachable_inputs`($\boldsymbol{x}, \boldsymbol{y}$) conservatively approximates the set of source symbols $x'$ such that extending $\boldsymbol{x}$ by $x'$ remains compatible with $\boldsymbol{y}$, i.e., $\{x' \in \mathcal{X}: \langle \boldsymbol{x}x' \rangle \cap \mathcal{P}(\boldsymbol{y}) \neq \emptyset\}$. The pseudocode checks for transitions from frontier states whose extended buffer is prefix-compatible with $\boldsymbol{y}$; this may over-report when the resulting state cannot reach acceptance.
- `reachable_outputs`($\boldsymbol{x}, \boldsymbol{y}$) conservatively approximates the set of target symbols $y'$ such that some extension of $\boldsymbol{x}$ can produce $\boldsymbol{y}y'$ as an output prefix, i.e., $\{y' \in \mathcal{Y}: \langle \boldsymbol{x} \rangle \cap \mathcal{P}(\boldsymbol{y}y') \neq \emptyset\}$. The pseudocode checks single-step transitions from frontier states, which may over-report (a transition from a frontier state that cannot reach acceptance) because the frontier is not cotrimmed. Symbols requiring multiple transitions to extend past $\boldsymbol{y}$ are handled by the expansion loop.

All three are computed directly from the frontier $\mathrm{run}_y(x)$:

```
134  def is_exact_member(x, y):            140  def reachable_outputs(x, y):
135     return ∃(s, b) ∈ run_y(x):        141     F ← run_y(x)
136        b = y ∧ s ∈ F                   142     # y' is read from b (committed)
                                           143     # or y (boundary)
137  def reachable_inputs(x, y):           144     return {y' | (s, b) ∈ F,
138     return {x' | (s, b) ∈ run_y(x),
139        (s --x':y'--> s') ∈ T, by' ⪯ y ∨ y ⪯ by'}  145     (s --·:y'--> s') ∈ T(s), yy' ⪯ by'}
```

**Expansion loop.** The following algorithm combines the seeding, shortcuts, and helpers above into a single BFS that computes the per-symbol quotients and remainders for every $y \in \mathcal{Y}$ simultaneously.

```
146  def decompose_next(y):
147     (Q, R) ← decompose(y)
148     qs ← {}; rs ← {}            # map y ↦ set of x
149     ps ← ∅
150     # Remainders were not cylinders for y, hence not for any yy (Prop. C.3)
151     non_cyl ← R
152     q ← Q ∪ R
153     while |q| > 0:
154        q' ← ∅
155        for x in q:
156           if is_exact_member(x, y) : ps.add(x)      # f(x) = y exactly
157           # Check reachable y; at most one y can have x as a cylinder (Prop. C.4).
158           ŷ ← ⊥
159           for y in reachable_outputs(x, y) :
160              if ŷ = ⊥ and x ∉ non_cyl and is_cylinder(x, yy):
161                 qs[y].add(x); ŷ ← y; continue
162              if is_member(x, yy):
163                 rs[y].add(x)
164           if ŷ ≠ ⊥: continue     # absorbed ⇒ skip extension for all y
165           for x' ∈ reachable_inputs(x, y) :
166              q'.add(xx')
167        q ← prune(q')
168     return (qs, rs, ps)
```

Figure 11: Fast next-token decomposition algorithm.

$\mathrm{decompose\_next}(y)$ computes $\mathcal{Q}(yy)$ and $\mathcal{R}(yy)$ for all $y \in \mathcal{Y}$ in a single BFS pass. The $\mathrm{is\_cylinder}(x, yy)$ call triggers $\mathrm{run}_{yy}(x)$, which is computed cheaply via target recursion (Prop. C.1) since the parent frontier $\mathrm{run}_y(x)$ is already cached. The non-cylinder shortcut avoids the universality check entirely for seeds in $\mathcal{R}(y)$. After the BFS, the cache is populated so that subsequent $\mathrm{decompose}(yy)$ calls are free. The exact preimage is collected as a byproduct: if $x$ is a cylinder for some $yy$, every extension produces output strictly longer than $y$, so no extension can be an exact preimage of $y$.

**Cache population.** After the BFS, the computed $\mathcal{Q}(yy)$ and $\mathcal{R}(yy)$ should be written into $\mathrm{decompose}$'s memoization cache. A subsequent $\mathrm{decompose}(yy)$ call (e.g., from $\mathrm{decompose\_next}(yy)$) hits the cache instead of re-running its own BFS.

```
169  def decompose_next(y):
170     ⋯
171     # Populate decompose's cache for all children
172     for y ∈ qs.keys() ∪ rs.keys():
173        cache[decompose, yy] ← (qs[y], rs[y])
174     return ⋯
```

## C.5 Next-Symbol Decomposition Shortcuts

The `decompose_next` algorithm (Fig. 11) shares work across all output symbols by making a single BFS pass for all output symbols at once instead of calling the decomposition for each symbol as in `decompose` (Fig. 2). By starting with the decomposition from the prior step and using target recursion, the overall frontier computations are reused. The remaining bottleneck is the per-symbol cylinder check `is_cylinder` (Fig. 9). Three shortcuts avoid or reduce this cost, each justified by a proposition about the structure of the precovers. *Non-cylinder monotonicity* and *cylinder uniqueness* enable skipping BFS for some settings. *Combined universality* resolves additional symbols by examining how a frontier partitions into committed and boundary elements.

**(1) Non-cylinder shortcut.** Strings in $\mathcal{R}(y)$ cannot be quotient elements for $y$. And since $\langle \mathcal{Q}(yy') \rangle \subseteq \langle \mathcal{Q}(y) \rangle$, they are not quotient elements for the strictly more restrictive $yy$ either. The expensive cylinder check (line 160) is thus skipped for these seeds (line 151).

**Proposition C.3** (Non-cylinder monotonicity). *If $x \in \mathcal{R}(y)$ then $x \notin \mathcal{Q}(yy)$ for all $y \in \mathcal{Y}$.*

*Proof.* We have that $x \in \mathcal{R}(y)$ implies $\langle x \rangle \not\subseteq \mathcal{P}(y)$. Since $\mathcal{P}(yy) \subseteq \mathcal{P}(y)$, and thus $\langle x \rangle \not\subseteq \mathcal{P}(yy)$, so $x \notin \mathcal{Q}(yy)$. ∎

**(2) Cylinder uniqueness shortcut.** Since the transformations we consider are functions, at most one output symbol $y$ can have $x$ as a cylinder element. Once a cylinder element is found, the cylinder check is skipped for the remaining output symbols (line 159).

**Proposition C.4** (Cylinder uniqueness). *For a function $f$ and target prefix $y$, $x \in \mathcal{Q}(yy) \cap \mathcal{Q}(yy') \implies y = y'$.*

*Proof.* Suppose $y \neq y'$ and $x \in \mathcal{Q}(yy) \cap \mathcal{Q}(yy')$. Then $\langle x \rangle \subseteq \mathcal{P}(yy) \cap \mathcal{P}(yy') = f^{-1}(yy\mathcal{Y}^*) \cap f^{-1}(yy'\mathcal{Y}^*) = \emptyset$, where the last equality holds because $f$ is a function and $yy\mathcal{Y}^* \cap yy'\mathcal{Y}^* = \emptyset$. This contradicts $\langle x \rangle \neq \emptyset$. ∎

**(3) Combined universality shortcut.** Given $x \in \mathcal{Q}(y)$ and $\widehat{y} \in \mathcal{Y}$, we want to determine whether $x \in \mathcal{Q}(y\widehat{y})$ using the covering frontier at $y$ without running a full powerset BFS. The frontier $\mathsf{run}(x, y)$ partitions into *committed* elements $(s, b)$ with $b \succeq y\widehat{y}$ (the accumulated output already extends past $y$ with symbol $\widehat{y}$) and *boundary* elements $(s, b)$ with $b = y$ (the next output symbol is undetermined). The committed states alone may not be universal—some input symbols may lack outgoing transitions from committed states. Boundary states can fill the gaps, provided every transition from a boundary state produces $\widehat{y}$ as its next output symbol.

**Proposition C.5** (Combined universality). *Let $x \in \mathcal{Q}(y)$, $\widehat{y} \in \mathcal{Y}$, and define*

$$S_c = \{s \mid (s, b) \in \mathsf{run}(x, y),\ b \succeq y\widehat{y}\},$$
$$S_b = \{s \mid (s, b) \in \mathsf{run}(x, y),\ b = y\}$$

*If the union $S_c \cup S_b$ is universal (i.e., $[\![\mathsf{proj}_{\mathcal{X}}(f_{[S_c \cup S_b]})]\!] = \mathcal{X}^*$), $S_c \cap \mathsf{F} \neq \emptyset$, and every transition from a boundary state produces $\widehat{y}$, except that $\varepsilon$-input transitions may produce $\varepsilon$ (i.e., $s \in S_b$ and $s \xrightarrow{x:y} s' \in \mathsf{T}$ implies $y = \widehat{y}$ or $(x = \varepsilon \wedge y = \varepsilon)$), then $x \in \mathcal{Q}(y\widehat{y})$.*

*Proof.* Let $x' \in \mathcal{X}^*$ be arbitrary. By universality of $S_c \cup S_b$, there exists $(s, b) \in \mathsf{run}(x, y)$ with $s \in S_c \cup S_b$ such that $s$ has an accepting run on $x'$ in the input projection. If $s \in S_c$, then $b \succeq y\widehat{y}$, so the output of the corresponding path in $f$ already begins with $y\widehat{y}$, giving $xx' \in \mathcal{P}(y\widehat{y})$. If $s \in S_b$, then $b = y$. By the exclusivity hypothesis, every non-$\varepsilon$-input transition from $s$ produces output $\widehat{y}$, and $\varepsilon$-input transitions produce $\widehat{y}$ or $\varepsilon$. States reachable from $s$ via $\varepsilon:\varepsilon$ arcs are also in $S_b$ (since the frontier is $\varepsilon$-closed and the buffer is unchanged), so they satisfy the same constraint. Therefore the first non-$\varepsilon$ output along any path from $s$ is $\widehat{y}$, and the output begins with $y\widehat{y}$, giving $xx' \in \mathcal{P}(y\widehat{y})$. Since $x'$ was arbitrary, $\langle x \rangle \subseteq \mathcal{P}(y\widehat{y})$, i.e., $x \in \mathcal{Q}(y\widehat{y})$. ∎

This shortcut is complementary to the IP-universality shortcut (Prop. C.6): IP-universality is a per-state property, while combined universality is a set-level property that leverages partial coverage from both committed and boundary states.

The proposition requires set-level universality of $S_c \cup S_b$ (the input-projection NFA started from those states accepts $\mathcal{X}^*$). In practice, we use a stronger but cheaper sufficient condition: we require every individual state in $S_c \cup S_b$ to be in the precomputed $\mathsf{U}$. This runs in $O(|\mathcal{F}|)$ time, whereas the set-level check requires a powerset BFS that may be as expensive as the full `is_cylinder` check this shortcut is meant to avoid.

```
175  def combined_universal(F, yŷ):
176    S_c ← {s | (s, b) ∈ F, b ⪰ yŷ}    # covered
177    S_b ← {s | (s, b) ∈ F, b = y}      # boundary
178    if ∃s ∈ S_c ∪ S_b: s ∉ U: return False
179    # Non-ε input must produce ŷ; ε input may produce ε
180    for s ∈ S_b:
181      for (_ --x:y--> s') ∈ T(s):
182        if not (y = ŷ or (x = ε ∧ y = ε)):
183          return False
184    return S_c ∩ F ≠ ∅
```

### C.6    Input-Projection Universality Shortcut (`fast_univ_filter`)

Recall that the cylinder check `is_cylinder` verifies whether every extension of a source prefix maps through the target prefix, using a powerset BFS that can be expensive. A cheap sufficient condition avoids it in many cases: a state $s \in \mathsf{S}$ is **input-projection universal** if the input projection of the transducer started from $s$ accepts all of $\mathcal{X}^*$—intuitively, no matter what input follows, the transducer can still produce output. Let $\mathsf{U} \subseteq \mathsf{S}$ be the set of all input-projection universal states.

**Proposition C.6** (IP-universality shortcut). *If some covering frontier state of $\boldsymbol{x}$ with respect to $\boldsymbol{y}$ is IP-universal, then $\boldsymbol{x}$ is a cylinder:* $(\exists (s, \boldsymbol{b}) \in \mathrm{run}(\boldsymbol{x}, \boldsymbol{y}): \boldsymbol{b} \succeq \boldsymbol{y} \wedge s \in \mathsf{U}) \implies \boldsymbol{x} \in \mathcal{Q}(\boldsymbol{y}).$

*Proof.* Let $(s, \boldsymbol{b}) \in \mathrm{run}(\boldsymbol{x}, \boldsymbol{y})$ with $\boldsymbol{b} \succeq \boldsymbol{y}$ and $s \in \mathsf{U}$ be the witness from the antecedent. Then $[\![\mathrm{proj}_{\mathcal{X}}(\mathsf{f}_{[s]})]\!] = \mathcal{X}^*$, so for any $\boldsymbol{x}' \in \mathcal{X}^*$, there exists an accepting run of $\mathsf{f}$ on $\boldsymbol{x}\boldsymbol{x}'$ passing through $s$ with output prefix $\boldsymbol{y}$. Hence $\boldsymbol{x}\boldsymbol{x}' \in \mathcal{P}(\boldsymbol{y})$. Since $\boldsymbol{x}'$ was arbitrary, $\langle \boldsymbol{x} \rangle \subseteq \mathcal{P}(\boldsymbol{y})$, i.e., $\boldsymbol{x} \in \mathcal{Q}(\boldsymbol{y})$. ∎

Fig. 12 shows how to efficiently precompute the set of input-projection universal states. We first input-project the transducer and remove $\varepsilon$-transitions (replacing each path $s \xrightarrow{\varepsilon} \cdots \xrightarrow{\varepsilon} s' \xrightarrow{x} s''$ with a direct transition $s \xrightarrow{x} s''$; see, e.g., Hopcroft et al. 2001, Ch. 2) to obtain an $\varepsilon$-free NFA $\mathsf{A}$ over $\mathcal{X}$. Both operations preserve the state set: input projection drops output labels, and $\varepsilon$-removal modifies only transitions (adding shortcuts through $\varepsilon$-closures), so $\mathsf{S_A} = \mathsf{S}$ and states in $\mathsf{U}$ correspond directly to transducer states in the frontier. A state $s$ of $\mathsf{A}$ is input-projection universal iff it is accepting and, for every $x \in \mathcal{X}$, has at least one $x$-successor that is also input-projection universal—a greatest-fixpoint condition. We compute the fixpoint in linear time using an algorithm analogous to Kahn's algorithm for topological sorting (Kahn, 1962): initialize the candidate set $\mathsf{U}$ to the accepting states of $\mathsf{A}$ and maintain per-state, per-symbol successor counts. When a count reaches zero, remove the state and propagate the change to its predecessors via a reverse transition index. Each transition is processed at most once, yielding $\mathcal{O}(|\mathsf{T_A}|)$ time.

The shortcut can be integrated into `is_cylinder` (Fig. 9) as a fast path in `is_cylinder`:

```
205  def is_cylinder(x, y):  # memoize
206    if fast_univ_filter(x, y): return True  # fast path
207    N ← |y|
208    ...  # continue BFS as before
```

For transducers in which most accepting states are input-projection universal, this shortcut avoids the universality BFS in `is_cylinder` for the vast majority of cylinder checks. In BPE, where all states are input-projection universal, the universality BFS is never needed.

### C.7    Memoization and Batching of $p_{\mathcal{X}}$

The source LM is queried in two places: first during decomposition, where probability-mass pruning calls $\overrightarrow{p_{\mathcal{X}}}(\boldsymbol{x})$ to rank candidates, and then during scoring, where `prefix_prob` and `next_dist` sum $\overrightarrow{p_{\mathcal{X}}}(\boldsymbol{x})$ and $p_{\mathcal{X}}(\boldsymbol{x})$ over the quotient and remainder. These calls involve many overlapping source prefixes. Naïvely, each call recomputes the entire chain $\overrightarrow{p_{\mathcal{X}}}(x_1) \cdot \overrightarrow{p_{\mathcal{X}}}(x_2 \mid x_1) \cdots$ from scratch. Memoizing the conditional $\overrightarrow{p_{\mathcal{X}}}(\cdot \mid \boldsymbol{x})$ avoids this: given a cached state for $\boldsymbol{x}$, extending by one symbol $x$ requires a single conditional evaluation rather than replaying the entire history.

For transformer language models, the cached state is the *key–value (KV) cache*: the accumulated key and value tensors from the self-attention layers. Extending by one token reuses the cached KV entries and runs a single forward pass for the new position. Moreover, most LMs return the complete next-symbol distribution $\overrightarrow{p_{\mathcal{X}}}(\cdot \mid \boldsymbol{x})$ in a single call, so we cache the full distribution rather than

```
185  # Preprocess: input-project and ε-remove
186  A ← eps-remove(proj_𝒳(f))
187  # queue-based greatest fixpoint
188  U ← F_A;    q ← ∅
189  count[·, ·] ← 0
190  for (s --x--> s') ∈ T_A with s' ∈ U:
191      count[s, x] += 1
192  for s ∈ U:
193      if ∃x ∈ 𝒳: count[s, x] = 0:
194          q.add(s)
195  while q ≠ ∅:
196      s' ← q.pop()
197      U ← U \ {s'}
198      for (s, x) with (s --x--> s') ∈ T_A:
199          count[s, x] -= 1
200          if count[s, x] = 0 and s ∈ U:
201              q.add(s)
202  def fast_univ_filter(x, y):
203      # Does any covering frontier state have a universal input projection?
204      return ∃(s, b) ∈ run(x, y): b ⪰ y ∧ s ∈ U
```

Figure 12: Efficient computation of input-projection universal states and the `fast_univ_filter` method that uses them.

individual values. When the extension loop generates children $xx_1, ..., xx_k$, their prefix probabilities are read directly from the cached distribution without additional LM calls.[26]

**Batching while pruning.** The prune step (§C.3) needs $\overrightarrow{p_𝒳}(x)$ for every string $x$ in the queue. Many of these strings share a common prefix but extend by different symbols. We group the queue by parent: for each parent $x_{<N}$ whose next-symbol distribution $\overrightarrow{p_𝒳}(\cdot \mid x_{<N})$ is not yet cached, we issue a single batched LM call. Each call returns the full distribution, from which $\overrightarrow{p_𝒳}(x_{<N}x) = \overrightarrow{p_𝒳}(x_{<N}) \cdot \overrightarrow{p_𝒳}(x \mid x_{<N})$ is read off for all children simultaneously. On GPU-based LMs, batching these parent contexts into a single forward pass amortizes the per-call overhead.

## C.8 REALIZING THE AUTOREGRESSIVE INTERFACE

Given the decomposition, the transduced LM interface from §C.1 is implemented as follows. The prefix probability $\overrightarrow{p_𝒴}(y)$ sums LM prefix probabilities over Q and LM string probabilities over R. The string probability $p_𝒴(y)$ sums over the exact preimage. The next-symbol distribution $\overrightarrow{p_𝒴}(\cdot \mid y)$ is computed via `decompose_next`, which provides the per-symbol decompositions and exact preimage in a single BFS pass; the computational cost is dominated by `decompose_next`, and once the per-symbol quotients, remainders, and preimage are available, the scoring reduces to lookups into the memoized source LM (§C.7).

```
209  def prefix_prob(y):
210      (Q, R) ← decompose(y)
211      return ∑_{x∈Q} p⃗_𝒳(x) + ∑_{x∈R} p_𝒳(x)

212  def prob(y):
213      (_, _, ps) ← decompose_next(y)
214      return ∑_{x∈ps} p_𝒳(x)
```

```
215  def next_dist(y):
216      (qs, rs, ps) ← decompose_next(y)
217      p̄ ← {};  Z ← prefix_prob(y)
218      for x ∈ ps:  p̄[EOS] += p_𝒳(x)/Z
219      for y ∈ qs.keys() ∪ rs.keys():
220          for x ∈ qs[y]:  p̄[y] += p⃗_𝒳(x)/Z
221          for x ∈ rs[y]:  p̄[y] += p_𝒳(x)/Z
222      return p̄
```

Figure 13: Fast implementation of the autoregressive interface using `decompose_next`.

---

[26] Our implementation uses genlm (https://github.com/genlm/genlm-backend) for its seamless caching and auto-batching interface.

For reference, the naïve next-symbol distribution computes `prefix_prob`($\boldsymbol{y}y$) independently for each $y$ (see §C.1). The reference implementation is correct, but inefficient because it does not share work: each `prefix_prob`($\boldsymbol{y}y'$) call triggers its own `decompose` BFS, and it does not benefit from the non-cylinder shortcut, target recursion, or cache population described in §C.4.

### C.9  ALL-UNIVERSAL FAST PATH

When every transducer state is IP-universal, the remainder is always empty and every newly discovered element is immediately a quotient element, so decomposition terminates in a single BFS round. If, additionally, every non-$\varepsilon$-input arc produces at least one output symbol, then boundary frontier elements can be resolved in closed form via a precomputed first-output table. Together, these two conditions let `next_dist` (Fig. 13) simplify into `all_universal_next_dist` (Fig. 14). We describe the simplifications below (normalization is unchanged).

**Decompose (simplified).**  Because every state is IP-universal, $\mathcal{R}(\boldsymbol{y}) = \emptyset$ for all $\boldsymbol{y}$: every source prefix in the precover is in the quotient. The call to `decompose`($\boldsymbol{y}$) (Fig. 2) terminates in a single BFS round—every newly discovered element is immediately classified as a quotient element. This makes `decompose_next` unnecessary: the per-symbol classification is absorbed into the scoring below.

**Score (simplified).**  For each $(\mathcal{F}, \boldsymbol{x}) \in \mathtt{Q}$, the frontier partitions into *committed* elements $(s, \boldsymbol{b})$ with $\boldsymbol{b} \succ \boldsymbol{y}$ and *boundary* elements with $\boldsymbol{b} = \boldsymbol{y}$, as in the combined universality shortcut (§C.5). Committed elements have already determined their next output symbol $\widehat{y} = \boldsymbol{b}_{N+1}$; because all states are IP-universal, each contributes $\overrightarrow{p_{\mathcal{X}}}(\boldsymbol{x})$ directly to $\overline{p}[\widehat{y}]$; thus, no expansion is needed.

Boundary elements ($\boldsymbol{b} = \boldsymbol{y}$) need one more source symbol to commit to an output. The no-$\varepsilon$-output precondition ensures that each non-$\varepsilon$-input arc produces at least one output symbol, so a precomputed *first-output table* $\mathtt{fo}(S, x)$ (defined in the pseudocode) maps each input symbol to the unique output it produces from a given powerstate. This symbol is unique: functionality of $f$ requires that all transitions on $x$ from states in $S$ produce the same output, since IP-universality guarantees each successor has an accepting continuation (so both outputs would be realized). Each boundary element is thus resolved with a single batched LM call: $\ell = \overrightarrow{p_{\mathcal{X}}}(\cdot \mid \boldsymbol{x})$ is queried once per quotient element, and for each $x \in \mathcal{X}$, the mass $\overrightarrow{p_{\mathcal{X}}}(\boldsymbol{x}) \cdot \ell(x)$ is attributed to $\mathtt{fo}(S, x)$. If any boundary state is accepting, the LM's EOS probability contributes $\overrightarrow{p_{\mathcal{X}}}(\boldsymbol{x}) \cdot \ell(\text{EOS})$ to $\overline{p}[\text{EOS}]$.

The $\mathtt{f}_\alpha$ transducers satisfy both preconditions (Tab. 4: all states IP-universal, no $\varepsilon$-output on non-$\varepsilon$-input arcs), explaining their substantially higher throughput (§7). The DNA transducer $\mathtt{f}_{\text{dna2aa}}$ has all states IP-universal but uses $\varepsilon$-output arcs (the first two bases of each codon emit no amino acid), so the fast path does not apply.

```
223 def all_universal_next_dist(y):
224     (Q,_) ← decompose(y) # R = ∅ always
225     p̄ ← {}; Z ← prefix_prob(y)
226     for (F, x) in Q:
227         # Committed: output past target
228         for ŷ ∈ {b_{N+1} | (s, b) ∈ F, b ≻ y}:
229             p̄[ŷ] += p⃗_X(x)
230         # Boundary: output = target
231         S_b ← {s | (s, b) ∈ F, b = y}
232         if S_b ≠ ∅:
233             ℓ ← p⃗_X(· | x)        # one LM call
234             for x ∈ X:
235                 ŷ ← fo(S_b, x)
236                 if ŷ ≠ ⊥:
237                     p̄[ŷ] += p⃗_X(x) · ℓ(x)
238             if ∃ s ∈ S_b: s ∈ F:
239                 p̄[EOS] += p⃗_X(x) · ℓ(EOS)
240     return p̄/Z
```

```
241 def fo(S, x):
242     for (_ --x:y--> s') ∈ T(S, x):
243         # Unique if functional &
244         # all states IP-universal
245         return y
246     return ⊥
```

Figure 14: `all_universal_next_dist`: specialization of `next_dist` (Fig. 13) when all states are IP-universal and every non-$\varepsilon$-input arc produces output. $R = \emptyset$ always, so only Q contributes. Boundary elements are resolved via `fo`, requiring one LM call per quotient element.

# D PROOFS FOR FINITENESS CONDITIONS

## D.1 PROPERTIES OF PREFIX MONOTONE MAPS

**Proposition 6.1.** *The following are equivalent: (i) $f$ is prefix monotone (ii) $f(\langle x \rangle) \subseteq \langle f(x) \rangle$ for all $x \in \mathcal{X}^*$ (iii) $\mathcal{P}(f(x)) = \mathcal{C}(f(x))$ for all $x \in \mathcal{X}^*$ (iv) $f$ is prefix-continuous.*

*Proof.* $(1) \Rightarrow (2)$: Prefix monotonicity means that for any $x, x' \in \mathcal{X}^*$, if we have that $x \preceq xx'$ then $f(x) \preceq f(xx')$ and thus that there exists a $y \in \mathcal{Y}^*$ such that $f(xx') = f(x)y$. And since $x'$ was chosen arbitrarily $(2)$ holds.

$(2) \Rightarrow (3)$: Let $x \in \mathcal{X}^*$. Suppose $f(\langle x \rangle) \subseteq \langle f(x) \rangle$. Then, $\langle x \rangle \subseteq f^{-1}(f(\langle x \rangle)) \subseteq \mathcal{P}(f(x))$. Note that, for any $x' \in \mathcal{P}(f(x))$, $\mathcal{P}(f(x')) \subseteq \mathcal{P}(f(x))$. Therefore, $\langle x' \rangle \subseteq \mathcal{P}(f(x')) \subseteq \mathcal{P}(f(x))$ and so $x' \in \mathcal{C}(f(x))$. This shows that $\mathcal{P}(f(x)) = \mathcal{C}(f(x))$.

$(3) \Rightarrow (4)$: By assumption, any element in the precover is an extension of a quotient element, and thus a member in the cylinder over quotient elements.

$(4) \Rightarrow (1)$: Assume $f$ is prefix-continuous, i.e., $\mathcal{P}(y)$ is cylindrical for all $y$. Let $x \preceq x'$. Since $f(x) \preceq f(x)$, we have $x \in \mathcal{P}(f(x))$. By prefix-continuity, $\mathcal{P}(f(x))$ is cylindrical, so $\langle x \rangle \subseteq \mathcal{P}(f(x))$. In particular, $x' \in \mathcal{P}(f(x))$, which gives $f(x) \preceq f(x')$. ∎

## D.2 Quotient Bound for Strict-Prefix Monotone Maps

The following proposition bounds the size of the quotient when the map is strict-prefix monotone.

**Proposition D.1.** *Let $f: \mathcal{X}^* \to \mathcal{Y}^*$ be a strict-prefix monotone map. Then, for every $\boldsymbol{y} \in \mathcal{Y}^*$:*

1. $f^{-1}(\boldsymbol{y}) = \mathcal{Q}(\boldsymbol{y}) \setminus \bigsqcup_{y \in \mathcal{Y}} \mathcal{Q}(\boldsymbol{y}y)$
2. $\bigsqcup_{y \in \mathcal{Y}} \mathcal{Q}(\boldsymbol{y}y) \subseteq \mathcal{Q}(\boldsymbol{y})(\mathcal{X} \sqcup \{\varepsilon\})$
3. $|\mathcal{Q}(\boldsymbol{y})| \leq (|\mathcal{X}| + 1)^{|\boldsymbol{y}|}$

*Proof.* (1) Observe that (using Proposition 6.1) for every $\boldsymbol{y} \in \mathcal{Y}^*$

$$\mathcal{C}(\boldsymbol{y}) = \mathcal{P}(\boldsymbol{y}) = f^{-1}(\boldsymbol{y}) \sqcup \bigsqcup_{y \in \mathcal{Y}} \mathcal{P}(\boldsymbol{y}y) = f^{-1}(\boldsymbol{y}) \sqcup \bigsqcup_{y \in \mathcal{Y}} \mathcal{C}(\boldsymbol{y}y) \tag{11}$$

In particular, the minimality of $\mathcal{Q}(\boldsymbol{y})$ in $\mathcal{P}(\boldsymbol{y})$ implies that

$$\mathcal{Q}(\boldsymbol{y}) \cap \bigsqcup_{y \in \mathcal{Y}} \mathcal{Q}(\boldsymbol{y}y) = \mathcal{Q}(\boldsymbol{y}) \cap \bigsqcup_{y \in \mathcal{Y}} \langle \mathcal{Q}(\boldsymbol{y}y) \rangle \tag{12}$$

Consequently, we obtain the identity $f^{-1}(\boldsymbol{y}) = \mathcal{C}(\boldsymbol{y}) \setminus \bigsqcup_{y \in \mathcal{Y}} \mathcal{C}(\boldsymbol{y}y) = \langle \mathcal{Q}(\boldsymbol{y}) \rangle \setminus \bigsqcup_{y \in \mathcal{Y}} \langle \mathcal{Q}(\boldsymbol{y}y) \rangle$, which in turn yields the inclusion

$$\mathcal{Q}(\boldsymbol{y}) \setminus \bigsqcup_{y \in \mathcal{Y}} \mathcal{Q}(\boldsymbol{y}y) = \mathcal{Q}(\boldsymbol{y}) \setminus \bigsqcup_{y \in \mathcal{Y}} \langle \mathcal{Q}(\boldsymbol{y}y) \rangle \subseteq f^{-1}(\boldsymbol{y}) \tag{13}$$

For the reverse inclusion, let $\boldsymbol{x}' \in f^{-1}(\boldsymbol{y})$. Then there exists a unique $\boldsymbol{x}_q \in \mathcal{Q}(\boldsymbol{y})$ such that $\boldsymbol{x}_q \preceq \boldsymbol{x}'$, by definition of the quotient set. Since $f$ is strict-prefix monotone and $f(\boldsymbol{x}_q) \preceq f(\boldsymbol{x}') = \boldsymbol{y}$, we must have $\boldsymbol{x}_q = \boldsymbol{x}'$. This shows that $f^{-1}(\boldsymbol{y}) \subseteq \mathcal{Q}(\boldsymbol{y})$, and completes the proof of (1).

(2) Fix $y \in \mathcal{Y}$, $\boldsymbol{y} \in \mathcal{Y}^*$ and let $\boldsymbol{x} \in \mathcal{Q}(\boldsymbol{y}y)$. There exists a unique $\boldsymbol{x}_y \in \mathcal{Q}(\boldsymbol{y})$ such that $\boldsymbol{x}_y \preceq \boldsymbol{x}$. Intersecting $\langle \boldsymbol{x}_y \rangle$ with both sides of (11), one obtains

$$\langle \boldsymbol{x}_y \rangle = (\langle \boldsymbol{x}_y \rangle \cap f^{-1}(\boldsymbol{y})) \sqcup \bigsqcup_{y' \in \mathcal{Y}} (\langle \boldsymbol{x}_y \rangle \cap \mathcal{C}(\boldsymbol{y}y')) \tag{14}$$

$$= (\langle \boldsymbol{x}_y \rangle \cap f^{-1}(\boldsymbol{y})) \sqcup \langle \boldsymbol{x} \rangle \sqcup C_{\boldsymbol{x}} \tag{15}$$

where $C_{\boldsymbol{x}} \stackrel{\text{def}}{=} \bigsqcup_{y' \in \mathcal{Y}} (\langle \boldsymbol{x}_y \rangle \cap \mathcal{C}(\boldsymbol{y}y')) \setminus \langle \boldsymbol{x} \rangle$. Note that this set is cylindrical. By strict monotonicity we have $|\langle \boldsymbol{x}_y \rangle \cap f^{-1}(\boldsymbol{y})| \leq 1$. Hence either: (i) $\langle \boldsymbol{x}_y \rangle \cap f^{-1}(\boldsymbol{y}) = \emptyset$, in which case

$$\boldsymbol{x} = \boldsymbol{x}_y \in \mathcal{Q}(\boldsymbol{y}) \cap \mathcal{Q}(\boldsymbol{y}y),$$

or (ii) $\langle \boldsymbol{x}_y \rangle \cap f^{-1}(\boldsymbol{y}) = \{\boldsymbol{x}_y\}$. Then there exists a word $z \in \mathcal{X}^*$ such that $\boldsymbol{x} = \boldsymbol{x}_y z$. Let $z' \prec z$. If $z' \neq \varepsilon$, then $\boldsymbol{x}_y z' \in C_{\boldsymbol{x}}$. Since $C_{\boldsymbol{x}}$ is cylindrical, it contains $\langle \boldsymbol{x}_y z' \rangle$ and in particular it must contain $\langle \boldsymbol{x} \rangle$ contradicting the definition of $C_{\boldsymbol{x}}$. Therefore no non-empty proper prefix $z' \prec z$ exists, so $z$ is a single symbol $x \in \mathcal{X}$ and

$$\boldsymbol{x} = \boldsymbol{x}_y x \in \mathcal{Q}(\boldsymbol{y})\mathcal{X}.$$

(3) For any $\boldsymbol{y} \in \mathcal{Y}^*$ and any $y \in \mathcal{Y}$, we have from (2),

$$|\mathcal{Q}(\boldsymbol{y}y)| \leq (|\mathcal{X}| + 1)|\mathcal{Q}(\boldsymbol{y})| \tag{16}$$

Iterating this bound along any string $\boldsymbol{y} \in \mathcal{Y}^*$, we have:

$$|\mathcal{Q}(\boldsymbol{y})| \leq (|\mathcal{X}| + 1)^{|\boldsymbol{y}|}|\mathcal{Q}(\varepsilon)| = (|\mathcal{X}| + 1)^{|\boldsymbol{y}|} \qquad \blacksquare$$

### D.3 Sufficient Conditions for Finite Quotients and Remainders

Lemma 6.1 below gives sufficient conditions for when one can derive a finite precover decomposition. An example of such a transducer is given in Example 3.

**Lemma 6.1.** *Let $f \colon \mathcal{X}^* \to \mathcal{Y}^*$ be a function realized by a transducer $\mathsf{f}$. The decomposition $(\mathcal{Q}(\boldsymbol{y}), \mathcal{R}(\boldsymbol{y}))$ is finite for every $\boldsymbol{y} \in \mathcal{Y}^*$ if:*

*(i) No $\varepsilon$-output cycles: $\mathsf{f}$ contains no cycle in which every arc outputs $\varepsilon$.*

*(ii) Safety: Every state of $\mathsf{f}$ is **safe**, defined inductively as the smallest set such that $s$ is safe if: (a) $s$ is IP-universal; (b) $|[\![\mathsf{f}_{[s]}]\!]| < \infty$ (finite closure); or (c) for all transitions $s \xrightarrow{x:y} s'$, $s'$ is safe.*

*Proof.* Fix $\boldsymbol{y} \in \mathcal{Y}^*$; we show that $\mathcal{Q}(\boldsymbol{y})$ and $\mathcal{R}(\boldsymbol{y})$ are finite. Let $\Pi_{\boldsymbol{y}}$ be the set of all paths in $\mathsf{f}$ that emit exactly $\boldsymbol{y}$, i.e., $\Pi_{\boldsymbol{y}} \stackrel{\text{def}}{=} \{s_0 \xrightarrow{x_1:y_1} s_1 \cdots s_{N-1} \xrightarrow{x_N:y_N} s_N \mid s_0 \in \mathsf{I}, y_1 \cdots y_N = \boldsymbol{y}\}$. By condition (i), the $\varepsilon$-output subgraph of $\mathsf{f}$ is acyclic, so any sub-path that emits no output has length at most $|\mathsf{S}|$. Since the total output is $\boldsymbol{y}$, each path has bounded length, and $\Pi_{\boldsymbol{y}}$ is finite. Let $\Pi_{\succeq \boldsymbol{y}}$ be the set of all valid paths formed by extending the roots in $\Pi_{\boldsymbol{y}}$ until they reach a state satisfying a safety base case. By condition (ii), every state of $\mathsf{f}$ is safe—in particular, the end states of paths in $\Pi_{\boldsymbol{y}}$. This implies that no extension path can continue indefinitely without satisfying a base case (IP-universality or finite closure). Since a finite-state transducer has finite branching and no infinite valid extension paths, Kőnig's Lemma implies that the tree of extensions is finite.[27] Thus, $\Pi_{\succeq \boldsymbol{y}}$ is a finite set.

Let $\mathsf{S}_U \stackrel{\text{def}}{=} \{s \in \mathsf{S} \colon s \text{ is IP-universal}\}$ and $\mathsf{S}_C \stackrel{\text{def}}{=} \{s \in \mathsf{S} \colon |[\![\mathsf{f}_{[s]}]\!]| < \infty\}$ be the sets of states satisfying conditions (a) and (b), respectively, and let $\mathsf{S}_S \stackrel{\text{def}}{=} \mathsf{S}_U \cup \mathsf{S}_C$. We can decompose the set of extended paths disjointly based on their end state:

$$\Pi_{\succeq \boldsymbol{y}} = \{\rho \in \Pi_{\succeq \boldsymbol{y}} \colon \rho_{|\rho|} \in \mathsf{S}_U\} \sqcup \{\rho \in \Pi_{\succeq \boldsymbol{y}} \colon \rho_{|\rho|} \in \mathsf{S}_C\} \tag{17}$$

$$= \{\boldsymbol{\pi} \cdot s_{|\boldsymbol{\pi}|} \xrightarrow{\cdots} s_N \mid \boldsymbol{\pi} \in \Pi_{\boldsymbol{y}}, s_N \in \mathsf{S}_U, \{s_{|\boldsymbol{\pi}|}, \dots, s_{N-1}\} \cap \mathsf{S}_S = \emptyset\}$$

$$\sqcup \{\boldsymbol{\pi} \cdot s_{|\boldsymbol{\pi}|} \xrightarrow{\cdots} s_N \mid \boldsymbol{\pi} \in \Pi_{\boldsymbol{y}}, s_N \in \mathsf{S}_C, \{s_{|\boldsymbol{\pi}|}, \dots, s_{N-1}\} \cap \mathsf{S}_S = \emptyset\} \tag{18}$$

The first term corresponds to $\mathcal{Q}(\boldsymbol{y})$ by definition. Since $\Pi_{\succeq \boldsymbol{y}}$ is finite, this set is finite. The second term collects paths leading to states $s \in \mathsf{S}_C$. The remainder $\mathcal{R}(\boldsymbol{y})$ consists of the input strings from these paths, concatenated with the finite language accepted by $s$. Since the number of paths is finite and the closure of each stopping state is finite, $\mathcal{R}(\boldsymbol{y})$ is finite. ∎

**Example 3.** *Consider the transducer below over $\mathcal{X} = \{\mathsf{a}, \mathsf{b}\}$, $\mathcal{Y} = \{\mathsf{a}, \mathsf{b}, \mathsf{d}\}$, encoding a (partial) function $f$. The four states illustrate all cases of safety. State $s_3$ is universal: it is accepting and loops on every input symbol with $\varepsilon$-output, so $L(s_3) = \mathcal{X}^*$ (base case a). State $s_2$ has finite closure: it is accepting with no outgoing transitions, so $L(s_2) = \{\varepsilon\}$ (base case b). State $s_1$ is neither universal nor finite closure, but is safe by the recursive step (c): its successors $s_3$ (via $\mathsf{a}$) and $s_2$ (via $\mathsf{b}$) are both safe. Similarly, $s_0$ is safe because its successors $s_1$ (via $\mathsf{a}$) and $s_2$ (via $\mathsf{b}$) are both safe.*

*For target $\boldsymbol{y} = \mathsf{a}$, the path set $\Pi_{\mathsf{a}} = \{s_0 \xrightarrow{\mathsf{a}:\mathsf{a}} s_1\}$ is finite (condition i), with terminal state $s_1$ safe (condition ii). Extending from $s_1$: input $\mathsf{a}$ reaches the universal state $s_3$, contributing $\mathsf{aa}$ to $\mathcal{Q}(\mathsf{a})$; input $\mathsf{b}$ reaches the finite-closure state $s_2$, contributing $\mathsf{ab}$ to $\mathcal{R}(\mathsf{a})$. The root string $\mathsf{a}$ is also in $\mathcal{R}(\mathsf{a})$: $s_1$ is accepting with $f(\mathsf{a}) = \mathsf{a} \succeq \mathsf{a}$, but $\mathsf{a}$ is not cylindrical since $\mathsf{ab}$ cannot be further extended (stuck at $s_2$). Thus $\mathcal{Q}(\mathsf{a}) = \{\mathsf{aa}\}$ and $\mathcal{R}(\mathsf{a}) = \{\mathsf{a}, \mathsf{ab}\}$.*

---

[27] Kőnig's Lemma states that every infinite tree with finite branching must have an infinite path. Since our alphabet is finite (finite branching) and the Safety condition forbids cycles among non-base-case states (no infinite paths), the resulting tree of extensions must be finite.

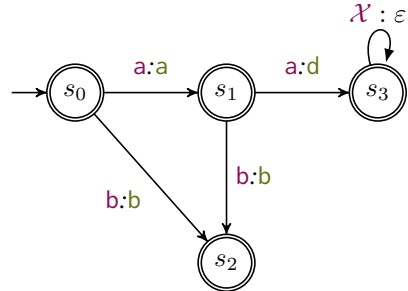

## E    EXAMPLE DECOMPOSITIONS

We give three concrete examples of decompositions and prefix probabilities.

**Example 4.** *The transducer below merges consecutive backticks (` `` `) into a quotation mark (") but otherwise keeps the symbols* a *and* ` *fixed. It can be seen as a minimal example of text normalization.*

$$f = \begin{bmatrix} & \end{bmatrix} \tag{19a}$$

*When computing the prefix probability* $\overrightarrow{p_{\mathcal{Y}}}(\text{`})$*, sequences starting with two backticks get mapped to a quote and thus do not appear in the precover. Instead, a single backtick remains in the remainder, while a backtick followed by* a *goes into the quotient, as any further extensions preserve the initial backtick. We thus have (by visually canceling out continuations that get mapped to quotation marks),*

$$\mathcal{P}(\text{`}) = \{\text{`}, \cancel{\text{``}}, \cancel{\text{```}}, \cancel{\text{``a}}, ...\} \cup \{\text{`a}, \text{`a`}, \text{`aa}, ...\} \tag{19b}$$

$$= \{\text{`}\} \sqcup \{\text{`a}\}\,\mathcal{X}^* \tag{19c}$$

*Thus,* $\overrightarrow{p_{\mathcal{Y}}}(\text{`}) = p_{\mathcal{X}}(\text{`}) + \overrightarrow{p_{\mathcal{X}}}(\text{`a})$*. Similarly, we have* $\mathcal{P}(") = \{\text{``}\}\mathcal{X}^*$ *and thus* $\overrightarrow{p_{\mathcal{Y}}}(") = \overrightarrow{p_{\mathcal{X}}}(\text{``})$*.*

Example 4 demonstrates the efficiency of using the remainder and the quotient; both sets only have a single element, yet they fully specify what probabilities contribute to $\mathcal{P}(\text{`})$. There are, however, edge cases that do not lend themselves to such efficient decompositions.

**Example 5.** *Consider the following mapping* $f$ *and its representation as a transducer* f*:*

$$f(\mathsf{a}^n) = \mathsf{b}^n \text{ \textbf{if} } n \text{ is even \textbf{else} } \mathsf{c}^n \qquad f = \begin{bmatrix} & \end{bmatrix} \tag{20a}$$

*with* $\mathcal{X} = \{\mathsf{a}\}$ *and* $\mathcal{Y} = \{\mathsf{b}, \mathsf{c}\}$*. The precover for* b *is given by*

$$\mathcal{P}(\mathsf{b}) = \{\cancel{\varepsilon}, \cancel{\mathsf{a}}, \mathsf{aa}, \cancel{\mathsf{aaa}}, \mathsf{aaaa}, \cancel{\mathsf{aaaaa}}, \mathsf{aaaaaa}, ...\} \tag{20b}$$

*This example is, unfortunately, challenging for our approach as the remainder set is not finite:*

$$\mathcal{R}(\mathsf{b}) = \left\{\mathsf{a}^{2\,n} \mid n > 0\right\}, \mathcal{Q}(\mathsf{b}) = \emptyset \tag{20c}$$

*In other words, the mapping never releases the evenness constraint, and we sum forever.*

Finally, we give an example of a simple transducer with a single remainder element in Example 6.

**Example 6.** *Consider the following mapping* $f$*, visualized in Fig. 15.*

$$f(\mathsf{a}^n) = \mathsf{b}^n \text{ \textbf{if} } n \neq 2 \text{ \textbf{else} } \mathsf{c} \tag{21a}$$

$\mathcal{X} = \{\mathsf{a}\}$ *and* $\mathcal{Y} = \{\mathsf{b}, \mathsf{c}\}$*. Suppose we want to compute the prefix probability* $\overrightarrow{p_{\mathcal{Y}}}(\mathsf{b})$*.*

$$\mathcal{P}(\mathsf{b}) = \{\mathsf{a}, \cancel{\mathsf{aa}}, \mathsf{aaa}, \mathsf{aaaa}, \mathsf{aaaaa}, ...\} \tag{21b}$$

$$\mathcal{R}(\mathsf{b}) = \{\mathsf{a}\}, \mathcal{Q}(\mathsf{b}) = \{\mathsf{aaa}\} \tag{21c}$$

*Thus,*

$$\overrightarrow{p_{\mathcal{Y}}}(\mathsf{b}) = \sum_{\boldsymbol{x} \in \mathcal{R}(\mathsf{b})} p_{\mathcal{X}}(\boldsymbol{x}) + \sum_{\boldsymbol{x} \in \mathcal{Q}(\mathsf{b})} \overrightarrow{p_{\mathcal{X}}}(\boldsymbol{x}) \tag{21d}$$

$$= p_{\mathcal{X}}(\mathsf{a}) + \overrightarrow{p_{\mathcal{X}}}(\mathsf{aaa}) \tag{21e}$$

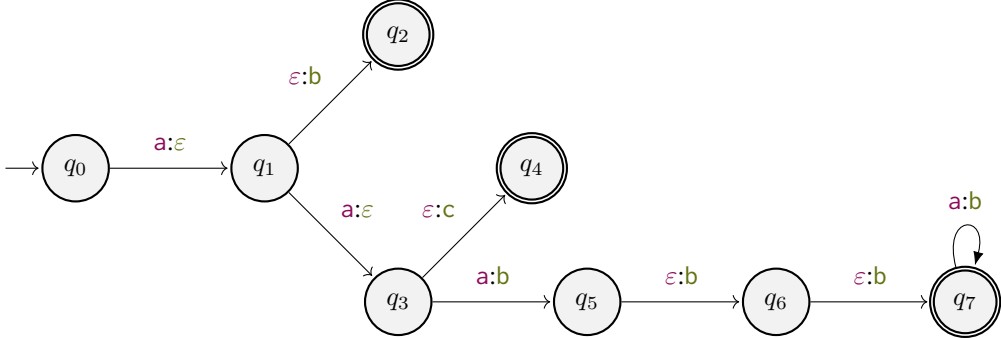

Figure 15: An FST that maps $n$ occurrences of 'a' to the same number of 'b's, except when the input is exactly two 'a's, which are mapped to 'c'.

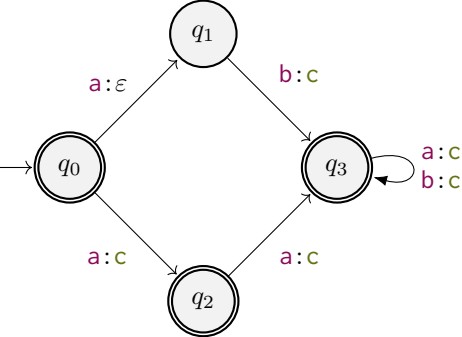

Figure 16: An FST where the $\varepsilon$-output arc $q_0 \xrightarrow{\text{a}:\varepsilon} q_1$ creates a frontier element whose output has not yet reached the target. For target $y = \text{c}$ and source prefix $x = \text{a}$, the frontier is $\mathcal{F} = \{(q_1, \varepsilon), (q_2, \text{c})\}$. The covering states $\{q_2\}$ are not input-projection universal ($q_2$ has no arc on b), so `fast_univ_filter`(a, c) fails and a would be misclassified as non-quotient. However, $q_1$ handles input b and produces c, matching the target. The BFS in `is_cylinder` confirms universality over the full frontier, correctly placing a in $\mathcal{Q}(\text{c})$.

# F  EXPERIMENTAL SETUP

Here we detail the experimental setup for reproducing the experiments in §7 and §G.5.

## F.1  DATASETS

For the tokens-to-byte and PTB experiments in §7, we choose the first 10 paragraphs (excluding headers) of the test split in the `wikitext-2-raw-v1` dataset (Merity et al., 2017) (corresponding to the first 7684 bytes) from the 🤗 Hugging Face datasets library. We use the first 256 bytes of the same dataset and split to run the experiments in §G.5. For the DNA experiments in §7, we sample 65 human proteins[28], each consisting of 4-12 amino acids, with their accession numbers given in Tab. 1.

Table 1: Accession numbers used in this study.

| | | | | |
|---|---|---|---|---|
| C0HLZ5 | P01858 | P0DPI4 | P0DUS0 | P84464 |
| P84465 | P0DOY5 | P67857 | P67858 | P67859 |
| P81826 | P23210 | P85003 | P84071 | P86168 |
| C0HJF1 | C0HJG0 | P02729 | P81010 | P86909 |
| P86922 | B3EWE5 | P0DMM6 | P0DQM6 | P0DQM7 |
| P12481 | P85002 | B3EWR3 | P01358 | P02728 |
| P0C005 | P0DKX2 | P0DMM7 | P0DQM9 | P0DX30 |
| P22103 | P84200 | P84785 | P84868 | P86600 |
| A8C8X2 | B3A0L6 | B3EUR5 | C0HJM6 | C0HLK7 |
| P0DJC3 | P0DJF4 | P69208 | P85444 | P85870 |
| P86942 | A0A0A0MT89 | B3EWS0 | C0HJB6 | C0HL84 |
| C0HL88 | P0C8I8 | P0DQH7 | P0DQH8 | P0DQX4 |
| P0DQX5 | P58805 | P69437 | P82820 | P83127 |

## F.2  MODELS

We conduct experiments using GPT-2 Large (Radford et al., 2019), Llama 3.2-1B, Llama 3.1-8B and Phi-4 (14B; Abdin et al., 2024)[29] from the 🤗 Hugging Face hub (Wolf et al., 2020). We use the **GenLM** library[30] and the √LLM (Kwon et al., 2023) backend to efficiently evaluate the models.

For the PTB experiments we use **GenLM.bytes**[31] to convert token-level models into byte-level models and compose them with $f_\text{ptb}$. For **GenLM.bytes** we use a beam size of $K$=5 and a pruning threshold of 0.001. For the DNA experiments we train a custom GPT-2 Small model[32] on a human DNA dataset[33]. The token set of the model is $\mathcal{X} = \{A, C, G, T\}$, eliminating the need for composing the model with a transducer $f_\alpha$ that maps from subword tokens to bytes. For training parameters, see Tab. 2, for training and validation metrics, see Tab. 3.

## F.3  PARAMETERS

For all experiments, we use the pruning heuristic described in §C.3. For the experiments in §7 we report the results for different values of $n_\text{max} \in \{5000, 10000, 15000, 20000, 25000, 30000\}$. For all other experiments, we set $n_\text{max} = \infty$.

## F.4  GPU USAGE

All experiments in §7 use a single NVIDIA GeForce RTX 4090 (24 GB), except Phi-4, which requires two. The benchmarks in §G.5 use an NVIDIA GeForce RTX 3090.

---

[28] https://www.uniprot.org/uniprotkb?query=Human

[29] Available under openai/gpt2-large, meta-llama/Llama-3.2-1B, meta-llama/Llama-3.1-8B and microsoft/phi-4 at https://huggingface.co.

[30] https://github.com/genlm/genlm-backend

[31] https://github.com/genlm/genlm-bytes

[32] https://huggingface.co/vesteinn/gpt2-dna

[33] https://huggingface.co/datasets/simecek/Human_DNA_v0.

Table 2: Training parameters for the GPT-2 small model, trained on a human DNA.

| Parameter | Value |
|---|---|
| Learning Rate | 0.0003 |
| Optimizer | AdamW |
| | $\beta_1 = 0.9, \beta_2 = 0.999,$ |
| | $\epsilon = 1e{-}8$ |
| Learning Rate Scheduler | Linear |
| Warm-up Steps | 1000 |
| Train Batches (per device) | 64 |
| Eval Batches (per device) | 8 |
| Total Train Batches | 256 (262,144 tokens) |
| Total Eval Batches | 32 |
| Epochs | 10 |
| Seed | 42 |
| Distributed Training | Multi-GPU (4 devices) |
| Mixed Precision | Native AMP |

Table 3: Training and validation metrics for the GPT-2 small model, trained on a human DNA.

| Step | Epoch | Train Loss | Val Loss | Acc (%) |
|---|---|---|---|---|
| 5k | 0.69 | 1.1252 | 1.1206 | 47.45 |
| 10k | 1.38 | 1.0835 | 1.0814 | 49.91 |
| 15k | 2.07 | 1.0641 | 1.0639 | 51.03 |
| 20k | 2.76 | 1.0563 | 1.0547 | 51.63 |
| 25k | 3.45 | 1.0504 | 1.0486 | 52.04 |
| 30k | 4.14 | 1.0439 | 1.0439 | 52.33 |
| 35k | 4.84 | 1.0425 | 1.0407 | 52.54 |
| 40k | 5.53 | 1.0365 | 1.0380 | 52.71 |
| 45k | 6.22 | 1.0325 | 1.0361 | 52.84 |
| 50k | 6.91 | 1.0322 | 1.0341 | 52.96 |
| 55k | 7.60 | 1.0307 | 1.0328 | 53.05 |
| 60k | 8.29 | 1.0267 | 1.0316 | 53.13 |
| 65k | 8.98 | 1.0273 | 1.0306 | 53.20 |
| 70k | 9.67 | 1.0270 | 1.0299 | 53.24 |

### F.5 DETAILS ON TRANSDUCERS USED IN EXPERIMENTS

Tab. 4 contains the number of states, IP-universal states, and transitions for the transducers described in §7. We construct all finite-state transducers using Pynini (Gorman, 2016). Note that for experiments using the Penn Treebank FST ($f_{\text{ptb}}$), we realize $p_{\mathcal{X}} \circ f_\alpha$ using **GenLM.bytes**, thereby keeping the number of states and arcs constant.

Table 4: Number of states, IP-universal states, and transitions.

| Model | States | IP-Univ. States | Transitions |
|---|---|---|---|
| **Tokens to Bytes** | | | |
| $p_{\text{gpt2}} \circ f_\alpha$ | 75,723 | 75,723 | 125,979 |
| $p_{\text{llama1B}} \circ f_\alpha$ | 176,990 | 176,990 | 305,244 |
| $p_{\text{llama8B}} \circ f_\alpha$ | 176,990 | 176,990 | 305,244 |
| $p_{\text{phi4}} \circ f_\alpha$ | 115,244 | 115,244 | 215,593 |
| **Tokens to Words** | | | |
| $p_{\text{gpt2}} \circ f_\alpha \circ f_{\text{ptb}}$ | 479 | 56 | 26,962 |
| $p_{\text{llama1B}} \circ f_\alpha \circ f_{\text{ptb}}$ | 479 | 56 | 26,962 |
| $p_{\text{llama8B}} \circ f_\alpha \circ f_{\text{ptb}}$ | 479 | 56 | 26,962 |
| $p_{\text{phi4}} \circ f_\alpha \circ f_{\text{ptb}}$ | 479 | 56 | 26,962 |
| **DNA to amino acids** | | | |
| $p_{\text{dna}} \circ f_{\text{dna2aa}}$ | 21 | 21 | 84 |

**Constructing the PTB transducer.** We construct the PTB FST by encoding each tokenizer rule[34] as an FST that segments character sequences by inserting a distinguished separator symbol SEP $\notin \mathcal{Y}$. Specifically, the transducer inserts SEP at punctuation and clitic boundaries, and collapses whitespace characters (spaces, tabs, newlines, etc.) into SEP. Note that the resulting transduced language model is thus not a true distribution over PTB tokens, but over characters and separators corresponding to the same boundaries that the PTB tokens would have. This is a pragmatic decision, as the PTB tokenizer can tokenize any sentence into orthographic words. In other words, it would accept an infinite vocabulary. Such a transducer can be built on the fly and would be equivalent to one with infinitely many states. In this paper, we only use the finite version. An example of such a rule is given in Fig. 5, which inserts SEP before a comma if it is not followed by a digit. We then compose

---

[34] See `https://www.nltk.org/_modules/nltk/tokenize/treebank.html#TreebankWordTokenizer` for the full specification.

these FSTs into a single transducer ($f_{\text{ptb}}$). Note that context-dependent rules, such as the one given in Fig. 5, introduce non-IP-universal states. For example, state $q_2$ only accepts $x \in \mathcal{X} \setminus \{\texttt{0-9,}\}$. In fact, of the 479 states in $f_{\text{ptb}}$, just 56 are IP-universal.

**The DNA to amino acid transducer.**  The DNA to amino-acid transducer, described in §7, is partially shown in Fig. 17.

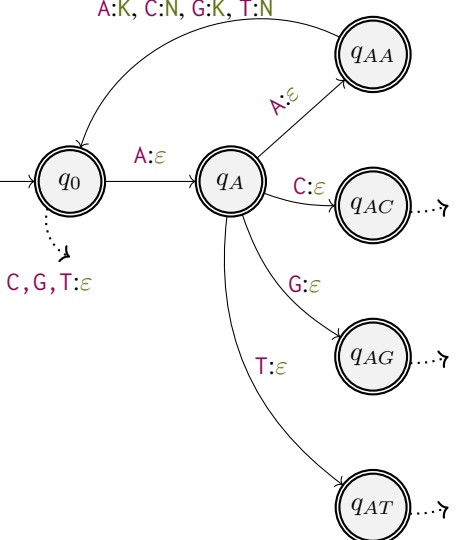

Figure 17: An FST for converting DNA sequences to amino acids. Each triplet of nucleobases maps to one of 20 different amino acids. We only show a proportion of the machine.

# G  ADDITIONAL EXPERIMENTAL RESULTS

Here we provide complementary results to the experiments presented in §7, using the algorithms described in §C. First, in §G.1 we compare our transducer-based method using $f_\alpha$ to **GenLM.bytes** (Vieira et al., 2025a), then give full JSD (§G.2) comparisons for $f_\alpha$, $f_{\text{ptb}}$, and $f_{\text{dna2aa}}$. In §G.4 we report cross-entropy comparisons for $f_\alpha$, and finally ablate the importance of IP-universal states (§G.5).

## G.1  COMPARISON TO VIEIRA ET AL. (2025A)

Vieira et al. (2025a) convert token-level models to byte-level models using a beam-search algorithm parameterized by a beam size $K$. We compare our transducer-based approach with their method by computing the JSD between the two resulting distributions over the paragraphs from WikiText (§F). Tab. 6 uses genlm-bytes ($K = 32$, pruning threshold of 0.0) as a reference and varies our pruning threshold $\tau$. As $\tau$ decreases, JSD drops, and our distributions converge to the baseline. At moderate thresholds ($\tau \le$ 1e-3), JSD is already below 1.5e-3 for all models, confirming that the two methods produce closely matching distributions.

To verify that the two methods compute the same underlying distribution in the exact setting, we compare them with no pruning: $\tau = 0$ (no pruning) for our method and exact inference for genlm-bytes ($K = \infty$). Tab. 5 shows the per-position JSD and maximum absolute difference for the first 10 bytes of the WikiText evaluation data using $p_{\text{gpt2}} \circ f_\alpha$. The JSD values are on the order of 1e-14–1e-15, and the maximum absolute probability difference is at most 3.4e-6, confirming that both methods give the same distribution within floating-point precision.

Table 5: Exact comparison: our method ($\tau = 0$) vs. genlm-bytes (exact) for $p_{\text{gpt2}} \circ f_\alpha$.

| Pos | Byte | JSD | $\max|\Delta p|$ |
|---|---|---|---|
| 0 | R | 3.9e-15 | 7.9e-7 |
| 1 | o | 6.5e-16 | 1.2e-6 |
| 2 | b | 3.5e-15 | 1.2e-6 |
| 3 | e | 4.4e-14 | 1.7e-6 |
| 4 | r | 4.5e-15 | 3.4e-6 |
| 5 | t | 1.8e-15 | 2.4e-6 |
| 6 | ␣ | 1.1e-14 | 1.4e-6 |
| 7 | B | 1.9e-15 | 1.8e-6 |
| 8 | o | 6.1e-15 | 1.2e-6 |
| 9 | u | 1.3e-14 | 1.5e-6 |

Tab. 7 reports the JSD and throughput of the baseline itself at $K \in \{8, 16, 32\}$, showing the accuracy–speed trade-off within their method. Reducing K from 32 to 8 substantially increases throughput while introducing only modest changes in JSD, indicating that the baseline's approximation error at $K = 32$ is small. Note that the method of Vieira et al. (2025a) is designed specifically for strictly prefix monotone transformations, a property that $f_\alpha$ satisfies. This specialization enables a trie-based algorithm that achieves a better accuracy–speed trade-off than our more general-purpose approach on this class of transformations.

Table 6: Average Jensen–Shannon divergence (JSD) and bytes/sec for various thresholds $\tau$ using $p_{\mathcal{X}} \circ f_\alpha$ against a reference distribution from Vieira et al. (2025a) with a beam size of $K$=32. 95% confidence intervals are given in parentheses.

| | $p_{\text{gpt2}} \circ f_\alpha$ | | $p_{\text{llama1B}} \circ f_\alpha$ | |
|---|---|---|---|---|
| $\tau$ | average JSD / byte | bytes / sec | average JSD / byte | bytes / sec |
| 1e-1 | 4.0e-2 (3.7e-2, 4.2e-2) | 70.95 (64.14, 79.65) | 2.8e-2 (2.5e-2, 3.0e-2) | 84.21 (71.77, 99.54) |
| 3e-2 | 1.9e-2 (1.8e-2, 2.1e-2) | 44.68 (40.59, 48.66) | 1.2e-2 (1.1e-2, 1.4e-2) | 72.23 (65.87, 78.10) |
| 1e-2 | 6.7e-3 (5.8e-3, 7.8e-3) | 25.13 (23.83, 26.69) | 4.2e-3 (3.5e-3, 4.9e-3) | 40.45 (37.60, 43.43) |
| 3e-3 | 2.5e-3 (1.9e-3, 3.3e-3) | 10.48 (9.89, 11.12) | 1.1e-3 (8.4e-4, 1.3e-3) | 22.09 (20.94, 23.51) |
| 1e-3 | 1.4e-3 (9.3e-4, 1.9e-3) | 6.21 (5.89, 6.59) | 4.0e-4 (2.9e-4, 5.6e-4) | 11.86 (11.27, 12.49) |
| 3e-4 | 7.6e-5 (6.0e-5, 9.9e-5) | 1.97 (1.84, 2.11) | 1.3e-4 (1.1e-4, 1.5e-4) | 5.90 (5.60, 6.22) |
| 1e-4 | 3.7e-5 (3.0e-5, 4.7e-5) | 0.65 (0.61, 0.71) | 8.8e-5 (8.1e-5, 9.5e-5) | 2.15 (2.04, 2.26) |
| 3e-5 | 2.5e-5 (2.4e-5, 2.7e-5) | 0.21 (0.19, 0.22) | 7.8e-5 (7.3e-5, 8.3e-5) | 1.20 (1.15, 1.27) |
| 1e-5 | 2.5e-5 (2.3e-5, 2.7e-5) | 0.19 (0.17, 0.21) | 7.4e-5 (6.9e-5, 7.7e-5) | 0.62 (0.59, 0.66) |

| | $p_{\text{llama8B}} \circ f_\alpha$ | | $p_{\text{phi4}} \circ f_\alpha$ | |
|---|---|---|---|---|
| $\tau$ | average JSD / byte | bytes / sec | average JSD / byte | bytes / sec |
| 1e-1 | 2.2e-2 (2.0e-2, 2.4e-2) | 79.03 (71.08, 87.28) | 2.5e-2 (2.3e-2, 2.7e-2) | 41.04 (36.96, 45.33) |
| 3e-2 | 8.8e-3 (7.7e-3, 1.0e-2) | 66.41 (62.69, 70.69) | 9.7e-3 (8.6e-3, 1.1e-2) | 36.20 (33.26, 38.87) |
| 1e-2 | 3.5e-3 (2.9e-3, 4.2e-3) | 45.47 (43.10, 47.91) | 4.6e-3 (3.8e-3, 5.4e-3) | 22.85 (21.14, 24.77) |
| 3e-3 | 1.3e-3 (9.3e-4, 1.8e-3) | 26.71 (25.11, 28.34) | 1.8e-3 (1.4e-3, 2.4e-3) | 14.80 (13.70, 15.83) |
| 1e-3 | 5.1e-4 (2.9e-4, 8.4e-4) | 15.73 (14.88, 16.60) | 8.5e-4 (5.0e-4, 1.3e-3) | 8.75 (8.30, 9.27) |
| 3e-4 | 1.3e-4 (1.1e-4, 1.5e-4) | 9.08 (8.68, 9.57) | 1.9e-4 (1.4e-4, 2.5e-4) | 4.87 (4.60, 5.16) |
| 1e-4 | 9.1e-5 (8.0e-5, 1.0e-4) | 4.03 (3.84, 4.23) | 1.0e-4 (8.8e-5, 1.2e-4) | 2.19 (2.07, 2.33) |
| 3e-5 | 7.5e-5 (6.8e-5, 8.3e-5) | 2.38 (2.27, 2.49) | 7.6e-5 (6.8e-5, 8.4e-5) | 1.30 (1.23, 1.37) |
| 1e-5 | 7.3e-5 (6.6e-5, 8.1e-5) | 1.40 (1.33, 1.47) | 7.9e-5 (7.2e-5, 8.8e-5) | 0.74 (0.70, 0.79) |

Table 7: Throughput and Jensen–Shannon divergence (JSD) of Vieira et al. (2025a) at beam sizes $K \in \{8, 16, 32\}$. JSD is measured against $K$=32. 95% confidence intervals are given in parentheses.

| | $p_{\text{gpt2}} \circ f_\alpha$ | | $p_{\text{llama1B}} \circ f_\alpha$ | |
|---|---|---|---|---|
| $K$ | JSD vs $K$=32 | bytes / sec | JSD vs $K$=32 | bytes / sec |
| 8 | 3.0e-5 (2.5e-5, 3.9e-5) | 37.81 (37.59, 37.98) | 7.1e-5 (6.7e-5, 7.6e-5) | 46.94 (45.69, 47.97) |
| 16 | 2.4e-5 (2.3e-5, 2.7e-5) | 28.83 (28.53, 29.13) | 7.1e-5 (6.7e-5, 7.5e-5) | 28.76 (28.42, 29.08) |
| 32 | (not applicable) | 18.12 (17.87, 18.36) | (not applicable) | 5.34 (4.21, 7.19) |

| | $p_{\text{llama8B}} \circ f_\alpha$ | | $p_{\text{phi4}} \circ f_\alpha$ | |
|---|---|---|---|---|
| $K$ | JSD vs $K$=32 | bytes / sec | JSD vs $K$=32 | bytes / sec |
| 8 | 6.8e-5 (6.3e-5, 7.4e-5) | 18.30 (18.17, 18.43) | 4.6e-5 (4.3e-5, 4.9e-5) | 11.59 (11.29, 11.80) |
| 16 | 6.9e-5 (6.2e-5, 7.8e-5) | 10.55 (10.45, 10.65) | 5.3e-5 (4.9e-5, 5.6e-5) | 7.90 (7.84, 7.96) |
| 32 | (not applicable) | 5.85 (5.79, 5.91) | (not applicable) | 4.51 (4.44, 4.56) |

## G.2 JENSEN–SHANNON DIVERGENCE

We benchmark how well the algorithm given in §C approximates the exact distribution when using high pruning thresholds $\tau$ in the probability mass pruning described in §C.3. Across all three settings—token-to-byte, PTB tokenization, and DNA-to-amino-acid—JSD decreases as $\tau$ decreases, at the cost of throughput (bytes/sec).

Tab. 8 reports the token-to-byte results ($p_{\mathcal{X}} \circ f_\alpha$) for all four models. JSD drops by two to three orders of magnitude from $\tau = $ 1e-1 to $\tau = $ 3e-5, with the tightest thresholds reaching JSD below 1e-4 for all models.

Tab. 9 gives the corresponding results for PTB tokenization ($p_{\mathcal{X}} \circ f_\alpha \circ f_{\text{ptb}}$), complementing Fig. 6 (right). Although $f_{\text{ptb}}$ has fewer states than the token-to-byte transducers, only 56 of its 479 states are IP-universal, making lower thresholds computationally expensive; the reference is therefore $\tau = $ 1e-4 (3e-4 for $p_{\text{phi4}}$) rather than 1e-5. Despite this, JSD converges quickly, reaching the order of 1e-5 at $\tau = $ 3e-4 for $p_{\text{gpt2}}$, $p_{\text{llama1B}}$, and $p_{\text{llama8B}}$, while $p_{\text{phi4}}$ converges more slowly.

Tab. 10 reports the DNA-to-amino-acid results ($p_{\mathrm{dna}} \circ f_{\mathrm{dna2aa}}$), where we additionally vary the candidate-set cap $n_{\mathrm{max}}$ to mitigate the combinatorial blow-up inherent in the three-to-one nucleotide-to-amino-acid mapping. Tighter thresholds generally reduce JSD. However, the interaction between $\tau$ and $n_{\mathrm{max}}$ is less uniform, as both the evaluated and reference distributions depend on the cap.

**Incomplete runs.** As shown in §G.3, the decomposition size grows rapidly at tight thresholds— mean $|\mathbb{Q}|$ exceeds 35,000 at $\tau = $ 1e-5 for $f_\alpha$. For some model–threshold combinations, this growth prevents calculating the full distribution at each position in all ten evaluation paragraphs. Specifically, $p_{\mathrm{gpt2}} \circ f_\alpha$ in Tab. 8 completed only 7 of 10 paragraphs at $\tau = $ 1e-5. In Tab. 9, $p_{\mathrm{phi4}} \circ f_\alpha \circ f_{\mathrm{ptb}}$ completed 7 of 10 paragraphs at its reference threshold of $\tau = $ 3e-4, and $p_{\mathrm{llama1B}} \circ f_\alpha \circ f_{\mathrm{ptb}}$ completed 9 of 10 at $\tau = $ 1e-4. Additionally, $p_{\mathrm{phi4}} \circ f_\alpha \circ f_{\mathrm{ptb}}$ completed only 9 of 10 paragraphs at $\tau = $ 1e-1 due to an unrecoverable backtracking failure (Fig. 10). In these cases, JSD and throughput are computed over the paragraphs shared across all thresholds for that model–transducer combination.

Table 8: Average Jensen–Shannon divergence (JSD) and bytes/sec for various thresholds $\tau$ using $p_{\mathcal{X}} \circ f_\alpha$ against a reference distribution ($\tau = $ 1e-5). 95% confidence intervals are given in parentheses.

| $\tau$ | $p_{\mathrm{gpt2}} \circ f_\alpha$ average JSD / byte | bytes / sec | $p_{\mathrm{llama1B}} \circ f_\alpha$ average JSD / byte | bytes / sec |
|---|---|---|---|---|
| 1e-1 | 3.8e-2 (3.4e-2, 4.1e-2) | 76.50 (66.94, 89.01) | 2.8e-2 (2.6e-2, 3.0e-2) | 84.21 (72.49, 99.24) |
| 3e-2 | 1.7e-2 (1.5e-2, 1.9e-2) | 45.61 (39.82, 51.46) | 1.2e-2 (1.1e-2, 1.4e-2) | 72.23 (66.25, 78.94) |
| 1e-2 | 5.9e-3 (4.8e-3, 7.1e-3) | 25.71 (24.08, 27.89) | 4.2e-3 (3.5e-3, 5.0e-3) | 40.45 (37.65, 43.43) |
| 3e-3 | 2.1e-3 (1.4e-3, 2.9e-3) | 11.22 (10.44, 12.07) | 1.1e-3 (8.5e-4, 1.3e-3) | 22.09 (20.90, 23.34) |
| 1e-3 | 1.5e-3 (9.1e-4, 2.3e-3) | 6.49 (6.02, 6.99) | 4.0e-4 (2.8e-4, 5.4e-4) | 11.86 (11.25, 12.54) |
| 3e-4 | 5.5e-5 (4.7e-5, 6.7e-5) | 2.28 (2.09, 2.48) | 1.2e-4 (1.1e-4, 1.4e-4) | 5.90 (5.59, 6.23) |
| 1e-4 | 3.6e-5 (2.9e-5, 4.8e-5) | 1.00 (0.92, 1.09) | 7.4e-5 (6.9e-5, 8.0e-5) | 2.15 (2.03, 2.26) |
| 3e-5 | 2.4e-5 (2.2e-5, 2.7e-5) | 0.44 (0.40, 0.49) | 5.2e-5 (4.9e-5, 5.6e-5) | 1.20 (1.14, 1.26) |
| 1e-5 | (not applicable) | 0.19 (0.17, 0.21) | (not applicable) | 0.62 (0.59, 0.65) |

| $\tau$ | $p_{\mathrm{llama8B}} \circ f_\alpha$ average JSD / byte | bytes / sec | $p_{\mathrm{phi4}} \circ f_\alpha$ average JSD / byte | bytes / sec |
|---|---|---|---|---|
| 1e-1 | 2.3e-2 (2.1e-2, 2.5e-2) | 79.03 (71.62, 87.24) | 2.5e-2 (2.3e-2, 2.7e-2) | 41.04 (36.75, 45.30) |
| 3e-2 | 8.9e-3 (7.8e-3, 1.0e-2) | 66.41 (61.96, 70.51) | 9.9e-3 (8.7e-3, 1.1e-2) | 36.20 (33.36, 38.85) |
| 1e-2 | 3.5e-3 (2.9e-3, 4.4e-3) | 45.47 (43.16, 48.08) | 4.7e-3 (3.8e-3, 5.6e-3) | 22.85 (21.06, 24.55) |
| 3e-3 | 1.3e-3 (9.4e-4, 1.8e-3) | 26.71 (25.17, 28.24) | 1.8e-3 (1.3e-3, 2.4e-3) | 14.80 (13.74, 15.78) |
| 1e-3 | 5.2e-4 (3.0e-4, 8.4e-4) | 15.73 (14.89, 16.61) | 8.3e-4 (4.8e-4, 1.2e-3) | 8.75 (8.30, 9.27) |
| 3e-4 | 1.2e-4 (1.1e-4, 1.4e-4) | 9.08 (8.64, 9.60) | 1.9e-4 (1.1e-4, 2.9e-4) | 4.87 (4.61, 5.19) |
| 1e-4 | 8.3e-5 (7.3e-5, 9.6e-5) | 4.03 (3.83, 4.26) | 7.4e-5 (6.1e-5, 9.2e-5) | 2.19 (2.05, 2.33) |
| 3e-5 | 5.9e-5 (5.3e-5, 6.6e-5) | 2.38 (2.27, 2.49) | 3.7e-5 (3.4e-5, 4.0e-5) | 1.30 (1.23, 1.37) |
| 1e-5 | (not applicable) | 1.40 (1.34, 1.47) | (not applicable) | 0.74 (0.70, 0.79) |

Table 9: Average Jensen–Shannon divergence (JSD) and bytes/sec for various thresholds $\tau$ using $p_{\mathcal{X}} \circ f_\alpha \circ f_{\mathrm{ptb}}$ against a reference distribution ($\tau = $ 1e-4; 3e-4 for $p_{\mathrm{phi4}}$). 95% confidence intervals are given in parentheses.

| | $p_{\mathrm{gpt2}} \circ f_\alpha \circ f_{\mathrm{ptb}}$ | | $p_{\mathrm{llama1B}} \circ f_\alpha \circ f_{\mathrm{ptb}}$ | |
| $\tau$ | average JSD / byte | bytes / sec | average JSD / byte | bytes / sec |
|---|---|---|---|---|
| 1e-1 | 3.9e-3 (3.3e-3, 4.7e-3) | 26.65 (25.79, 27.48) | 3.0e-3 (2.6e-3, 3.5e-3) | 30.44 (28.31, 32.19) |
| 3e-2 | 1.4e-3 (1.2e-3, 1.5e-3) | 17.03 (16.42, 17.66) | 1.1e-3 (9.3e-4, 1.3e-3) | 17.45 (16.67, 18.27) |
| 1e-2 | 5.7e-4 (5.0e-4, 6.7e-4) | 10.79 (10.23, 11.36) | 3.7e-4 (3.4e-4, 4.0e-4) | 10.54 (10.03, 11.09) |
| 3e-3 | 1.7e-4 (1.5e-4, 1.9e-4) | 4.52 (4.29, 4.79) | 1.3e-4 (1.2e-4, 1.4e-4) | 8.30 (7.98, 8.67) |
| 1e-3 | 5.5e-5 (5.1e-5, 6.1e-5) | 2.01 (1.88, 2.15) | 6.1e-5 (5.8e-5, 6.5e-5) | 4.13 (3.91, 4.36) |
| 3e-4 | 2.0e-5 (1.7e-5, 2.4e-5) | 0.81 (0.77, 0.85) | 3.3e-5 (3.1e-5, 3.5e-5) | 1.67 (1.59, 1.76) |
| 1e-4 | (not applicable) | 0.20 (0.16, 0.25) | (not applicable) | 0.70 (0.66, 0.74) |

| | $p_{\mathrm{llama8B}} \circ f_\alpha \circ f_{\mathrm{ptb}}$ | | $p_{\mathrm{phi4}} \circ f_\alpha \circ f_{\mathrm{ptb}}$ | |
| $\tau$ | average JSD / byte | bytes / sec | average JSD / byte | bytes / sec |
|---|---|---|---|---|
| 1e-1 | 2.5e-3 (2.1e-3, 3.0e-3) | 19.25 (18.79, 19.76) | 4.6e-3 (3.2e-3, 6.5e-3) | 10.07 (9.53, 10.67) |
| 3e-2 | 1.0e-3 (8.6e-4, 1.2e-3) | 13.38 (12.99, 13.83) | 3.8e-3 (2.6e-3, 5.1e-3) | 5.87 (5.37, 6.40) |
| 1e-2 | 6.0e-4 (4.7e-4, 7.8e-4) | 9.49 (9.16, 9.88) | 3.2e-3 (2.2e-3, 4.4e-3) | 2.94 (2.59, 3.35) |
| 3e-3 | 1.1e-4 (1.0e-4, 1.2e-4) | 5.90 (5.62, 6.20) | 4.2e-4 (3.0e-4, 5.8e-4) | 2.95 (2.68, 3.26) |
| 1e-3 | 5.1e-5 (4.7e-5, 5.5e-5) | 3.29 (3.13, 3.45) | 3.6e-4 (2.5e-4, 4.8e-4) | 1.54 (1.37, 1.74) |
| 3e-4 | 2.3e-5 (2.1e-5, 2.5e-5) | 1.42 (1.35, 1.49) | (not applicable) | 0.48 (0.43, 0.54) |
| 1e-4 | (not applicable) | 0.56 (0.53, 0.59) | (not applicable) | (not applicable) |

Table 10: Average Jensen–Shannon divergence (JSD) and bytes/sec for various thresholds $\tau$ and a reference ($\tau =$ 1e-6) using $p_{\mathsf{dna}} \circ \mathsf{f}_{\mathrm{dna2aa}}$. 95% confidence intervals are given in parentheses. We limit the candidate-set size ($n_{\max}$) to mitigate the combinatorial blow-up with increasing sequence length.

| $\tau$ | $p_{\mathsf{dna}} \circ \mathsf{f}_{\mathrm{dna2aa}}$ ($n_{\max} = 5000$) | | $p_{\mathsf{dna}} \circ \mathsf{f}_{\mathrm{dna2aa}}$ ($n_{\max} = 10000$) | |
|---|---|---|---|---|
| | average JSD / byte | bytes / sec | average JSD / byte | bytes / sec |
| 1e-1 | 2.8e-2 (2.4e-2, 3.2e-2) | 9.55 (8.44, 10.90) | 2.8e-2 (2.5e-2, 3.3e-2) | 9.54 (8.45, 10.82) |
| 3e-2 | 3.2e-3 (2.8e-3, 3.6e-3) | 2.90 (2.57, 3.26) | 3.3e-3 (2.9e-3, 3.7e-3) | 2.03 (1.74, 2.39) |
| 1e-2 | 6.6e-4 (4.4e-4, 9.0e-4) | 2.20 (1.99, 2.47) | 7.4e-4 (5.0e-4, 1.0e-3) | 1.42 (1.25, 1.63) |
| 3e-3 | 1.6e-4 (7.8e-5, 2.7e-4) | 1.89 (1.72, 2.11) | 2.7e-4 (1.1e-4, 4.9e-4) | 1.16 (1.03, 1.31) |
| 1e-3 | 7.1e-5 (1.7e-5, 1.6e-4) | 1.79 (1.62, 1.97) | 6.6e-5 (1.7e-5, 1.5e-4) | 1.08 (0.97, 1.22) |
| 3e-4 | 4.9e-5 (9.1e-6, 1.1e-4) | 1.73 (1.58, 1.93) | 3.5e-5 (6.7e-6, 7.9e-5) | 1.04 (0.93, 1.17) |
| 1e-4 | 5.8e-5 (1.1e-5, 1.2e-4) | 1.72 (1.56, 1.91) | 2.1e-5 (9.9e-7, 7.9e-5) | 1.02 (0.92, 1.16) |
| 3e-5 | 3.0e-5 (7.9e-7, 7.9e-5) | 1.71 (1.55, 1.89) | 2.1e-5 (6.6e-7, 6.2e-5) | 1.02 (0.92, 1.14) |
| 1e-5 | 2.2e-5 (1.0e-6, 5.4e-5) | 1.71 (1.56, 1.89) | 3.9e-6 (5.1e-7, 8.9e-6) | 1.02 (0.91, 1.15) |
| 3e-6 | 1.0e-5 (2.3e-7, 2.9e-5) | 1.70 (1.55, 1.89) | 1.2e-5 (6.1e-7, 3.2e-5) | 1.01 (0.91, 1.12) |
| 1e-6 | (not applicable) | 1.71 (1.55, 1.90) | (not applicable) | 1.02 (0.92, 1.14) |

| $\tau$ | $p_{\mathsf{dna}} \circ \mathsf{f}_{\mathrm{dna2aa}}$ ($n_{\max} = 15000$) | | $p_{\mathsf{dna}} \circ \mathsf{f}_{\mathrm{dna2aa}}$ ($n_{\max} = 20000$) | |
|---|---|---|---|---|
| | average JSD / byte | bytes / sec | average JSD / byte | bytes / sec |
| 1e-1 | 2.9e-2 (2.5e-2, 3.3e-2) | 9.52 (8.46, 10.82) | 2.9e-2 (2.5e-2, 3.3e-2) | 9.35 (8.21, 10.65) |
| 3e-2 | 3.2e-3 (2.8e-3, 3.5e-3) | 1.63 (1.37, 1.94) | 3.2e-3 (2.8e-3, 3.7e-3) | 1.41 (1.18, 1.71) |
| 1e-2 | 6.8e-4 (4.5e-4, 9.8e-4) | 1.10 (0.96, 1.27) | 8.3e-4 (5.0e-4, 1.3e-3) | 0.91 (0.78, 1.07) |
| 3e-3 | 2.2e-4 (8.2e-5, 4.2e-4) | 0.88 (0.77, 1.00) | 2.0e-4 (7.6e-5, 3.9e-4) | 0.71 (0.62, 0.83) |
| 1e-3 | 5.0e-5 (8.5e-6, 1.3e-4) | 0.80 (0.71, 0.91) | 6.6e-5 (1.7e-5, 1.5e-4) | 0.64 (0.56, 0.73) |
| 3e-4 | 2.8e-5 (3.6e-6, 7.2e-5) | 0.77 (0.68, 0.88) | 3.0e-5 (3.9e-6, 7.9e-5) | 0.61 (0.55, 0.71) |
| 1e-4 | 2.4e-5 (1.1e-6, 7.0e-5) | 0.75 (0.66, 0.84) | 6.0e-5 (1.8e-6, 1.6e-4) | 0.60 (0.54, 0.69) |
| 3e-5 | 6.0e-5 (1.0e-6, 1.6e-4) | 0.74 (0.66, 0.84) | 1.8e-4 (5.0e-5, 3.7e-4) | 0.60 (0.53, 0.68) |
| 1e-5 | 3.0e-7 (7.1e-9, 8.3e-7) | 0.74 (0.66, 0.84) | 7.8e-7 (5.3e-8, 2.1e-6) | 0.59 (0.53, 0.68) |
| 3e-6 | 9.3e-7 (1.1e-8, 2.7e-6) | 0.73 (0.65, 0.83) | 7.3e-7 (8.9e-9, 2.1e-6) | 0.59 (0.53, 0.67) |
| 1e-6 | (not applicable) | 0.74 (0.65, 0.84) | (not applicable) | 0.60 (0.53, 0.69) |

| $\tau$ | $p_{\mathsf{dna}} \circ \mathsf{f}_{\mathrm{dna2aa}}$ ($n_{\max} = 25000$) | | $p_{\mathsf{dna}} \circ \mathsf{f}_{\mathrm{dna2aa}}$ ($n_{\max} = 30000$) | |
|---|---|---|---|---|
| | average JSD / byte | bytes / sec | average JSD / byte | bytes / sec |
| 1e-1 | 2.9e-2 (2.5e-2, 3.3e-2) | 9.50 (8.31, 10.78) | 2.8e-2 (2.5e-2, 3.3e-2) | 9.52 (8.43, 10.99) |
| 3e-2 | 3.3e-3 (2.9e-3, 3.7e-3) | 1.28 (1.06, 1.60) | 3.3e-3 (2.9e-3, 3.8e-3) | 1.18 (0.96, 1.45) |
| 1e-2 | 7.9e-4 (4.9e-4, 1.3e-3) | 0.81 (0.70, 0.95) | 8.2e-4 (4.9e-4, 1.2e-3) | 0.72 (0.61, 0.86) |
| 3e-3 | 2.3e-4 (8.1e-5, 4.7e-4) | 0.62 (0.54, 0.72) | 2.5e-4 (7.9e-5, 4.8e-4) | 0.55 (0.47, 0.64) |
| 1e-3 | 5.9e-5 (1.1e-5, 1.4e-4) | 0.56 (0.49, 0.65) | 6.0e-5 (1.1e-5, 1.4e-4) | 0.49 (0.42, 0.57) |
| 3e-4 | 2.6e-5 (3.5e-6, 6.8e-5) | 0.53 (0.46, 0.61) | 3.4e-5 (4.9e-6, 8.5e-5) | 0.46 (0.41, 0.53) |
| 1e-4 | 2.4e-5 (8.2e-7, 6.9e-5) | 0.52 (0.46, 0.59) | 2.4e-5 (1.0e-6, 8.0e-5) | 0.45 (0.39, 0.51) |
| 3e-5 | 7.2e-5 (1.9e-5, 1.5e-4) | 0.51 (0.45, 0.59) | 2.4e-5 (5.6e-7, 7.1e-5) | 0.45 (0.39, 0.52) |
| 1e-5 | 1.9e-6 (5.6e-8, 4.6e-6) | 0.51 (0.45, 0.59) | 1.7e-7 (5.7e-9, 3.8e-7) | 0.44 (0.39, 0.51) |
| 3e-6 | 1.3e-6 (1.2e-8, 3.3e-6) | 0.51 (0.45, 0.58) | 1.6e-6 (1.7e-8, 3.7e-6) | 0.44 (0.38, 0.51) |
| 1e-6 | (not applicable) | 0.51 (0.45, 0.59) | (not applicable) | 0.44 (0.39, 0.51) |

## G.3 DECOMPOSITION SIZE

Tab. 11 and Fig. 18 show how the quotient and remainder sizes grow as $\tau$ decreases, explaining the throughput reduction observed in the JSD experiments.

For the all-IP-universal token-to-byte and DNA transducers, the remainder is empty ($|\mathsf{R}| = 0$) at every position, so only the quotient size is reported. In the token-to-byte setting, the mean quotient size grows from 62 at $\tau = $ 1e-1 to over 35,000 at $\tau = $ 1e-5, directly accounting for the throughput reduction at tight thresholds. The DNA transducer saturates early due to $n_{\max} = 5000$: mean $|\mathsf{Q}|$ reaches 465 by $\tau = $ 3e-5 and remains flat beyond that.

For $\mathsf{f}_{\mathrm{ptb}}$, which has non-IP-universal states, both $|\mathsf{Q}|$ and $|\mathsf{R}|$ are reported. $|\mathsf{Q}|$ grows steadily with a tighter $\tau$, while $|\mathsf{R}|$ initially decreases as more candidates enter the quotient but then grows as the expansion loop over non-IP-universal states discovers additional remainder elements. The presence of a non-trivial remainder is the key structural difference from the all-IP-universal transducers: each

remainder element requires a full string probability $p_{\mathcal{X}}(\boldsymbol{x})$ rather than just a prefix probability, making it more expensive per element.

Table 11: Mean and maximum quotient size $|\mathsf{Q}|$ across sequence positions for three transducers at varying pruning thresholds $\tau$. Results are computed on paragraph 1 of WikiText (833 bytes for $p_{\mathsf{gpt2}} \circ \mathsf{f}_\alpha$, 850 bytes for $p_{\mathsf{gpt2}} \circ \mathsf{f}_\alpha \circ \mathsf{f}_{\mathrm{ptb}}$) and on the longest protein in the test set (P83127, 12 amino acids for $p_{\mathsf{dna}} \circ \mathsf{f}_{\mathrm{dna2aa}}$). The $\mathsf{f}_\alpha$ and $\mathsf{f}_{\mathrm{dna2aa}}$ transducers have all-universal states ($|\mathsf{R}| = 0$ everywhere). The $\mathsf{f}_{\mathrm{ptb}}$ transducer has non-universal states, so we additionally report the remainder size $|\mathsf{R}|$.

<table>
<tr><td colspan="3">(a) $p_{\mathsf{gpt2}} \circ \mathsf{f}_\alpha$</td><td colspan="3">(b) $p_{\mathsf{dna}} \circ \mathsf{f}_{\mathrm{dna2aa}}$</td></tr>
<tr><td>$\tau$</td><td>Mean $|\mathsf{Q}|$</td><td>Max $|\mathsf{Q}|$</td><td>$\tau$</td><td>Mean $|\mathsf{Q}|$</td><td>Max $|\mathsf{Q}|$</td></tr>
<tr><td>1e-1</td><td>62</td><td>2,238</td><td>1e-1</td><td>104</td><td>344</td></tr>
<tr><td>3e-2</td><td>200</td><td>5,736</td><td>3e-2</td><td>288</td><td>1,274</td></tr>
<tr><td>1e-2</td><td>439</td><td>11,998</td><td>1e-2</td><td>346</td><td>1,329</td></tr>
<tr><td>3e-3</td><td>873</td><td>18,761</td><td>3e-3</td><td>439</td><td>2,011</td></tr>
<tr><td>1e-3</td><td>1,536</td><td>25,341</td><td>1e-3</td><td>454</td><td>2,027</td></tr>
<tr><td>3e-4</td><td>3,161</td><td>65,536</td><td>3e-4</td><td>462</td><td>2,044</td></tr>
<tr><td>1e-4</td><td>7,800</td><td>131,072</td><td>1e-4</td><td>463</td><td>2,044</td></tr>
<tr><td>3e-5</td><td>17,232</td><td>524,288</td><td>3e-5</td><td>465</td><td>2,046</td></tr>
<tr><td>1e-5</td><td>35,768</td><td>2,097,152</td><td>1e-5</td><td>465</td><td>2,046</td></tr>
<tr><td></td><td></td><td></td><td>3e-6</td><td>465</td><td>2,046</td></tr>
</table>

(c) $p_{\mathsf{gpt2}} \circ \mathsf{f}_\alpha \circ \mathsf{f}_{\mathrm{ptb}}$

| $\tau$ | Mean $|\mathsf{Q}|$ | Max $|\mathsf{Q}|$ | Mean $|\mathsf{R}|$ | Max $|\mathsf{R}|$ |
|---|---|---|---|---|
| 1e-1 | 1.3 | 31 | 1.0 | 113 |
| 3e-2 | 1.9 | 57 | 0.38 | 113 |
| 1e-2 | 3.6 | 142 | 0.48 | 25 |
| 3e-3 | 7.3 | 353 | 1.2 | 53 |
| 1e-3 | 16 | 686 | 3.5 | 129 |
| 3e-4 | 47 | 1,807 | 11.4 | 408 |
| 1e-4 | 128 | 5,101 | 33.9 | 1,195 |

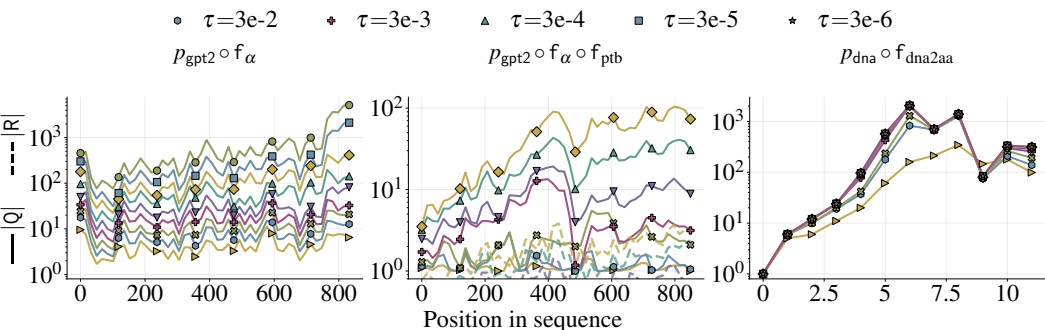

Figure 18: Growth of the quotient size $|\mathsf{Q}|$ (solid lines) and remainder size $|\mathsf{R}|$ (dashed lines) with sequence position for three transducer compositions at varying pruning thresholds $\tau$. Left: $p_{\mathsf{gpt2}} \circ \mathsf{f}_\alpha$ on paragraph 1 of WikiText (833 bytes). Center: $p_{\mathsf{gpt2}} \circ \mathsf{f}_\alpha \circ \mathsf{f}_{\mathrm{ptb}}$ on paragraph 1 (850 bytes after transduction); dashed lines show $|\mathsf{R}|$, which is non-zero because $\mathsf{f}_{\mathrm{ptb}}$ contains non-IP-universal states. Right: $p_{\mathsf{dna}} \circ \mathsf{f}_{\mathrm{dna2aa}}$ on the longest protein in the test set (P83127, 12 amino acids). For $\mathsf{f}_\alpha$ and $\mathsf{f}_{\mathrm{dna2aa}}$, all FST states are IP-universal, so $|\mathsf{R}| = 0$ everywhere. The $\mathsf{f}_{\mathrm{dna2aa}}$ panel uses a candidate-set cap of $n_{\mathrm{max}} = 5,000$, which limits the quotient size at lower thresholds and breaks monotonic growth at later positions.

### G.4 CROSS-ENTROPY

Cross-entropy measures how well the transduced model assigns probability to held-out text; unlike JSD, it only requires evaluating $\overrightarrow{p_{\mathcal{Y}}}(y \mid \boldsymbol{y})$ for the observed next symbol rather than computing the full next-symbol distribution $\overrightarrow{p_{\mathcal{Y}}}(\cdot \mid \boldsymbol{y})$. Tab. 12 reports cross-entropy (in nats and bits per byte) together with throughput for $(p_{\mathcal{X}} \circ f_{\alpha})$. Because only a single probability is needed per position, throughput is higher than in the JSD tables above. Cross-entropy converges quickly as $\tau$ decreases: for all four models, the tightest thresholds yield nearly identical values, confirming that moderate pruning suffices for accurate sequence scoring.

Table 12: Cross-entropy and throughput for various thresholds $\tau$ using $p_{\mathcal{X}} \circ f_{\alpha}$. 95% confidence intervals are given in parentheses. $p_{\text{gpt2}} \circ f_{\alpha}$ at $\tau = 1\text{e-}5$ is computed over 6 paragraphs; all other configurations use 10.

| $\tau$ | bytes / sec | | Bits / byte | | Cross-entropy | |
|---|---|---|---|---|---|---|
| | Mean | 95% CI | Mean | 95% CI | Mean | 95% CI |
| | | | $p_{\text{gpt2}} \circ f_{\alpha}$ | | | |
| 1e-1 | 82.61 | (73.33, 92.34) | 1.2434 | (1.1884, 1.2936) | 0.8619 | (0.8237, 0.8966) |
| 3e-2 | 49.10 | (44.48, 54.24) | 1.1466 | (1.0978, 1.1949) | 0.7948 | (0.7609, 0.8282) |
| 1e-2 | 28.46 | (26.73, 30.33) | 1.0652 | (1.0199, 1.1068) | 0.7383 | (0.7070, 0.7672) |
| 3e-3 | 14.20 | (13.47, 14.98) | 1.0385 | (0.9956, 1.0806) | 0.7199 | (0.6901, 0.7490) |
| 1e-3 | 6.46 | (6.08, 6.84) | 1.0293 | (0.9909, 1.0664) | 0.7134 | (0.6868, 0.7392) |
| 3e-4 | 2.18 | (2.04, 2.33) | 1.0220 | (0.9831, 1.0604) | 0.7084 | (0.6815, 0.7350) |
| 1e-4 | 0.75 | (0.70, 0.80) | 1.0220 | (0.9823, 1.0610) | 0.7084 | (0.6809, 0.7355) |
| 3e-5 | 0.35 | (0.33, 0.39) | 1.0200 | (0.9781, 1.0633) | 0.7070 | (0.6779, 0.7370) |
| 1e-5 | 0.34 | (0.30, 0.38) | 1.0025 | (0.9461, 1.0550) | 0.6949 | (0.6558, 0.7313) |
| | | | $p_{\text{llama1B}} \circ f_{\alpha}$ | | | |
| 1e-1 | 129.22 | (112.82, 146.19) | 0.9922 | (0.9467, 1.0405) | 0.6877 | (0.6562, 0.7212) |
| 3e-2 | 90.74 | (82.63, 99.33) | 0.9128 | (0.8682, 0.9542) | 0.6327 | (0.6018, 0.6614) |
| 1e-2 | 47.81 | (44.04, 51.52) | 0.8626 | (0.8252, 0.9027) | 0.5979 | (0.5720, 0.6257) |
| 3e-3 | 26.15 | (24.71, 27.73) | 0.8402 | (0.8037, 0.8790) | 0.5823 | (0.5571, 0.6093) |
| 1e-3 | 14.89 | (14.18, 15.69) | 0.8364 | (0.8001, 0.8725) | 0.5797 | (0.5546, 0.6048) |
| 3e-4 | 6.80 | (6.46, 7.18) | 0.8358 | (0.7999, 0.8724) | 0.5793 | (0.5545, 0.6047) |
| 1e-4 | 2.62 | (2.49, 2.77) | 0.8362 | (0.8018, 0.8699) | 0.5796 | (0.5558, 0.6030) |
| 3e-5 | 1.42 | (1.35, 1.49) | 0.8361 | (0.8002, 0.8717) | 0.5796 | (0.5547, 0.6042) |
| 1e-5 | 0.73 | (0.69, 0.77) | 0.8360 | (0.8002, 0.8717) | 0.5795 | (0.5546, 0.6042) |
| | | | $p_{\text{llama8B}} \circ f_{\alpha}$ | | | |
| 1e-1 | 88.65 | (79.18, 99.61) | 0.8176 | (0.7757, 0.8661) | 0.5667 | (0.5377, 0.6003) |
| 3e-2 | 76.39 | (71.27, 81.35) | 0.7392 | (0.7007, 0.7759) | 0.5124 | (0.4857, 0.5378) |
| 1e-2 | 52.05 | (49.41, 54.93) | 0.7153 | (0.6778, 0.7507) | 0.4958 | (0.4698, 0.5203) |
| 3e-3 | 28.81 | (27.18, 30.44) | 0.6959 | (0.6627, 0.7282) | 0.4824 | (0.4594, 0.5047) |
| 1e-3 | 16.86 | (16.01, 17.82) | 0.6913 | (0.6584, 0.7251) | 0.4791 | (0.4564, 0.5026) |
| 3e-4 | 9.73 | (9.24, 10.25) | 0.6870 | (0.6513, 0.7179) | 0.4762 | (0.4515, 0.4976) |
| 1e-4 | 4.35 | (4.15, 4.58) | 0.6870 | (0.6557, 0.7221) | 0.4762 | (0.4545, 0.5005) |
| 3e-5 | 2.60 | (2.47, 2.73) | 0.6869 | (0.6533, 0.7208) | 0.4761 | (0.4528, 0.4996) |
| 1e-5 | 1.57 | (1.49, 1.66) | 0.6869 | (0.6552, 0.7180) | 0.4761 | (0.4541, 0.4977) |
| | | | $p_{\text{phi4}} \circ f_{\alpha}$ | | | |
| 1e-1 | 42.39 | (37.95, 47.19) | 0.8714 | (0.8217, 0.9208) | 0.6040 | (0.5695, 0.6382) |
| 3e-2 | 34.48 | (31.73, 37.33) | 0.7748 | (0.7317, 0.8180) | 0.5371 | (0.5072, 0.5670) |
| 1e-2 | 24.89 | (23.05, 26.59) | 0.7547 | (0.7165, 0.7959) | 0.5231 | (0.4966, 0.5517) |
| 3e-3 | 16.55 | (15.34, 17.92) | 0.7301 | (0.6959, 0.7690) | 0.5061 | (0.4824, 0.5330) |
| 1e-3 | 9.91 | (9.41, 10.52) | 0.7241 | (0.6882, 0.7595) | 0.5019 | (0.4770, 0.5264) |
| 3e-4 | 5.81 | (5.51, 6.14) | 0.7170 | (0.6803, 0.7509) | 0.4970 | (0.4716, 0.5205) |
| 1e-4 | 2.55 | (2.41, 2.70) | 0.7163 | (0.6824, 0.7493) | 0.4965 | (0.4730, 0.5194) |
| 3e-5 | 1.54 | (1.46, 1.62) | 0.7162 | (0.6793, 0.7490) | 0.4965 | (0.4709, 0.5192) |
| 1e-5 | 0.86 | (0.81, 0.91) | 0.7162 | (0.6828, 0.7502) | 0.4964 | (0.4733, 0.5200) |

## G.5 BENCHMARKING THE COMPUTATIONAL SHORTCUT

We benchmark the computational shortcut inherent in the prefix decomposition described in §4, which allows us to decompose the precover into remainder and quotient representatives. We generate suboptimal decompositions by randomly selecting $n$ IP-universal states in the transducer (see §4) and treating them as non-IP-universal in our algorithms (see §5). This gradually decreases the size of the quotient and increases the remainder correspondingly. Tab. 13 shows the average JSD between the original and modified distributions over the first 256 bytes of the first 5 paragraphs of the `wikitext-2-raw-v1` dataset (Merity et al., 2017). For each value of $n$, we repeat the sampling of new non-universal states three times, and report the mean JSD.

The distributions diverge rapidly, and after converting roughly 15–20% of IP-universal states, the algorithm repeatedly encounters dead ends. This number varies between runs; as shown in Fig. 19, each repeat exhibits a distinct step function: JSD remains low until a particular high-connectivity state is converted, then jumps by 2–3 orders of magnitude and plateaus. In the PTB transducer, one IP-universal state (state 193) serves as a routing hub with 274 arcs spanning 245 output symbols. When this state is converted, frontiers containing it move from the quotient Q to the remainder R, causing the expansion loop to require fallback scoring (Fig. 10). Different runs encounter this hub at different values depending on the permutation order, producing the staircase pattern. This shows that IP-universal states have a hierarchical importance structure: distribution quality is governed by a small number of high-connectivity hub states, while the majority can be converted with negligible impact (JSD $\leq$ 1e-4).

Table 13: Average Jensen–Shannon divergence (JSD) for the PTB transducer ($\tau$ =1e-3) after randomly converting $n$ of the IP-universal states to non-IP-universal. 95% confidence intervals are given in parentheses. JSD is computed against the unmodified transducer ($n = 0$).

| Converted States ($n$) | $p_{\mathsf{llama1B}} \circ f_\alpha \circ f_{\mathrm{ptb}}$ | $p_{\mathsf{llama8B}} \circ f_\alpha \circ f_{\mathrm{ptb}}$ | $p_{\mathsf{phi4}} \circ f_\alpha \circ f_{\mathrm{ptb}}$ |
|---|---|---|---|
| 0 | (not applicable) | (not applicable) | (not applicable) |
| 2 | 3.9e-5 (3.5e-5, 4.3e-5) | 1.3e-4 (7.4e-5, 2.1e-4) | 6.6e-5 (3.1e-5, 1.1e-4) |
| 5 | 9.5e-4 (7.5e-4, 1.2e-3) | 9.0e-4 (7.5e-4, 1.1e-3) | 7.8e-4 (6.4e-4, 9.4e-4) |
| 8 | 9.8e-4 (8.0e-4, 1.2e-3) | 9.2e-4 (7.7e-4, 1.1e-3) | 8.1e-4 (6.6e-4, 9.6e-4) |
| 11 | 3.0e-2 (2.6e-2, 3.3e-2) | 2.8e-2 (2.5e-2, 3.1e-2) | 2.6e-2 (2.3e-2, 2.9e-2) |
| 14 | 2.9e-2 (2.6e-2, 3.3e-2) | 2.8e-2 (2.5e-2, 3.1e-2) | 2.6e-2 (2.3e-2, 2.9e-2) |
| 16 | 2.9e-2 (2.6e-2, 3.2e-2) | 2.8e-2 (2.5e-2, 3.2e-2) | 2.6e-2 (2.3e-2, 2.9e-2) |
| 19 | 2.9e-2 (2.6e-2, 3.3e-2) | 2.8e-2 (2.5e-2, 3.2e-2) | 2.6e-2 (2.3e-2, 2.9e-2) |

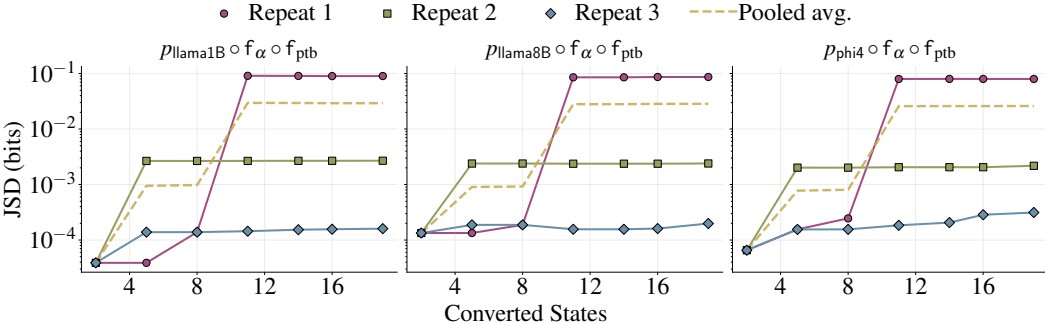

Figure 19: Per-repeat Jensen–Shannon divergence (JSD) vs. number of states converted from IP-universal to non-IP-universal for the PTB transducer ($\tau$ =1e-3). Each repeat uses a different random conversion order. The abrupt jumps are caused by individual high-connectivity states, whose conversion causes a jump in JSD.

## H  STATE-BASED DECOMPOSITION

The algorithms presented in the main text enumerate source *strings*: the quotient $\mathcal{Q}(\boldsymbol{y})$ and remainder $\mathcal{R}(\boldsymbol{y})$ are represented as explicit sets of strings. When these sets are infinite, the string-enumeration approach does not terminate without pruning.

An alternative is to operate on the *state space* of the precover DFA rather than its string space. Since the precover DFA $\mathsf{P}_{\boldsymbol{y}}$ (§5.1) has finitely many states, a state-based traversal always terminates, even when $\mathcal{Q}(\boldsymbol{y})$ or $\mathcal{R}(\boldsymbol{y})$ are infinite. The key idea is to represent the quotient and remainder as DFAs $\mathsf{Q}$ and $\mathsf{R}$—automata that *accept* the (possibly infinite) sets $\mathcal{Q}(\boldsymbol{y})$ and $\mathcal{R}(\boldsymbol{y})$ respectively—rather than enumerating their elements. The full precover is then recovered as $\mathcal{P}(\boldsymbol{y}) = \langle [\![\mathsf{Q}]\!] \rangle \sqcup [\![\mathsf{R}]\!]$.

The algorithm in Fig. 20 performs a BFS over the states of $\mathsf{P}_{\boldsymbol{y}}$. At each state $s$, it checks whether $s$ is *universal*—i.e., whether the sub-automaton rooted at $s$ accepts $\mathcal{X}^*$ (see `is_cylinder`). If so, $s$ is marked as a quotient member and accepting state, and its successors are not explored, since all extensions from a universal state are covered. If $s$ is accepting but not universal, it is marked as a remainder member and accepting state, and exploration continues through its outgoing arcs.

The result is a pair of DFAs that share the same state space and truncated arc set but differ in their accepting states. The quotient automaton $\mathsf{Q}$ has the universal states as accepting states, and arcs leaving universal states are dropped (since the BFS does not expand past them); thus $[\![\mathsf{Q}]\!] = \mathcal{Q}(\boldsymbol{y})$, accepting exactly the quotient elements. The remainder automaton $\mathsf{R}$ uses the same truncated arc set but marks only the non-universal accepting states; thus, $[\![\mathsf{R}]\!] = \mathcal{R}(\boldsymbol{y})$.

```
247  def dfa_decomposition(f, y):
248     P_y ← trim(determinize(proj_X(f ∘ yY*)))
249     q ← QUEUE(I_y)
250     V ← ∅; arcs ← ∅
251     Q ← ∅; R ← ∅
252     while q:
253        s ← q.pop()
254        if s ∈ V: continue
255        V.add(s)
256        if s ∈ F_y:
257           if is_cylinder(s):
258              Q.add(s)
259              continue  # do not expand
260           else:
261              R.add(s)
262        for x ∈ X:
263           s' ← step_y(s, x)
264           if s' = ∅: continue
265           q.push(s')
266           arcs.add(s →ˣ s')
267     return DFA(V, arcs, Q), DFA(V, arcs, R)

268  def is_cylinder(s):
269     V ← {s}; q ← QUEUE({s})
270     while q:
271        s ← q.pop()
272        if s ∉ F_y:
273           return False
274        for x ∈ X:
275           s' ← step_y(s, x)
276           if s' = ∅: return False
277           if s' ∉ V:
278              V.add(s'); q.push(s')
279     return True
```

Figure 20: State-based decomposition of the precover into DFAs. *Left*: BFS over the precover DFA $\mathsf{P}_{\boldsymbol{y}}$. Universal states become accepting in $\mathsf{Q}$; non-universal accepting states become accepting in $\mathsf{R}$. Arcs leaving universal states are not collected. *Right*: `is_cylinder` is the same universality BFS as in Fig. 3, but takes a DFA state directly rather than computing it from a source string via `run`$_{\boldsymbol{y}}$.

**Comparison with string-based algorithms.**  The state-based algorithm always terminates in finite time—even when $\mathcal{Q}(\boldsymbol{y})$ or $\mathcal{R}(\boldsymbol{y})$ are infinite—because $\mathsf{P}_{\boldsymbol{y}}$ has finitely many states. The resulting DFAs $\mathsf{Q}$ and $\mathsf{R}$ provide compact, finite representations of these potentially infinite sets.

However, implementing the autoregressive interface (§C.1) still requires enumerating the strings accepted by these machines: computing $\overrightarrow{p_{\mathcal{Y}}}(\boldsymbol{y})$ via Eq. (8a) sums $p_{\mathcal{X}}(\boldsymbol{x})$ over elements of $\mathcal{Q}(\boldsymbol{y})$ and $\mathcal{R}(\boldsymbol{y})$, and when these sets are infinite, the sum must be truncated regardless. Thus, while the DFA representation guarantees a finite decomposition, it does not, on its own, yield finite-time scoring.

# I   RELATED WORK

Modern language models define probability distributions over sequences of tokens (see §2). For efficiency and vocabulary (a.k.a. their alphabet) management, they usually rely on subword schemes such as BPE (Sennrich et al., 2016; Gage, 1994) or Unigram (Kudo, 2018). Although these approaches have been remarkably successful, their units often don't coincide with linguistic boundaries, and any given string typically admits an exponential number of tokenization variants with non-zero probability mass under the language model. Recent work has tackled this issue by enforcing canonical tokenization to remove probability mass from noncanonical encodings (Vieira et al., 2025b), while Geh et al. (2024) have shown that aggregating the probability mass of noncanonical tokenization choices carries a useful signal that can boost downstream accuracy.

Subword segmentation also gives rise to the prompt-boundary problem (Vieira et al., 2025a), where imperceptible changes to the final characters of a prompt (e.g., appending a single whitespace) can push the encoded token sequence onto a completely different path in token space, causing the model to abandon otherwise highly probable continuations. To overcome these issues, Vieira et al. (2025a) introduce an algorithm for transforming token-based language models into language models over characters. Although their contribution centers around characters, the underlying idea can be generalized (as shown in this work).

Many applications need a method to accurately convert the probability mass learned on subword tokens to other types of units, such as bytes, words, or morphemes in NLP, or amino acids in computational biology, as pointed out in §1. An additional example is found in psycholinguistics, where researchers often require fine-grained estimates of surprisal, e.g., to predict a reader's likelihood of skipping a word based on how predictable its first three characters are (Rayner et al., 1982; Blanchard et al., 1989). To this end, recent studies have tackled the challenges posed by subword tokenization (Nair & Resnik, 2023; Beinborn & Pinter, 2023; Pimentel & Meister, 2024; Oh & Schuler, 2024; Giulianelli et al., 2024). For example, Oh & Schuler (2024) and Pimentel & Meister (2024) argue that leading whitespace tokenization introduces a confounder in surprisal estimates and instead advocate for incorporating the probability of trailing whitespaces into such calculations.

Furthermore, Pimentel & Meister (2024) give a bespoke procedure for converting token-based language models to word-based language models. However, their method does not model the contextually sensitive nature of English word segmentation, e.g., it treats both periods in Ex. (1) identically, where English orthography does not. Additionally, the justification of the procedure requires that there exists a set of distinguished end-of-word markers that appear at the end of a token, if at all. We now consider how such a transducer can be constructed. Let $f_\alpha$ be a transducer that converts a token alphabet to a character alphabet, and $D$ be the set of delimiters. The transducer $f_D$ is given in Fig. 21. Given a language model $p_\mathcal{X}$ over $\mathcal{X}$, we can then compose them into a transducer $p_\mathcal{X} \circ f_\alpha \circ f_D$ to get a transduced language model over separator-delimited words. However, such an approach would be rather naïve. Unfortunately, delimiter-based separation would not be able to distinguish when the dot should be its own symbol or not, as in ex. (3).

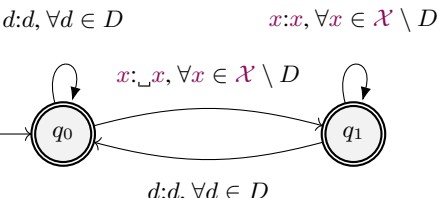

Figure 21: A simple FST that segments character streams into words without contextual information, inserting ␣ at the start of each word.

The delimiter-based approach also fails for most BPE-based language models because of the clustering of delimiter candidates. For instance, GPT-4o's alphabet contains the token $_{10880}$, which consists

solely of end-of-word symbols. Under PTB guidelines, for example, $_{10880}$ should be broken into three consecutive orthographic words ! ! ! . In contrast, we argue that the proper tokenization scheme for psycholinguistic modeling should be specified based on the goals of the study and not based on the properties of any one specific tokenizer.

Tokenization challenges are not unique to modeling natural language. In computational biology, DNA, RNA, and protein sequences are long, unsegmented strings over small alphabets that pose challenges in tokenization. Researchers thus alternate between different tokenization schemas, such as k-mers, learned subwords, and motif-aware segmenters (Ji et al., 2021; Nguyen et al., 2023; Dotan et al., 2024; Wang et al., 2024; Qiao et al., 2024). Because language models are often trained under different tokenization schemas, their autoregressive predictions—step-by-step probabilities and conditional distributions—are not directly comparable across tokenization schemes.

Transducer-based approaches to tokenization are well-established—WordPiece (Wu et al., 2016) can be implemented as a transducer (Song et al., 2021), and deterministic finite automata have been constructed for BPE (Berglund & van der Merwe, 2023; Berglund et al., 2024). Moreover, transducers also have a long history in language modeling (Mohri, 1997) and have been adopted for constrained decoding, where an FST enforces lexical or structural constraints (Allauzen et al., 2014; Ghazvininejad et al., 2016; Stahlberg et al., 2019; Willard & Louf, 2023; Koo et al., 2024; Cognetta et al., 2025).

Closely related are neural finite-state transducers (Lin et al., 2019) and earlier neural FST hybrids (Rastogi et al., 2016), which assign (neural) weights to transitions and compute prefix/sequence probabilities as path-sums over all paths consistent with a given output. Our construction admits a similar view: we marginalize over all source strings (paths) that map to a given output. However, rather than learning transition weights, we focus on transducing an off-the-shelf pretrained LM, enabling efficient inference via quotient-remainder decomposition.

In this study, we generalize character-level conversion and extend Vieira et al. (2025a) into a framework that enables transforming a language model into another language model, beyond the limited setting of strict-prefix monotonic mappings. We support conversions between sets of units and unit-preserving transformations, provided that the mapping between them can be described by a finite-state transducer.

## J    LIMITATIONS

**Empirical scope.** Although our framework theoretically enables transduction of any language model to any unit of interest, given a valid transducer, we test only a limited set of architectures (GPT-2, LLaMA 3, and Phi-4) and target specific units (bytes, Penn Treebank tokens, and amino acids). Future research could broaden the analysis to a wider range of models, datasets, and units.

**Expressiveness.** Our analysis has focused on functional finite-state transducers and regular languages. Future work could consider dynamically built transducers and distributions over more expressive languages, as well as stochastic maps encoded by non-functional transducers—where the notion of universality would need to be adjusted.

**Approximation quality.** Our pruning-based inference algorithm performs well when the source language model concentrates most of its probability mass on a small number of prefixes that map to the current target prefix under the transducer—i.e., when the effective size of the quotient and remainder after pruning is small. When the mass is dispersed across many source prefixes (such as in the DNA to amino-acid case), pruning must discard a larger fraction of the total probability, and the approximation degrades. An importance-sampling approach (§ C.3) could provide a less systematically biased alternative.

**Speed.** The speed of our algorithms and implementations may not suit every use case. Obtaining the full distribution (for example, for decoding or model comparisons) currently requires speeds of around 10–20 bytes/sec. While this is not prohibitive for many tasks, it leaves ample room for improvement.

**Marginalization.** A core property of transduced language models is the *marginalization*: the transduction sums source-string probabilities to compute target-string probabilities, aggregating mass across all source strings that map to the same target. Since natural language allows the same meaning to be expressed in multiple ways, a transduction could normalize these to obtain a more representative output distribution over the space of interest. Consider, for instance, a math problem: "If 12 students are younger than Tom and 12 students are older, how many are there in total?" The answer could be '25', or 'twenty-five', or perhaps '12+1+12'. If any of these have significant probability mass, we would like to sum them. In chain-of-thought reasoning, we can similarly imagine summing and normalizing probabilities across samples from the system, an approach known as *self-consistency* (Wang et al., 2023). We hope to see transduced language models applied in such settings in future work.

