# OpenReview forum: "Transducing Language Models"
_ICLR.cc/2026/Conference — ICLR 2026 Poster_

### Official Review · Reviewer_Sa4U · 2025-10-30

**Soundness:** 2
**Presentation:** 3
**Contribution:** 3
**Rating:** 6
**Confidence:** 4

**Summary:**

This work presents a general framework for transforming a language model over source strings into an LM over target strings, focusing on FST transformations. A major technical contribution is prefix decomposition of the preimage of a target prefix into a quotient and a remainder (Eq 6), which allows discussion of finite / infinite cases of quotient / remainder sets.

This paper also discusses exact and approximate algorithms for building the two sets.

As for experiments, this paper discusses 3 LM transduction applications: 1) token to character 2) token to word 3) DNA to Amino Acid.

**Strengths:**

- Clean formalization: the remainder-quotient decomposition allows quite general discussion of exact / approximate algorithms.
- Makes algorithmic contributions.
- Interesting applications that show effectiveness of transduced LMs.

**Weaknesses:**

- The formalism / accompanying algorithms / experiments seem to be designed specifically with FSTs in mind, which restricts its applicability. For example, Fig 3 (left) does not discuss the decidability of membership test in L10 and L13, which can actually be nontrivial in real LM usages (for example suppose you are trying to transform a code LM to a code-output LM). Decidability and complexity aspects should be discussed in a more rigorous fashion for the general case.
- Limiting ourselves to the FST-as-transformation special case, the paper would benefit from a more extensive comparison to existing NLP approaches that compose LM outputs with classical formalisms, such as [neural finite-state transducers](https://aclanthology.org/N19-1024/): the proposed method appears to be a special case of the neural finite-state transducers where the prefix probability can be seen as a pathsum of some machine that transduces the precover of y to y.
- Analysis of the pruning algorithm discussed in G.4 would strengthen the paper: e.g., can you bound the error?
- The paper would also be strengthened if it could discuss learning scenarios, for example if the source LM is parametric, what are the gradients of next-token probabilities under $p_Y$ ?

**Questions:**

- Can you provide error bounds for the approximate algorithms?
- Measure-theoretic foundations are implicit: could you spell out the underlying probability space / measurability assumptions of $p_Y$?

---

> ### Author Response · Authors · 2025-11-25
>
> Thanks for the careful review. We respond to your comments below.
>
> ### W1: The work seems to be designed specifically with FSTs in mind. For example, Fig 3 (left) does not discuss the decidability of membership test in L10 and L13, which can actually be nontrivial in real LM usages (for example, suppose you are trying to transform a code LM to a code-output LM). Decidability and complexity aspects should be discussed in a more rigorous fashion for the general case.
>
> Only a part of the formalism applies to FSTs. The abstract algorithm shown on the left in Figure 3 assumes some way to check for “Continuity”, “Discontinuity”, and “Candidacy”. These are set-theoretic ideas. The general decidability and complexity of these is not possible, as membership in a set is not decideable (consider e.g. turing machines that halt). In this work we give FST-based algorithms that show exactly how to check membership and the complexity of doing so is linear over the length of the prefix as we can construct machines that recognize the quotient and remainder respectively. In other words, the left algorithm in Figure 3 is an abstract mathematical algorithm; to the right of it, we make it concrete for the FST showing it is decideable. We do foresee extending the notions of the quotient and remainder to more expressive language classes in future work however. But this will be non-trivial, for CFGs for instance, the universality check is not decideable (see Hendrik Jan Hoogeboom. 2015. Undecidable problems for context-free grammars) and would need to be adapted to a “non-optimal” decomposition. Does this answer your question?
>
> ### W2: The proposed method appears to be a special case of the neural finite-state transducers where the prefix probability can be seen as a pathsum of some machine that transduces the precover of y to y.
>
> We do not see our work as a special case of NFSTs in which the weights are neural. Instead, we combine any pre-trained language model with a transduce, whereas neural FST uses a neural model applied at each transition to predict the weight. Both are language models and constrained to FSTs, but the methods and configurations involved are very different. We can take any off-the-shelf LM and transform it, this is not something NFSTs do.
>
>
> ### W4: The paper would also be strengthened if it could discuss learning scenarios, for example if the source LM is parametric, what are the gradients of next-token probabilities under p_y ?
>
> The idea of fine-tuning the source LM (is this what you meant?) through transduction is really interesting and is something we may consider in future work. If pruning is not used, then we foresee the gradients can be calculated since the probability of the target sequence is a sum of source model probabilities. The approximation using pruning would however, make it non-differentiable, so this would not be trivial.
>
>
> ### W3/Q1:Can you provide error bounds for the approximate algorithms?
>
> We keep at least $1-\tau$ proportion of mass in each step where $\tau$ is a threshold parameter for how much mass is at most dropped.
> This means that we, in $T$ input steps, at least maintain $(1-\tau)^T$ mass, and an upper bound on the mass lost is $1-(1-\tau)^T$. For example, at $T=1024,\tau=1e-6$ we can at most drop $(1-0.000001)^{1024}=0.1$% of the total mass.
> However, this global bound does not guarantee that the accuracy of low-probability sequences where $p(y)<<\tau$.
> We point to our experimental results (e.g. Fig 6) that demonstrate the convergence of the approximate algorithm as we lower $\tau$ and its proximity (low JSD) to that in the prior work (see e.g., Table 6).
> We will expand the limitations section to discuss what can happen in cases where the probability mass is too evenly spread.
>
> ### Q2: Measure-theoretic foundations are implicit: could you spell out the underlying probability space / measurability assumptions of p_y?
>
> Regarding the measure-theoretic foundations: We assume the standard measurable space for the discrete countable sets $X^* $ and $Y^* $, where the $\sigma$-algebra is the power set. Under this assumption, any map $f: X^* \to Y^* $ is measurable. The transduced language model then implements a **push-forward measure**: $p_{Y}(y) = p_{X}(f^{-1}(y))$ (see, e.g., https://en.wikipedia.org/wiki/Pushforward_measure) where $f$ is the (rational) function that the transducer encodes so $p_{Y}(y) = p_{X}(f^{-1}(y)) = \sum_{x\in f^{-1}{y}}p_{X}(x)$. To see why $p_{Y}$ is a well-defined probability measure on $\mathrm{Im}(f)$ (and all of $Y$ by putting zero mass on the complement), note that since $f$ is functional that it partitions the countable domain $X^* $ into disjoint sets for each unique output. We thus have $\sum_{y\in\mathrm{Im}(f)}p_{Y}(y)=\sum_{x\in X^*}p_{X}(x)=1$.

---

> > ### Comment · Reviewer_Sa4U · 2025-11-28
> >
> > Thank you for the detailed response! I have some remaining concerns regarding the scope of the claims and the relationship to prior methods.
> >
> > ## Regarding W1 (General Scope)
> >
> > Thank you for the clarification, and the CFG counter example. If the framework relies on membership tests that are computationally intractable or undecidable outside of FSTs (Rational Relations), the claim of generality (_e.g._, L17, L95-99) seems a bit too strong.
> >
> > Suggestions: either
> > 1. Formalize the conditions under which Fig 3 left can terminate in finite time, and discuss the necessity / sufficiency of these conditions (maybe using FSMs / CFGs as illustrative examples?) Or
> > 2. Explicitly narrowing the paper's claims (Title, Abstract, Introduction) to focus on FST-based transduction.
> >
> > ## Regarding W2 (Relation to existing NLP work)
> >
> > IIUC, the prefix probability in your work (Eq. 3) is isomorphic to the weighted pathsum (partition function) of an NFST. The reduction is as follows:
> >
> > Given a target prefix $y$, a source LM $p_{\mathcal{X}}$, and a transformation FST $T$:
> >
> > 1. Construct the composition $T' = T \circ \langle y \rangle$.
> > 2. Define a scoring function $G$ that assigns weights to the transitions of $T'$ based on the log-probabilities of the input projections under $p_{\mathcal{X}}$.
> > 3. The value $\vec{p_{\mathcal{Y}}}(y)$ is exactly the partition function of the weighted machine $(T', G)$.
> >
> > The primary novelty of this work lies in the inference method (deterministic pruning via precover decomposition) rather than the model definition. Previous work tackling these intractable sums (e.g., Lin _et al._, 2019 discussed above) often utilized Monte Carlo (MC) methods.
> >
> > To further strengthen the paper, could you compare your pruning-based approach against a Monte Carlo baseline?
> >
> > ## Regarding W3 (error bound)
> >
> > Thank you for the clarification. Since JSD measures source / target divergences against source+target mixtures, errors of really low probability sequences may be smoothed out (by the other distribution). Would it be possible to get both KL divergence numbers?

---

> > > ### Author Response · Authors · 2025-12-01
> > >
> > > We thank the author for engaging with our response.
> > >
> > >
> > > > Thank you for the clarification, and the CFG counter example. If the framework relies on membership tests that are computationally intractable or undecidable outside of FSTs (Rational Relations), the claim of generality (e.g., L17, L95-99) seems a bit too strong.
> > >
> > > In lines 18-22, we mention that the generality is for *FSTs*.  To remove any doubt, we will modify the sentence in the abstract to ensure generality and *FST* occur closer together.
> > >
> > > In lines 95-99 we do explicitly say the composition is with transducers. In our experience, the term transducer is conventionally taken to refer to the finite-state case rather than e.g., a pushdown transducer or Turing machine. To remove any doubt, we will add “finite-state” there.
> > >
> > > >Explicitly narrowing the paper's claims (Title, Abstract, Introduction) to focus on FST-based transduction.
> > >
> > > As pointed out above, the FSTs are mentioned in both the abstract and the introduction, and we will clarify the two instances mentioned. The name of the paper is *Transducing Language Models*. While *Finite-State Transduction of Language Models* would be more explicit, we believe the general association between finite-state transducers without stacks or tapes and the term transducers warrants using the less unwieldy naming. In fact, finite-state and FST are mentioned almost 40 times in the paper.
> > >
> > >  > Formalize the conditions under which Fig 3 left can terminate in finite time, and discuss the necessity / sufficiency of these conditions (maybe using FSMs / CFGs as illustrative examples?)
> > >
> > > Lemma 4.1 makes a claim about sufficient conditions for finite quotients and remainders for (finite state) transduced language models (i.e., the case on the right in Figure 3). The algorithms that enumerate the quotient and remainder will thus finish in finite time, and the non-approximate ones will give the exact precover decomposition. The necessary conditions are stricter, and we believe the proof is much more complicated; this is something we will consider for future work. There are no general conditions for the abstract set-based algorithm in Figure 3, as it checks for set membership. As we point out in the prior work, it is not possible to check, e.g., the universality for CFGs.
> > >
> > > > IIUC, the prefix probability in your work (Eq. 3) is isomorphic to the weighted pathsum (partition function) of an NFST. The reduction is as follows: Given a target prefix $y$, a source LM $p_X$ , and a transformation FST $T$: 1. Construct the composition $T’=T\circ \langle y \rangle$. 2. Define a scoring function $G$ that assigns weights to the transitions of $T’$ based on the log-probabilities of the input projections under $p_X$. 3. The value $\overrightarrow{p_Y}(y)$ is exactly the partition function of the weighted machine $(T’, G)$
> > >
> > > After careful consideration we agree that such a construction can be made. Thank you so much for bringing it to our attention and for the translation of our problem set up into the language of the NFST paper. We will definitely explain this and give appropriate credit in the next version of the paper. For this to work, however, we add that we have to be careful to use an input-deterministic encoding of $T'$ to avoid the possibility that more than one path in $T'$ could generate a given source string $\boldsymbol{x}$.
> > >
> > >
> > > That said, we believe that our focus on functions (rather than relations) and pretrained language models (rather than general neural scoring functions) is an interesting aspect of the NFST landscape to narrow in on. As you can see, we had to perform some deep investigations into the special structure that some mappings $f$ have, which gave rise to our theory of decompositions, which are a common structure in the mappings that motivated this work. We also think the intended uses for TLMs are quite different from those that the NSFT authors seem focused on which are largely focused on NFSTs as a rich family of probabilistic models with potentially useful inductive bias, i.e., the structure of the transducer is essential to how the model is parameterized, but in our work the structure of the transducer is not important and we are free to transform the transducer by applying semantics-preserving transformations to it, such as input-determinization. **This is a key distinction: the object of study in our work is the transduced language model, irrespective of how it is encoded. While the NSFTs concern themselves with a given FST.**

---

> > > > ### Author Response · Authors · 2025-12-01
> > > >
> > > > > The primary novelty of this work lies in the inference method (deterministic pruning via precover decomposition) rather than the model definition. Previous work tackling these intractable sums (e.g., Lin et al., 2019 discussed above) often utilized Monte Carlo (MC) methods. To further strengthen the paper, could you compare your pruning-based approach against a Monte Carlo baseline?
> > > >
> > > > Tables 1 and 10 demonstrate what would happen with an MC estimate. Instead of using the computational shortcut from the precover decomposition, we convert quotient members (representing cylinder sets) to remainder elements (single values corresponding to their bases). The experiments show the resulting distributions diverge quickly.
> > > >
> > > > > Thank you for the clarification. Since JSD measures source / target divergences against source+target mixtures, errors of really low probability sequences may be smoothed out (by the other distribution). Would it be possible to get both KL divergence numbers?
> > > >
> > > > You are correct that there may be implicit smoothing involved in the JSD. Since we are approximating the distribution we sometimes lack support for very low-frequency symbols. In such cases, we get infinite KL. To be able to calculate the KL, we thus apply smoothing. With $\epsilon=1e-12$, we see the same trends as for the JSD https://imgur.com/a/0VySlP7 where $p$ is the approximation distribution and $q$ the reference with lowest threshold.

---

### Official Review · Reviewer_9rp9 · 2025-10-30

**Soundness:** 3
**Presentation:** 3
**Contribution:** 3
**Rating:** 8
**Confidence:** 3

**Summary:**

The paper studies how to compute or approximate target-prefix probabilities when a language model over one unit type (e.g., tokens, bytes, DNA) is passed through a deterministic mapping (finite-state transducer) into another unit type (e.g., characters, words, amino acids). Because the mapping can merge, split, or be context-dependent, a target prefix corresponds to a complex set of source prefixes.

Vieira et al. (2025a) handle the special case of strict-prefix-monotone mappings (e.g., tokens→characters), where target-prefix probabilities reduce to sums over source-prefix probabilities. This paper generalizes to arbitrary deterministic FSTs via a precover decomposition: split the relevant source prefixes into a quotient (prefixes whose outputs already force the target prefix) and a remainder (prefixes that must be summed explicitly).

For practicality, the authors add lazy determinization, memoization/precomputation universal states, and probability-mass pruning with a threshold $\tau$ (optionally with candidate-set caps and backtracking).

**Strengths:**

Clear, principled generalization of the strict-monotone case with an identity that is easy to implement (quotient + remainder). Practical algorithms with sensible speedups and demonstrated accuracy–speed trade-offs on non-trivial mappings. Overall the definitions, explanations, and conditions are crisp and easy to follow.

**Weaknesses:**

Key algorithms and experimental details are mostly in the appendix, and the approximations in 5.2 are described very briefly. Readers may miss core contributions if they don’t consult the appendix. Just a suggestion, feel free to ignore if it doesn't make sense: bring a minimal algorithm box/flowchart and a small table that lists each approximation knob (lazy determinization, memoization, pruning, caps) with default settings into the main text.

The paper varies $\tau$ but does not isolate the contribution of each approximation, nor does it provide a direct runtime/memory comparison against Vieira in the shared setting.

**Questions:**

Could you toggle, one at a time, (a) lazy determinization, (b) memoization/precompute, (c) composition strategy, (d) adaptive $\tau$  candidate-set caps, and report the effect on JSD/accuracy, throughput, and peak memory ?

On the shared tokens→characters setup, can you report a matched comparison (same model, dataset, hardware) with throughput, wall-clock, and peak memory with Vieira2025a ?

---

> ### Author Response · Authors · 2025-11-25
>
> Thank you for the thoughtful review. We are happy to read that you found the paper easy to follow and our approach practical.
>
> ### W1: Key algorithms and experimental details are mostly in the appendix, and the approximations in 5.2 are described very briefly. Readers may miss core contributions if they don’t consult the appendix. Suggestion: bring a minimal algorithm box/flowchart and a small table that lists each approximation knob (lazy determinization, memoization, pruning, caps) with default settings into the main text.
>
>
> It was a choice between balancing the introduction to the method with the details needed to speed it up. Since the algorithmic ´tricks’ needed are standard (memoization, recursion, lazy determinization, dynamic programming, etc.), we prioritized highlighting our specific contribution rather than the more classic engineering solutions. In fact, without the optimizations the method would often be painfully slow. We will think more about this. Your suggestion to introduce a high-level flow chart into the main text is excellent (thank you!).
>
> ### Q1: Could you toggle, one at a time, (a) lazy determinization, (b) memoization/precompute, (c) composition strategy, (d) adaptive candidate-set caps, and report the effect on JSD/accuracy, throughput, and peak memory ?
>
> Regarding the isolation of each algorithmic decision, our goal in building up the complexity of the algorithm is more pedagogical/constructive rather than claiming the steps taken are fully optional. For instance, eager determinization can result in an exponentially larger machine, making it infeasible to materialize (e.g., in the byte-level transformation in composition with the PTB transducer, this was completely infeasible). Lazy determinization avoids the exponential blow-up, by only constructing determinized states on demand, such that most of the potentially huge machine is never materialized. Likewise, omitting memoization and thus not reusing the decomposition for prior prefixes would similarly be prohibitively detrimental to speed. If you think that providing specific numbers for these ablations would substantially strengthen the paper, we will consider it; however, these would be largely unusably slow. Regardless of whether specific ablations are included, we will clarify how essential these techniques are in the manuscript.
>
> ### Q2: On the shared tokens→characters setup, can you report a matched comparison (same model, dataset, hardware) with throughput, wall-clock, and peak memory with Vieira2025a ?
>
> We provide the requested throughput/wall-time/memory comparison to Viera2025a (GenLM-bytes) below. Note, however, that the $K$ and Thresholds ($\tau$ in the paper) are for different algorithms, so we are not comparing apples-to-apples in terms of the efficiency of the algorithms. We note that while memory can peak for the lowest $\tau$ as mass may be very spread out, that memory usage is otherwise comparable. The GenLM-bytes speed is also a bit faster but only by a small factor for the lower thresholds and higher beam sizes.
>
>
> #### GPT-2 Large
>
> **Transduced LM:**
>
> | Threshold | Wall-clock time (s) | Throughput & 95% CI (bytes/s) | Peak GPU mem (GB) | Peak CPU mem (GB) |
> |-----------|---------------------|--------------------------------|--------------------|--------------------|
> | 0.1 | 220.15  | 34.90 [24.40–49.68]   | 17.24  | 13.99  |
> | 0.03   | 242.98  | 31.62 [19.56–51.58]   | 17.24  | 21.32  |
> | 0.01   | 200.71  | 38.29 [23.45–61.17]   | 17.23  | 13.94  |
> | 0.003  | 164.31  | 46.76 [32.31–70.65]   | 17.24  | 14.15  |
> | 0.001  | 158.91  | 48.35 [32.71–73.83]   | 17.26  | 14.55  |
> | 0.0003 | 219.12  | 35.07 [23.65–52.75]   | 17.30  | 15.71  |
> | 0.0001 | 256.75  | 29.93 [28.50–31.43]   | 17.36  | 17.27  |
> | 3e-05  | 471.42  | 16.30 [15.62–17.05]   | 17.37  | 21.88  |
> | 1e-05  | 729.11  | 10.54 [10.10–11.00]   | 17.38  | 26.02  |
> | 3e-06  | 1088.95 | 7.06 [6.77–7.37]   | 17.39  | 35.00  |
> | 1e-06  | 2483.32 | 3.09 [2.96–3.26]   | 17.40  | 106.85 |
>
>
> **GenLM-Bytes:**
>
> | K   | Wall-clock time (s) | Throughput & 95% CI (bytes/s) | Peak GPU mem (GB) | Peak CPU mem (GB) |
> |-----|---------------------|-------------------------------|--------------------|--------------------|
> | 2   | 108.75  | 70.66 [68.34–72.08]  | 17.43  | 13.72  |
> | 4   | 116.44  | 65.99 [65.70–66.31]  | 17.46  | 14.15  |
> | 8   | 134.92  | 56.95 [56.67–57.23]  | 17.48  | 14.58  |
> | 16  | 171.16  | 44.89 [44.16–45.60]  | 17.51  | 15.01  |
> | 32  | 265.72  | 28.92 [28.14–29.67]  | 17.58  | 15.47  |
> | 64  | 527.94  | 14.55 [14.16–14.98]  | 17.69  | 15.98  |
> | 128 | 1081.32 | 7.11 [6.95–7.28]  | 17.97  | 16.57  |

---

> > ### Author Response · Authors · 2025-11-25
> >
> > #### Llama 3.2 1B:
> >
> > **Transduced LM:**
> >
> > | Threshold | Wall-clock time (s) | Throughput & 95% CI (bytes/s)   | Peak GPU mem (GB) | Peak CPU mem (GB) |
> > |-----------|---------------------|------------------------------------|--------------------|--------------------|
> > | 0.1 | 156.96  | 48.96 [40.21–60.65]   | 16.58  | 49.21  |
> > | 0.03   | 131.44  | 58.46 [47.92–71.15]   | 16.58  | 49.14  |
> > | 0.01   | 200.93  | 38.24 [24.44–59.97]   | 16.59  | 49.20  |
> > | 0.003  | 181.49  | 42.34 [26.56–70.33]   | 16.61  | 49.19  |
> > | 0.001  | 98.44   | 78.06 [63.49–94.45]   | 16.64  | 49.27  |
> > | 0.0003 | 113.25  | 67.85 [60.44–76.37]   | 16.69  | 49.96  |
> > | 0.0001 | 194.11  | 39.59 [31.19–47.38]   | 16.91  | 50.99  |
> > | 3e-05  | 284.31  | 27.03 [25.48–28.55]   | 16.91  | 53.22  |
> > | 1e-05  | 408.56  | 18.81 [18.00–19.74]   | 16.91  | 55.70  |
> > | 3e-06  | 612.70  | 12.54 [12.04–13.08]   | 16.92  | 60.37  |
> > | 1e-06  | 1063.57 | 7.22 [6.92–7.53]   | 16.93  | 72.36  |
> >
> >
> > **GenLM-Bytes:**
> >
> > | K   | Wall-clock time (s) | Throughput & 95% CI (bytes/s) | Peak GPU mem (GB) | Peak CPU mem (GB) |
> > | --- | ------------------- | ----------------------------- | ----------------- | ----------------- |
> > | 2   | 76.42   | 100.56 [96.37–103.25]   | 17.05 | 49.01 |
> > | 4   | 79.93   | 96.13 [95.63–96.59]  | 17.07 | 49.38 |
> > | 8   | 105.82  | 72.61 [72.17–73.05]  | 17.11 | 49.83 |
> > | 16  | 163.35  | 47.04 [45.82–48.07]  | 17.21 | 50.34 |
> > | 32  | 297.14  | 25.86 [25.17–26.53]  | 17.40 | 50.94 |
> > | 64  | 600.40  | 12.80 [12.46–13.14]  | 17.69 | 51.57 |
> > | 128 | 1125.84 | 6.83 [6.69–6.98]  | 18.45 | 52.40 |
> >
> >
> >
> >
> > ### Llama 3.1 8B:
> >
> > **Transduced LM:**
> >
> > | Threshold | Wall-clock time (s) | Throughput & 95% CI (bytes/s) | Peak GPU mem (GB) | Peak CPU mem (GB) |
> > |-----------|---------------------|--------------------------------|--------------------|--------------------|
> > | 0.1 | 144.84  | 53.05 [41.80–67.11]   | 16.53  | 49.42  |
> > | 0.03   | 125.91  | 61.03 [52.19–69.95]   | 16.54  | 49.17  |
> > | 0.01   | 171.11  | 44.91 [31.57–61.73]   | 16.55  | 49.14  |
> > | 0.003  | 175.78  | 43.71 [30.37–62.96]   | 16.55  | 49.15  |
> > | 0.001  | 186.96  | 41.10 [23.16–70.22]   | 16.58  | 49.30  |
> > | 0.0003 | 198.03  | 38.80 [22.10–63.95]   | 16.64  | 49.54  |
> > | 0.0001 | 169.07  | 45.45 [43.10–47.65]   | 16.72  | 50.23  |
> > | 3e-05  | 260.23  | 29.53 [28.13–30.87]   | 16.79  | 51.69  |
> > | 1e-05  | 366.53  | 20.96 [20.15–21.87]   | 16.87  | 53.85  |
> > | 3e-06  | 604.55  | 12.71 [11.94–13.47]   | 16.91  | 57.47  |
> > | 1e-06  | 1081.50 | 7.10 [6.85–7.37]   | 16.94  | 66.69  |
> >
> > **GenLM-Bytes:**
> >
> >
> > | K   | Wall-clock time (s) | Throughput & 95% CI (bytes/s) | Peak GPU mem (GB) | Peak CPU mem (GB) |
> > | --- | ------------------- | ----------------------------- | ----------------- | ----------------- |
> > | 2   | 170.51  | 45.07 [44.09–45.64]  | 17.03 | 49.07 |
> > | 4   | 189.38  | 40.57 [40.48–40.68]  | 17.05 | 49.41 |
> > | 8   | 244.02  | 31.49 [31.12–31.82]  | 17.09 | 49.81 |
> > | 16  | 363.80  | 21.12 [20.87–21.35]  | 17.19 | 50.26 |
> > | 32  | 648.89  | 11.84 [11.68–12.00]  | 17.38 | 50.77 |
> > | 64  | 1232.03 | 6.24 [6.15–6.33]  | 17.67 | 51.54 |
> > | 128 | 3197.48 | 2.40 [2.35–2.46]  | 18.42 | 52.45 |

---

### Official Review · Reviewer_CpW4 · 2025-11-01

**Soundness:** 4
**Presentation:** 4
**Contribution:** 4
**Rating:** 10
**Confidence:** 4

**Summary:**

The authors put forward an algorithm to sample and compute language model probabilities over transduced strings from the model's own distribution.
They also propose an approximate algorithm that estimates the transduced probabilities at a cheaper computational cost.

**Strengths:**

The idea is creative, well motivated, theoretically interesting, and practically useful.
This is a classic application of algorithms, probability, and formal language theory.
The examples in the appendix are informative and give a good intuition for how the method works.

**Weaknesses:**

I am quite satisfied with the presentation and ideas in this paper and do not see any significant reasons why it should not be accepted.

**Questions:**

Does this method potentially allow conversion between different language model tokenizers? This could be useful for cross-family language model distillation.

---

> ### Author Response · Authors · 2025-11-25
>
> Thank you for the thoughtful review!  We are very pleased that you found this work well-motivated, interesting, and practical!
>
>
> ### Question: Does this method potentially allow conversion between different language model tokenizers? This could be useful for cross-family language model distillation.
>
> Yes! Our method enables conversion between different tokenizers and, thus, cross-family distillation. Concretely, we represent each tokenizer’s encode and decode functions as FSTs (see Berglund et al., 2024, Cognetta et al., 2025, Vieira et al., 2025, for details on constructing these machines for BPE).
> We can then map each model’s subword tokens to bytes using a transformation $T_{\tau\to\alpha}$. Alternatively, we can choose one of the tokenizers as the target tokenizer: if $\tau_1$ is the target tokenizer we can apply $T_{\tau_2\to\alpha} \circ T_{\alpha\to\tau_1}$ to bring the model over $\tau_2$’s tokens into the space of the model over $\tau_1$’s tokens.
>
> One can imagine targeted distillation (or regularization) from a model that is superior on some tasks, and using another model for other tasks. Similarly, as reviewer **bhMG** suggested, we foresee approaches such as contrastive decoding, speculative decoding, or ensambling being made possible between model families. We very much look forward to seeing this kind of application in future work.

---

### Official Review · Reviewer_bhMG · 2025-11-01

**Soundness:** 4
**Presentation:** 3
**Contribution:** 3
**Rating:** 8
**Confidence:** 3

**Summary:**

The paper is about transduction -- taking a language model that was trained over a certain granularity of vocabulary and converting it to a model operating over a different granularity of vocabulary through a deterministic mapping. It extends prior work on converting subword models to character-level models by providing both exact and reasonable approximate algorithms for converting to arbitrary new vocabularies, then demonstrates this approach on three example transformations.

**Strengths:**

S1. I think this is a really interesting problem, and it's both well-explained and well-explored in this work. I can immediately see how this would be useful.

S2. I really like the framing of this with FSTs; it's an intuitive way to think about the class of transformations and a really nice formalism. The walkthrough of the math is detailed and cleanly developed.

**Weaknesses:**

W1. While overall I think the explanation is good, there are several parts of the paper that I think require context from "From Language Models over Tokens to Language Models over Characters" to fully understand. The two most obvious ones to me: (1) the discussion in lines 67-69 of applications to computational psycholinguistics and controlled generation; (2) the relation between transducing and token healing.

W2. The work is largely a generalization of a prior (also cool!) result. I think this is still a valid contribution, but I could see the argument that this is a bit incremental.

W3. While the speed of decoding is not a major focus of the work, it would be nice to see a bit more detail on the efficiency. In particular, bytes/sec is a bit hard to interpret, and a few baselines of the language model decoding in its original vocabulary would be helpful to put these numbers in context.

**Questions:**

Q1. This defines a fairly broad class of transductions that can be performed exactly; do you have any examples of reasonable-to-want transformations that *don't* fall into this class?

Q2. One thing that I don't see mentioned here is that this makes it much easier to compare distributions across language models-- you could transduce model 1 into model 2's vocabulary space. Do you think this would ever be practical to do e.g. for speculative or contrastive decoding?

Comments/other notes:
- purely a stylistic gripe, but I think the formatting for library names (e.g. GenLM.bytes) is super distracting to read in-line.
- The color of symbols is also slightly annoying to me, but I see the argument for that as a way to track notation; I would just confirm they are colorblind-friendly where possibly
- typo in line 144: "efficienct"
- line 41: "Such as normalizing output, ..." is a sentence fragment
- line 850: "proportion" -> "portion"
- the introduction was much less dense relative to the rest of the paper-- I think the example could have been greatly condensed to spend a bit more main-body text time on the transduction
- the trained model for transducing seems a bit out of place in the rest of the work; it doesn't weaken the paper, but it feels a bit like a last-minute (unnecessary) addition to satisfy a reviewer

---

> ### Author Response · Authors · 2025-11-25
>
> We are glad you find the problem interesting and that you can see its practicality! We respond to each of your comments below.
>
>
> ### W1: Better explanations of psycholinguistics, controlled decoding, and token healing.
>
> Good points. We’ll add a reference to the related work in lines 67-69 that mention psycholinguistics. We briefly describe the psycholinguistic setup in Appendix H in lines 1243 and below, and we will expand on the controlled generation to provide more context. We will also point to the prompt-boundary problem that token healing (modifying the prompt+continuation to be tokenized jointly) aims to solve by referring to the description in the prior work by Viera et al.: the solution is implicit in e.g. the tokens-to-byte conversion, since we marginalize over all subword variants.
>
> ### W2: Difference to prior work “From Language Models over Tokens to Language Models over Characters”
>
> Only one of the three experiments in the paper is covered by the prior work, the strict-prefix monotone conversion from tokens to bytes. The conversion to amino acids from DNA is only  prefix-monotone (not strictly) and not covered by prior work; the conversion to orthographic words is also not prefix-monotone at all and thus also not covered by the prior work. Put simply, the prior work does not support scanning multiple input symbols to determine which output symbol to emit; our work introduces the theory for doing so, enabling a much broader class of useful transformations.
>
> ### W3. While the speed of decoding is not a major focus of the work, it would be nice to see a bit more detail on the efficiency. In particular, bytes/sec is a bit hard to interpret, and a few baselines of the language model decoding in its original vocabulary would be helpful to put these numbers in context.
>
> We give a direct comparison below using the same configuration as the previous numbers over the same ten Wikipedia paragraphs. These numbers are around 10-100x (in the extreme case) faster than the transduced language model, since no transducer operations are needed, there are multiple bytes in each subword, and no marginalization takes place. Note that the base model is also 10-100x faster than the numbers reported in “From Language Models over Tokens to Language Models over Characters”. We aim to narrow this gap in future work by, e.g., implementing our algorithm using matrix operations and considering sampling methods rather than full enumeration, as we discuss in the limitations section. Nevertheless, our general algorithm for supporting both prefix-monotone and non-prefix-monotone transformations is inherently more computationally expensive, as it requires additional lookahead for future symbols and powerstate tracking for lazy FST determinization.
>
> | Base Model  | Wall-Clock Time (s) | Throughput (bytes/s) | 95% CI (bytes/s) |
> |--------------|----------------------|------------------------|-----------------------------|
> | GPT-2 Large  | 21.52 | 355.56  | [330.49, 373.66]   |
> | LLaMA-1B  | 12.10 | 634.89  | [480.09, 758.96]   |
> | LLaMA-8B  | 33.72 | 227.80  | [217.93, 233.50]   |
>
>
>
> ### Q1: This defines a fairly broad class of transductions that can be performed exactly; do you have any examples of reasonable-to-want transformations that don't fall into this class?
>
> Many transformations require a more expressive formalism; for instance, deterministic translation engines, such as French-to-English translation or converting MATLAB code to Python, already require more than finite-state reasoning. Moreover, there are simple transductions that are not FST-definable (e.g., string reversal), and even among FSTs, there are simple examples (e.g., parity) that do not admit efficient decompositions in our framework. We would need to use unbounded memory or an infinite number of states to handle these more complicated relations.
>
> ### Q2: One thing that I don't see mentioned here is that this makes it much easier to compare distributions across language models-- you could transduce model 1 into model 2's vocabulary space. Do you think this would ever be practical to do e.g. for speculative or contrastive decoding?
>
>
> Excellent point: since we can transform one language model into another model's language (by choosing tokens), we can directly compare them on the same units. This enables applications such as contrastive decoding, ensemble approaches, distillation, and speculative decoding between models; we look forward to seeing these in practice.
>
> ### Notes
> Thank you, we will take all of these notes into consideration in the next revision!

---

### Author Response · Authors · 2025-12-02

We thank the reviewers for their time and their very positive evaluation of our work and kind words. The reviewers found the work interesting, useful, intuitive and well presented.
We received questions about decoding speed and memory consumption, to which we responded below. One reviewer also asked about more expressive transformations than those encoded by transducers, to which we responded that we will consider these in future work. The same reviewer also pointed out the relation to neural FSTs, which we acknowledged while also pointing out key differences

---

### Meta-Review · Area_Chair_RVfW · 2026-01-06

**Summary:**

The paper introduces a principled framework for transducing a pretrained language model defined over one tokenization into a proper functional LM over another/different alphabet via deterministic finite-state transducers (FSTs), with (i) an exact method, (ii) a practical approximate method, and (iii) analysis plus experiments across three meaningful domains.

**Reviewer Concerns:**

Three of the reviewers had only minor concerns, most of which seemed to have been addressed during rebuttal. One outstanding concern from those reviewers was the incremental contribution of the paper considering its strong technical overlap with "From Language Models over Tokens to Language Models over Characters". The authors responded to this concern, but the reviewer did not respond to that rebuttal in time. Yet, despite the distinctions pointed out by the authors, I believe the concern largely remains.

The reviewer with more concerns/weaknesses focused their criticism on the scope/claims of generality beyond FST, and comparison with other classic FST + neural hybrid methods in NLP. They seemed to have been satisfied with the authors' response but did not immediately raise their score.

**Reviewer Scores:**

My best guess is that bhMG would not have changed their score, and that Sa4U might have increased it to 8.

---

### Decision · Program_Chairs · 2026-01-26

Accept (Poster)